# Best Arm Identification with Fixed Budget:
# A Large Deviation Perspective

**Po-An Wang**
EECS
KTH, Stockholm, Sweden
wang9@kth.se

**Ruo-Chun Tzeng**
EECS
KTH, Stockholm, Sweden
rctzeng@kth.se

**Alexandre Proutiere**
EECS and Digital Futures
KTH, Stockholm, Sweden
alepro@kth.se

## Abstract

We consider the problem of identifying the best arm in stochastic Multi-Armed Bandits (MABs) using a fixed sampling budget. Characterizing the minimal instance-specific error probability for this problem constitutes one of the important remaining open problems in MABs. When arms are selected using a static sampling strategy, the error probability decays exponentially with the number of samples at a rate that can be explicitly derived via Large Deviation techniques. Analyzing the performance of algorithms with adaptive sampling strategies is however much more challenging. In this paper, we establish a connection between the Large Deviation Principle (LDP) satisfied by the empirical proportions of arm draws and that satisfied by the empirical arm rewards. This connection holds for any adaptive algorithm, and is leveraged (i) to improve error probability upper bounds of some existing algorithms, such as the celebrated SR (Successive Rejects) algorithm (Audibert et al., 2010), and (ii) to devise and analyze new algorithms. In particular, we present CR (Continuous Rejects), a truly adaptive algorithm that can reject arms in *any* round based on the observed empirical gaps between the rewards of various arms. Applying our Large Deviation results, we prove that CR enjoys better performance guarantees than existing algorithms, including SR. Extensive numerical experiments confirm this observation.

## 1 Introduction

We study the problem of best-arm identification in stochastic bandits in the fixed budget setting. In this problem, abbreviated by BAI-FB, a learner faces $K$ distributions or arms $\nu_1, \ldots, \nu_K$ characterized by their unknown means $\boldsymbol{\mu} = (\mu_1, \ldots, \mu_K)$ (we restrict our attention to distributions taken from a one-parameter exponential family). She sequentially pulls arms and observes samples of the corresponding distributions. More precisely, in round $t \geq 1$, she pulls an arm $A_t = k$ selected depending on previous observations and observes $X_k(t)$ a sample of a $\nu_k$-distributed random variable. $(X_k(t), t \geq 1, k \in [K])$ are assumed to be independent over rounds and arms. After $T$ arm draws, the learner returns $\hat{\imath}$, an estimate of the best arm $1(\boldsymbol{\mu}) := \arg\max_k \mu_k$. We assume that the best arm is unique, and denote by $\Lambda$ the set of parameters $\boldsymbol{\mu}$ such that this assumption holds. The objective is to devise an adaptive sampling algorithm minimizing the error probability $\mathbb{P}_{\boldsymbol{\mu}}[\hat{\imath} \neq 1(\boldsymbol{\mu})]$. This learning task is one of the most important problems in stochastic bandits, and despite recent research efforts, it remains largely open (Qin, 2022). In particular, researchers have so far failed

37th Conference on Neural Information Processing Systems (NeurIPS 2023).

at characterizing the minimal instance-specific error probability. This contrasts with other basic learning tasks in stochastic bandits such as regret minimization (Lai and Robbins, 1985) and BAI with fixed confidence (Garivier and Kaufmann, 2016), for which indeed, asymptotic instance-specific performance limits and matching algorithms have been derived. In BAI-FB, the error probability typically decreases exponentially with the sample budget $T$, i.e., it scales as $\exp(-R(\boldsymbol{\mu})T)$ where the instance-specific rate $R(\boldsymbol{\mu})$ depends on the sampling algorithm. Maximizing this rate over the set of adaptive algorithms is an open problem.

**Instance-specific error probability lower bound.** To guess the maximal rate at which the error probability decays, one may apply the same strategy as that used in regret minimization or BAI in the fixed confidence setting: (i) derive instance-specific lower bound for the error probability for some notion of *uniformly good* algorithms; (ii) devise a sampling strategy mimicking the optimal proportions of arm draws identified in the lower bound. Here the notion of uniformly good algorithms is that of *consistent* algorithms. Under such an algorithm, for any $\boldsymbol{\mu} \in \Lambda$, $\mathbb{P}_{\boldsymbol{\mu}}[\hat{\imath} = 1(\boldsymbol{\mu})] \to 1$ as $T \to \infty$. (Garivier and Kaufmann, 2016) conjectures the following asymptotic lower bound satisfied by any consistent algorithm (refer to Appendix J for details): as $T \to \infty$,

$$\frac{1}{T} \log \frac{1}{\mathbb{P}_{\boldsymbol{\mu}}[\hat{\imath} \neq 1(\boldsymbol{\mu})]} \leq \max_{\boldsymbol{\omega} \in \Sigma} \inf_{\boldsymbol{\lambda} \in \mathrm{Alt}(\boldsymbol{\mu})} \Psi(\boldsymbol{\lambda}, \boldsymbol{\omega}), \tag{1}$$

where $\Sigma$ is the $(K-1)$-dimensional simplex, $\Psi(\boldsymbol{\lambda}, \boldsymbol{\omega}) = \sum_{k=1}^{K} \omega_k d(\lambda_k, \mu_k)$, $\mathrm{Alt}(\boldsymbol{\mu}) = \{\boldsymbol{\lambda} \in \Lambda : 1(\boldsymbol{\mu}) \neq 1(\boldsymbol{\lambda})\}$ is the set of confusing parameters (those for which $1(\boldsymbol{\mu})$ is not the best arm), and $d(x, y)$ denotes the KL divergence between two distributions of parameters $x$ and $y$. Interestingly, the solution $\boldsymbol{\omega}^{\star} \in \Sigma$ of the optimization problem $\max_{\boldsymbol{\omega} \in \Sigma} \inf_{\boldsymbol{\lambda} \in \mathrm{Alt}(\boldsymbol{\mu})} \Psi(\boldsymbol{\lambda}, \boldsymbol{\omega})$ provides the best static proportions of arm draws. More precisely, an algorithm selecting arms according to the allocation $\boldsymbol{\omega}^{\star}$, i.e., selecting arm $k$ $\omega_k^{\star}T$ times and returning the best empirical arm after $T$ samples, has an error rate matching the lower bound (1). This is a direct consequence of the fact that, under a static algorithm with allocation $\boldsymbol{\omega}$, the empirical reward process $\{\hat{\boldsymbol{\mu}}(t)\}_{t \geq 1}$ satisfies a LDP with rate function $\boldsymbol{\lambda} \mapsto \Psi(\boldsymbol{\lambda}, \boldsymbol{\omega})$, see (Glynn and Juneja, 2004) and refer to Section 3 for more details.

**Adaptive sampling algorithms and their analysis.** The optimal allocation $\boldsymbol{\omega}^{\star}$ depends on the instance $\boldsymbol{\mu}$ and is initially unknown. We may devise an adaptive sampling algorithm that (i) estimates $\boldsymbol{\omega}^{\star}$ and (ii) tracks this estimated optimal allocation. In the BAI with fixed confidence, such tracking scheme exhibits asymptotically optimal performance (Garivier and Kaufmann, 2016). Here however, the error made estimating $\boldsymbol{\omega}^{\star}$ would inevitably impact the overall error probability of the algorithm. To quantify this impact or more generally to analyze the performance of adaptive algorithms, one would need to understand the connection between the statistical properties of the arm selection process and the asymptotic statistics of the estimated expected rewards.

To be more specific, any adaptive algorithm generates a stochastic process $\{Z(t)\}_{t \geq 1} = \{(\boldsymbol{\omega}(t), \hat{\boldsymbol{\mu}}(t))\}_{t \geq 1}$. $\boldsymbol{\omega}(t) = (\omega_1(t), \dots, \omega_K(t))$ represents the allocation realized by the algorithm up to round $t$ ($\omega_k(t) = N_k(t)/t$ and $N_k(t)$ denotes the number of times arm $k$ has been selected up to round $t$). $\hat{\boldsymbol{\mu}}(t) = (\hat{\mu}_1(t), \dots, \hat{\mu}_K(t))$ denotes the empirical average rewards of the various arms up to round $t$. Now assuming that at the end of round $T$, the algorithm returns the arm with the highest empirical reward, the error probability is $\mathbb{P}_{\boldsymbol{\mu}}[\hat{\imath} \neq 1(\boldsymbol{\mu})] = \mathbb{P}_{\boldsymbol{\mu}}[\hat{\boldsymbol{\mu}}(T) \in \mathrm{Alt}(\boldsymbol{\mu})]$. Assessing the error probability at least asymptotically requires understanding the asymptotic behavior of $\hat{\boldsymbol{\mu}}(t)$ as $t$ grows large. Ideally, one would wish to establish the Large Deviation properties of the process $\{Z(t)\}_{t \geq 1}$. This task is easy for algorithms using static allocations (Glynn and Juneja, 2004), but becomes challenging and open for adaptive algorithms. Addressing this challenge is the main objective of this paper.

**Contributions.** In this paper, we develop and leverage tools towards the analysis of adaptive sampling algorithms for the BAI-FB problem. More precisely, our contributions are as follows.

(a) We establish a connection between the LDP satisfied by the empirical proportions of arm draws $\{\boldsymbol{\omega}(t)\}_{t \geq 1}$ and that satisfied by the empirical arm rewards. This connection holds for any adaptive algorithm. Specifically, we show that if the rate function of $\{\boldsymbol{\omega}(t)\}_{t \geq 1}$ is lower bounded by $\boldsymbol{\omega} \mapsto I(\boldsymbol{\omega})$, then that of $(\hat{\boldsymbol{\mu}}(t))_{t \geq 1}$ is also lower bounded by $\boldsymbol{\lambda} \mapsto \min_{\boldsymbol{\omega} \in \Sigma} \max\{\Psi(\boldsymbol{\lambda}, \boldsymbol{\omega}), I(\boldsymbol{\omega})\}$. This result has interesting interpretations and implies the following asymptotic upper bound on the error probability of the algorithm considered: as $T \to \infty$,

$$\frac{1}{T} \log \frac{1}{\mathbb{P}_{\boldsymbol{\mu}}[\hat{\imath} \neq 1(\boldsymbol{\mu})]} \geq \inf_{\boldsymbol{\omega} \in \Sigma, \boldsymbol{\lambda} \in \mathrm{Alt}(\boldsymbol{\mu})} \max\{\Psi(\boldsymbol{\lambda}, \boldsymbol{\omega}), I(\boldsymbol{\omega})\}. \tag{2}$$

The above formula, when compared to the lower bound (1), quantifies the price of not knowing $\omega^\star$ initially, and relates the error probability to the asymptotic statistics of the sampling process used by the algorithm.

(b) We show that by simply applying our generic Large Deviation result, we may improve the error probability upper bounds of some existing algorithms, such as the celebrated SR algorithm (Audibert et al., 2010). Our result further opens up opportunities to devise and analyze new algorithms with a higher level of adaptiveness. In particular, we present CR (Continuous Rejects), an algorithm that, unlike SR, can eliminate arms in *each* round. This sequential elimination process is performed by comparing the empirical rewards of the various candidate arms using continuously updated thresholds. Leveraging the LDP tools developed in (a), we establish that CR enjoys better performance guarantees than SR. Hence CR becomes the algorithm with the lowest instance-specific and guaranteed error probability. We illustrate our results via numerical experiments, and compare CR to other BAI algorithms.

## 2  Related Work

We distinguish two main classes of algorithms to solve the best arm identification problem in the fixed budget setting. Algorithms from the first class, e.g. Successive Rejects (SR) (Audibert et al., 2010) and Sequential Halving (SH) (Karnin et al., 2013), split the sampling budget into phases of fixed durations, and discard arms at the end of each phase. Algorithms from the second class, e.g. UCB-E (Audibert et al., 2010) and UGapE (Gabillon et al., 2012) sequentially sample arms based on confidence bounds of their empirical rewards. It is worth mentioning that algorithms from the second class usually require some prior knowledge about the problem, for example, an upper bound of $H = \sum_{k \neq 1(\boldsymbol{\mu})} \frac{1}{(\mu_{1(\boldsymbol{\mu})} - \mu_k)^2}$. Without this knowledge, the parameters can be chosen in a heuristic way, but the performance gets worse.

Algorithms from the first class exhibit better performance numerically and are also those with the best instance-specific error probability guarantees. SR had actually the best performance guarantees so far: for example, when the reward distributions are supported on $[0, 1]$, the error probability of SR satisfies: $\liminf_{T \to \infty} \frac{1}{T} \log \frac{1}{\mathbb{P}_{\boldsymbol{\mu}}[\hat{i} \neq 1(\boldsymbol{\mu})]} \geq \frac{1}{H_2 \log K}$, where $H_2 = \max_{k \neq 1(\boldsymbol{\mu})} \frac{k}{(\mu_{1(\boldsymbol{\mu})} - \mu_k)^2}$. In this paper, we strictly improve this guarantee (see Section 3.4). Recently, (Barrier et al., 2022) also refined and extended the analysis of (Audibert et al., 2010) by replacing, in the analysis, Hoeffding's inequality by a large deviation result involving KL-divergences. Again here, we further improve this new guarantee. We also devise CR, an algorithm with error probability provably lower than our improved guarantees for SR.

Fundamental limits on the error probability have also been investigated. In the minimax setting, (Carpentier and Locatelli, 2016) established that for any algorithm, there exists a problem within the class of instances with given complexity $H$ such that the error probability is greater than $\exp(-\frac{400T}{H \log K}) \geq \exp(-\frac{400T}{H_2 \log K})$. Up to a universal constant (here 400), SR is hence minimax optimal. This lower bound was also recently revisited in (Ariu et al., 2021; Degenne, 2023; Wang et al., 2023) to prove that the instance-specific lower bound (1) cannot be achieved on all instances by a single algorithm . Deriving tight instance-specific lower bounds remains open (Qin, 2022).

We conclude this section by mentioning two interesting algorithms. In (Komiyama et al., 2022), the authors propose DOT, an algorithm trying to match minimax error probability lower bounds. To this aim, the algorithm requires to periodically call an oracle able to determine an optimal allocation, solution of an optimization problem with high and unknown complexity. DOT has minimax guarantees but is computationally challenging if not infeasible (numerically, the authors cannot go beyond simple instances with 3 arms). Finally, researchers have also looked at the best arm identification problem from a Bayesian perspective. For example, (Russo, 2016) devise variants of the celebrated Thompson Sampling algorithm, that could potentially work well in practice. Nevertheless, as discussed in (Komiyama, 2022), Bayesian algorithms cannot be analyzed nor provably perform well in the frequentist setting.

# 3 Large Deviation Analysis of Adaptive Sampling Algorithms

In this section, we first recall key concepts in Large Deviations (refer to the classical textbooks (Budhiraja and Dupuis, 2019; Dembo and Zeitouni, 2009; Dupuis and Ellis, 2011; Varadhan, 2016) for a more detailed exposition). We then apply these concepts to the performance analysis of adaptive sampling algorithms. Finally, we exemplify the analysis and apply it to improve existing performance guarantees for the SR algorithm (Audibert et al., 2010).

## 3.1 Large Deviation Principles

Consider the stochastic process $\{Y(t)\}_{t\geq 1}$ with values in a separable complete metric space (i.e., a Polish space) $\mathcal{Y}$. Large Deviations are concerned with the probabilities of rare events related to $\{Y(t)\}_{t\geq 1}$ that decay exponentially in the parameter $t$. The asymptotic decay rate is characterized by the *rate function* $I : \mathcal{Y} \to \mathbb{R}_+$ defined so that essentially $-\frac{1}{t}\log\mathbb{P}\left[Y(t) \in B\right]$ converges to $\min_{x\in B} I(x)$ for any Borel set $B$. We provide a more rigorous definition below.

**Definition 1.** *[Large Deviation Principle (LDP)] The stochastic process $\{Y(t)\}_{t\geq 1}$ satisfies a LDP with rate function $I$ if:*
*(i) $I$ is lower semicontinuous, and $\forall s \in [0,\infty]$, the set $\mathcal{K}_s = \{y \in \mathcal{Y} : I(y) \leq s\}$ is compact;*
*(ii) for every closed (resp. open) set $C \subset \mathcal{Y}$ (resp. $O \subset \mathcal{Y}$),*

$$\varliminf_{t\to\infty} \frac{1}{t}\log\frac{1}{\mathbb{P}\left[Y(t) \in C\right]} \geq \inf_{y\in C} I(y), \tag{3}$$

$$\varlimsup_{t\to\infty} \frac{1}{t}\log\frac{1}{\mathbb{P}\left[Y(t) \in O\right]} \leq \inf_{y\in O} I(y). \tag{4}$$

LDPs have been derived earlier in stochastic bandit literature. (Glynn and Juneja, 2004) have used Gärtner-Ellis Theorem (Ellis, 1984; Gärtner, 1977) to establish that under a static sampling algorithm with allocation $\boldsymbol{\omega} \in \Sigma$ (i.e., each arm $k$ is selected $\omega_k T$ times up to round $T$), the process $\{\hat{\boldsymbol{\mu}}(T)\}_{T\geq 1}$ satisfies an LDP with rate function $\boldsymbol{\lambda} \mapsto \Psi(\boldsymbol{\lambda}, \boldsymbol{\omega}) = \sum_{k=1}^{K} \omega_k d(\lambda_k, \mu_k)$. Our objective in the next subsection is to investigate how to extend this result to the case of adaptive sampling algorithms.

## 3.2 Analysis of adaptive sampling algorithms

An adaptive sampling algorithm generates a stochastic process $\{Z(t)\}_{t\geq 1} = \{(\boldsymbol{\omega}(t), \hat{\boldsymbol{\mu}}(t))\}_{t\geq 1}$. When the sampling budget $T$ is exhausted, should the algorithm returns the arm with the highest empirical reward, the error probability is $\mathbb{P}_{\boldsymbol{\mu}}[\hat{\imath} \neq 1(\boldsymbol{\mu})] = \mathbb{P}_{\boldsymbol{\mu}}[\hat{\boldsymbol{\mu}}(T) \in \text{Alt}(\boldsymbol{\mu})]$. To assess the rate at which this probability decays with the budget, we may try to establish a LDP for the empirical reward process $\{\hat{\boldsymbol{\mu}}(t)\}_{t\geq 1}$. Due to the intricate dependence between the sampling and the empirical reward processes, deriving such an LDP is very challenging. Instead, we establish a connection between the LDPs satisfied by these processes. This connection will be enough for us to derive tight upper bounds on the error probability. We present our main result in the following theorem.

**Theorem 1.** *Assume that under some adaptive sampling algorithm, $\{\boldsymbol{\omega}(t)\}_{t\geq 1}$ satisfies the LDP upper bound (3) with rate function $I$. Then $\{\hat{\boldsymbol{\mu}}(t)\}_{t\geq 1}$ satisfies the LDP upper bound (3) with rate function $\boldsymbol{\lambda} \mapsto \min_{\boldsymbol{\omega}\in\Sigma} \max\{\Psi(\boldsymbol{\lambda}, \boldsymbol{\omega}), I(\boldsymbol{\omega})\}$. Moreover, we have: for any bounded Borel subset $\mathcal{S}$ of $\mathbb{R}^K$ and any Borel subset $W$ of $\Sigma$,*

$$\varliminf_{t\to\infty} \frac{1}{t}\log\frac{1}{\mathbb{P}_{\boldsymbol{\mu}}\left[\hat{\boldsymbol{\mu}}(t) \in \mathcal{S}, \boldsymbol{\omega}(t) \in W\right]} \geq \inf_{\boldsymbol{\omega}\in\text{cl}(W)} \max\left\{F_{\mathcal{S}}(\boldsymbol{\omega}), I(\boldsymbol{\omega})\right\},$$

*where $F_{\mathcal{S}}(\boldsymbol{\omega}) := \inf_{\boldsymbol{\lambda}\in\text{cl}(\mathcal{S})} \Psi(\boldsymbol{\lambda}, \boldsymbol{\omega})$, and $\text{cl}(\mathcal{S})$ denotes the closure of $\mathcal{S}$.*

Before proving the above theorem, we make the following remarks and provide a simple corollary that will lead to improved upper bound on the error probability of the SR algorithm.

*(a) Not a complete LDP.* To upper bound the error probability of a given algorithm, we do not actually need to establish that $\{\hat{\boldsymbol{\mu}}(t)\}_{t\geq 1}$ satisfies a complete LDP. Instead, deriving a LDP upper bound is enough. Theorem 1 provides such an upper bound, but does not yield a complete LDP. We conjecture if $\{\boldsymbol{\omega}(t)\}_{t\geq 1}$ satisfies an LDP with rate function $I$, $\{\hat{\boldsymbol{\mu}}(t)\}_{t\geq 1}$ satisfies an LDP with rate function $\boldsymbol{\lambda} \mapsto \inf_{\boldsymbol{\omega}\in W} \max\{\Psi(\boldsymbol{\lambda}, \boldsymbol{\omega}), I(\boldsymbol{\omega})\}$. The conjecture holds for static sampling algorithms as shown

below. If it holds for adaptive algorithms, we show, in Appendix I, that when $K > 3$, no algorithm can attain the instance-specific lower bound (1) for all parameters.

*(b) Theorem 1 is tight for static sampling algorithms.* When the sampling rule is static, namely $\boldsymbol{\omega}(t) = \boldsymbol{\omega} \in \Sigma$, then $\{\boldsymbol{\omega}(t)\}_{t\geq 1}$ satisfies a LDP with rate function $I$ defined as $I(\boldsymbol{\omega}) = 0$ and $\infty$ elsewhere. Theorem 1 with $W = \Sigma$ states that $\{\hat{\boldsymbol{\mu}}(t)\}_{t\geq 1}$ satisfies the LDP upper bound (3) with rate function $F_{\mathcal{S}}$. In fact, as shown by (Glynn and Juneja, 2004), $\{\hat{\boldsymbol{\mu}}(t)\}_{t\geq 1}$ satisfies a complete LDP with this rate function.

*(c) A useful corollary.* From Theorem 1, we have:

$$\lim_{t\to\infty} \frac{1}{t} \log \frac{1}{\mathbb{P}_{\boldsymbol{\mu}}\left[\hat{\boldsymbol{\mu}}(t) \in \mathcal{S}, \boldsymbol{\omega}(t) \in W\right]} \geq \inf_{\boldsymbol{\omega} \in \mathrm{cl}(W)} F_{\mathcal{S}}(\boldsymbol{\omega}).$$

From there, we will be able to improve the performance guarantee for SR.

### 3.3 Proof of Theorem 1

*Proof.* Observe that when $\mathcal{S}$ or $W$ is empty, the result holds. Now recall that $F_{\mathcal{S}}(\cdot) = \inf_{\boldsymbol{\lambda} \in \mathrm{cl}(\mathcal{S})} \Psi(\boldsymbol{\lambda}, \cdot)$ is the infimum of a family of linear functions on a compact set, $\Sigma$, hence it is upper bounded. Denote $u > 0$ such an upper bound. $F_{\mathcal{S}}(\cdot)$ is also continuous (see Appendix F for details). For each integer $N \in \mathbb{N}$, we define a collection of closed sets:

$$W_n^N = \left\{\boldsymbol{\omega} \in \mathrm{cl}(W) : \frac{u(n-1)}{N} \leq F_{\mathcal{S}}(\boldsymbol{\omega}) \leq \frac{un}{N}\right\}, \quad \forall n \in [N]. \tag{5}$$

We observe that:

$$\mathbb{P}_{\boldsymbol{\mu}}[\hat{\boldsymbol{\mu}}(t) \in \mathcal{S}, \boldsymbol{\omega}(t) \in W] \leq \sum_{n=1}^{N} \mathbb{P}_{\boldsymbol{\mu}}[\hat{\boldsymbol{\mu}}(t) \in \mathcal{S}, \boldsymbol{\omega}(t) \in W_n^N]$$
$$\leq N \max_{n\in[N]} \mathbb{P}_{\boldsymbol{\mu}}[\hat{\boldsymbol{\mu}}(t) \in \mathcal{S}, \boldsymbol{\omega}(t) \in W_n^N].$$

Taking the logarithm on both sides and dividing them by $-t$ yields that

$$\lim_{t\to\infty} \frac{1}{t} \log \frac{1}{\mathbb{P}_{\boldsymbol{\mu}}\left[\hat{\boldsymbol{\mu}}(t) \in \mathcal{S}, \boldsymbol{\omega}(t) \in W\right]} \geq \lim_{t\to\infty} \min_{n\in[N]} \frac{1}{t} \log \frac{1}{\mathbb{P}_{\boldsymbol{\mu}}\left[\hat{\boldsymbol{\mu}}(t) \in \mathcal{S}, \boldsymbol{\omega}(t) \in W_n^N\right]}$$
$$= \min_{n\in[N]} \lim_{t\to\infty} \frac{1}{t} \log \frac{1}{\mathbb{P}_{\boldsymbol{\mu}}\left[\hat{\boldsymbol{\mu}}(t) \in \mathcal{S}, \boldsymbol{\omega}(t) \in W_n^N\right]}$$
$$\geq \min_{n\in[N]} \max\left\{\frac{u(n-1)}{N}, \inf_{\boldsymbol{\omega}\in W_n^N} I(\boldsymbol{\omega})\right\}, \tag{6}$$

where the last inequality follows from Lemma 1. Since for all $n \in [N]$,

$$\max\left\{\frac{u(n-1)}{N}, \inf_{\boldsymbol{\omega}\in W_n^N} I(\boldsymbol{\omega})\right\} = \inf_{\boldsymbol{\omega}\in W_n^N} \max\left\{\frac{u(n-1)}{N}, I(\boldsymbol{\omega})\right\},$$

the r.h.s. of (6) is equal to

$$\min_{n\in[N]} \inf_{\boldsymbol{\omega}\in W_n^N} \max\left\{\frac{u(n-1)}{N}, I(\boldsymbol{\omega})\right\} \geq \min_{n\in[N]} \inf_{\boldsymbol{\omega}\in W_n^N} \max\left\{F_{\mathcal{S}}(\boldsymbol{\omega}), I(\boldsymbol{\omega})\right\} - u/N$$
$$= \inf_{\boldsymbol{\omega}\in\mathrm{cl}(W)} \max\{F_{\mathcal{S}}(\boldsymbol{\omega}), I(\boldsymbol{\omega})\} - u/N,$$

where the first inequality is due to (5). As $N$ can be taken arbitrarily large, we conclude this theorem. $\square$

**Lemma 1.** *For any $N \in \mathbb{N}$, $n \in [N]$,*

$$\lim_{t\to\infty} \frac{1}{t} \log \frac{1}{\mathbb{P}_{\boldsymbol{\mu}}[\hat{\boldsymbol{\mu}}(t) \in \mathcal{S}, \boldsymbol{\omega}(t) \in W_n^N]} \geq \max\left\{\frac{u(n-1)}{N}, \inf_{\boldsymbol{\omega}\in W_n^N} I(\boldsymbol{\omega})\right\}$$

*Proof.* Recall $F_{\mathcal{S}}(\cdot) = \inf_{\boldsymbol{\lambda} \in \mathrm{cl}(\mathcal{S})} \Psi(\boldsymbol{\lambda}, \cdot)$. We deduce that

$$\mathbb{P}_{\boldsymbol{\mu}}\left[\hat{\boldsymbol{\mu}}(t) \in \mathcal{S}, \boldsymbol{\omega}(t) \in W_n^N\right] \leq \mathbb{P}_{\boldsymbol{\mu}}\left[X \geq tF_S(\boldsymbol{\omega}(t)), \boldsymbol{\omega}(t) \in W_n^N\right],$$

where $X$ denotes $t\Psi(\hat{\boldsymbol{\mu}}(t), \boldsymbol{\omega}(t))$ for short. Let $\alpha \in (0, 1)$, Markov's inequality implies that

$$\begin{aligned}
\mathbb{P}_{\boldsymbol{\mu}}\left[X \geq tF_S(\boldsymbol{\omega}), \boldsymbol{\omega}(t) \in W_n^N\right] &= \mathbb{P}_{\boldsymbol{\mu}}\left[\mathbb{1}\{\boldsymbol{\omega}(t) \in W_n^N\}e^{\alpha(X - tF_S(\boldsymbol{\omega}(t)))} \geq 1\right] \\
&\leq \mathbb{E}_{\boldsymbol{\mu}}\left[\mathbb{1}\{\boldsymbol{\omega}(t) \in W_n^N\}e^{\alpha(X - tF_S(\boldsymbol{\omega}(t)))}\right] \\
&\leq \mathbb{E}_{\boldsymbol{\mu}}\left[\mathbb{1}\{\boldsymbol{\omega}(t) \in W_n^N\}e^{\alpha X}\right]e^{-\frac{\alpha u(n-1)}{N}}, \quad (7)
\end{aligned}$$

where the last inequality uses the definition of $W_n^N$ (see (5)). By applying Hölder's inequality with $p, q$, where $p \in [1, 1/\alpha)$ and $q = p/(p-1)$ on r.h.s. of (7), we deduce that $\log \mathbb{P}_{\boldsymbol{\mu}}\left[\hat{\boldsymbol{\mu}}(t) \in \mathcal{S}, \boldsymbol{\omega}(t) \in W_n^N\right]$ is at most

$$(\log \mathbb{E}_{\boldsymbol{\mu}}\left[e^{\alpha p X}\right])/p + (\log \mathbb{E}_{\boldsymbol{\mu}}[\mathbb{1}\{\boldsymbol{\omega}(t) \in W_n^N\}])/q - \frac{\alpha u(n-1)}{N}.$$

As $\alpha p \in (0, 1)$, Lemma 2 in Appendix B shows that the first term above is $o(t)$. Using definition of the rate function, (3) with $C = W_n^N$, on the second term yields that $\varliminf_{t \to \infty} \frac{1}{t} \log \frac{1}{\mathbb{P}_{\boldsymbol{\mu}}[\hat{\boldsymbol{\mu}}(t) \in \mathcal{S}, \boldsymbol{\omega}(t) \in W_n^N]}$ is lower bounded by

$$(1/q) \inf_{\boldsymbol{\omega} \in W_n^N} I(\boldsymbol{\omega}) + \frac{\alpha u(n-1)}{N} = (1 - 1/p) \inf_{\boldsymbol{\omega} \in W_n^N} I(\boldsymbol{\omega}) + \frac{\alpha u(n-1)}{N}.$$

As $p$ can be arbitrarily close to $1/\alpha$, we get the lower bound $(1 - \alpha)I(\boldsymbol{\omega}) + \frac{\alpha u(n-1)}{N}$. Further choosing $\alpha$ close to either 1 or 0, the proof is completed.

$\square$

### 3.4 Improved analysis of the Successive Rejects algorithm

In SR, the set of candidate arms is initialized as $\mathcal{C}_K = [K]$. The budget of samples is partitioned into $K - 1$ phases, and at the end of each phase, SR discards the empirical worst arm from the candidate set. In each phase, SR uniformly samples the arms in candidate set. The lengths of phases are set as follows. Define $\overline{\log}K := \frac{1}{2} + \sum_{k=2}^{K} \frac{1}{k}$. The candidate set is denoted by $\mathcal{C}_j$ when it has $j > 2$ arms. In the corresponding phase, (i) each arm in $\mathcal{C}_j$ is sampled until the round $t$ when $\min_{k \in \mathcal{C}_j} N_k(t)$ reaches $T/(j\overline{\log}K)$ (recall that $N_k(t)$ is the number of times arm $k$ has been sampled up to round $t$); (ii) the empirical worst arm, denoted by $\ell_j$, is then discarded, i.e., $\mathcal{C}_{j-1} = \mathcal{C}_j \setminus \{\ell_j\}$. During the last phase, the algorithm equally samples the two remaining arms and finally recommends $\hat{\imath}$, the arm with higher empirical mean in $\mathcal{C}_2$. The pseudo code is presented in Algorithm 1.

---

**Algorithm 1: SR**

**initialization** $\mathcal{C}_K \leftarrow [K], j \leftarrow K$;

**for** $(t = 1, \ldots, T)$ **do**

    **if** $(j > 2$ *and* $\min_{k \in \mathcal{C}_j} N_k(t) \geq \frac{T}{j\overline{\log}K})$ **then**

        $\ell_j \leftarrow \operatorname{argmin}_{k \in \mathcal{C}_j} \hat{\mu}_k(t)$ (tie broken arbitrarily), $\mathcal{C}_{j-1} \leftarrow \mathcal{C}_j \setminus \{\ell_j\}$, and $j \leftarrow j - 1$;

    **end**

    sample $A_t \leftarrow \operatorname{argmin}_{k \in \mathcal{C}_j} N_k(t)$ (tie broken arbitrarily), update $\{N_k(t)\}_{k \in \mathcal{C}_j}$ and $\hat{\boldsymbol{\mu}}(t)$;

**end**

$\ell_2 \leftarrow \arg\min_{k \in \mathcal{C}_2} \hat{\mu}_k(T)$ and return $\hat{\imath} \leftarrow \arg\max_{k \in \mathcal{C}_2} \hat{\mu}_k(T)$ (tie broken arbitrarily).

---

We apply the corollary (c) in Section 3.2 to improve the existing performance guarantees of SR. To simplify the presentation, we assume wlog that $\mu_1 > \mu_2 \geq \ldots \geq \mu_K$. For $j = 2, \ldots, K$, define

$$\Gamma_j = \min_{J \in \mathcal{J}} \inf \left\{ \sum_{k \in J} d(\lambda_k, \mu_k) : \boldsymbol{\lambda} \in \mathbb{R}^K, \lambda_1 \leq \min_{k \in J} \lambda_k \right\}, \quad (8)$$

where $\mathcal{J} = \{J \subseteq [K] : |J| = j, 1 \in J\}$.

**Theorem 2.** *Let $\boldsymbol{\mu} \in \Lambda$. Under* SR, *we have: for $j = 2, \ldots, K$, $\varliminf_{T\to\infty} \frac{1}{T} \log \frac{1}{\mathbb{P}_{\boldsymbol{\mu}}[\ell_j=1]} \geq \frac{\Gamma_j}{j\overline{\log}K}$.*
*Hence, the error probability of* SR *is upper bounded by $\varliminf_{T\to\infty} \frac{1}{T} \log \frac{1}{\mathbb{P}_{\boldsymbol{\mu}}[\hat{i}\neq 1]} \geq \min_{j\neq 1} \frac{\Gamma_j}{j\overline{\log}K}$.*

*Proof of Theorem 2.* Fix $j \in \{2, \ldots, K\}$. Observe that

$$\mathbb{P}_{\boldsymbol{\mu}}[\ell_j = 1] = \sum_{J\in\mathcal{J}} \mathbb{P}_{\boldsymbol{\mu}}[\ell_j = 1, \mathcal{C}_j = J] \leq |\mathcal{J}| \max_{J\in\mathcal{J}} \mathbb{P}_{\boldsymbol{\mu}}[\ell_j = 1, \mathcal{C}_j = J],$$

which implies that

$$\varliminf_{T\to\infty} \frac{1}{T} \log \frac{1}{\mathbb{P}_{\boldsymbol{\mu}}[\ell_j = 1]} \geq \min_{J\in\mathcal{J}} \varliminf_{T\to\infty} \frac{1}{T} \log \frac{1}{\mathbb{P}_{\boldsymbol{\mu}}[\ell_j = 1, \mathcal{C}_j = J]} \tag{9}$$

as $|\mathcal{J}| < \infty$.

Since $\ell_j$ is selected at the $\theta T$-th round[1] where $\theta = (1 + \sum_{k=j+1}^{K} \frac{1}{k})/\overline{\log}K$, the event $\{\ell_j = 1, \mathcal{C}_j = J\}$ implies that $\{\hat{\boldsymbol{\mu}}(\theta T) \in \mathcal{S}, \boldsymbol{\omega}(\theta T) \in W\}$, where

$$\mathcal{S} = \left\{\boldsymbol{\lambda} \in \mathbb{R}^K : \lambda_1 \leq \lambda_k, \forall k \in J\right\} \text{ and } W = \left\{\boldsymbol{\omega} \in \Sigma : \omega_k = \frac{1}{\theta j \overline{\log}K}, \forall k \in J\right\}.$$

In other words, $\mathbb{P}_{\boldsymbol{\mu}}[\ell_j = 1, \mathcal{C}_j = J] \leq \mathbb{P}_{\boldsymbol{\mu}}[\hat{\boldsymbol{\mu}}(\theta T) \in \mathcal{S}, \boldsymbol{\omega}(\theta T) \in W]$. Applying (c) in Section 3.2 with the above $\mathcal{S}$ and $W$ yields that $\varliminf_{T\to\infty} \frac{1}{\theta T} \log \frac{1}{\mathbb{P}_{\boldsymbol{\mu}}[\hat{\boldsymbol{\mu}}(\theta T)\in\mathcal{S}, \boldsymbol{\omega}(\theta T)\in W]}$ is larger than

$$\inf_{\boldsymbol{\omega}\in W} \inf_{\boldsymbol{\lambda}\in\text{cl}(\mathcal{S})} \Psi(\boldsymbol{\lambda}, \boldsymbol{\omega}) \geq \frac{1}{\theta j\overline{\log}K} \inf\left\{\sum_{k\in J} d(\lambda_k, \mu_k) : \lambda_1 \leq \min_{k\in J}\lambda_k\right\} \geq \frac{\Gamma_j}{\theta j\overline{\log}K}, \tag{10}$$

where the first inequality uses the fact that KL-divergences and the components of $\boldsymbol{\omega}$ are nonnegative, and the second one is due to the definition (8) of $\Gamma_j$. Combining (9) and (10) completes the proof. $\square$

The upper bound derived in Theorem 2 is tighter than those recently derived in (Barrier et al., 2022). Indeed, for any $J \in \mathcal{J}$, since $|J| = j$, one can find at least one index in $J$ at least larger than $j$, say $k_J$. Hence,

$$\Gamma_j \geq \min_{J\in\mathcal{J}} \inf_{\boldsymbol{\lambda}\in\mathbb{R}^K, \lambda_1\leq\lambda_{k_J}} d(\lambda_1, \mu_1) + d(\lambda_{k_J}, \mu_{k_J}) \geq \inf_{\boldsymbol{\lambda}\in\mathbb{R}^K, \lambda_1\leq\lambda_j} d(\lambda_1, \mu_1) + d(\lambda_j, \mu_j).$$

The r.h.s. in the previous inequality corresponds to the upper bounds derived by (Barrier et al., 2022).

To simplify the presentation and avoid rather intricate computations involving the KL-divergences, in the remaining of the paper, we restrict our attention to specific classes of reward distributions.

**Assumption 1.** *The rewards are bounded with values in $(0, 1)$. The reward distributions $\nu_1, \ldots, \nu_K$ are Bernoulli distributions such that $\nu_a$ is of mean $a$, and for any $a \neq b$, $d(a, b) \geq 2(a-b)^2$ (this is a consequence of Pinsker's inequality as rewards are in $(0, 1)$).*

Under Assumption 1, we have $\Gamma_j \geq 2\xi_j$ (a direct consequence of Proposition 3 in Appendix D.1), where for $j = 2, \ldots, K$,

$$\xi_j = \inf_{\boldsymbol{\lambda}\in[0,1]^j} \left\{\sum_{k=1}^{j} (\lambda_k - \mu_k)^2 : \lambda_1 \leq \min_{k=1,\ldots,j} \lambda_k\right\}.$$

We give an explicit expression of $\xi_j$ in Proposition 1, presented in Appendix D.1. Moreover, $2\xi_j$ is clearly larger than $2\inf_{\lambda_1\leq\lambda_j}\left\{(\lambda_1 - \mu_1)^2 + (\lambda_j - \mu_j)^2\right\} = (\mu_1 - \mu_j)^2$, and hence $\min_{j\neq 1} \Gamma_j/(j\overline{\log}K) \geq \min_{j\neq 1}(\mu_1 - \mu_j)^2/(j\overline{\log}K)$. This implies that our error probability upper bound is better than that derived in (Audibert et al., 2010).

**Example 1.** To illustrate the improvement brought by Theorem 2 on the performance guarantees of SR, consider the simple example with 3 Bernoulli arms and $\boldsymbol{\mu} = (0.9, 0.1, 0.1)$. Then $\min_{j\neq 1}(\mu_1 - \mu_j)^2/(j\overline{\log}3) = 0.16$ for the upper bound presented in (Audibert et al., 2010). From Proposition 1, instead we get $\min_{j\neq 1} 2\xi_j/(j\overline{\log}3) = 0.21$.

---

[1]To simplify the presentation, we ignore cases where $\theta T$ is not an integer. Refer to Appendix E for details.

# 4 Continuous Rejects Algorithms

In this section, we present CR, a truly adaptive algorithm that can discard an arm in *any* round. We propose two variants of the algorithm, CR-C using a conservative criterion to discard arms and CR-A discarding arms more aggressively. Using the Large Deviation results of Theorem 1, we establish error probability upper bounds for both CR-C and CR-A.

## 4.1 The CR-C and CR-A algorithms

As SR, CR initializes its candidate set as $\mathcal{C}_K = [K]$. For $j \geq 2$, $\mathcal{C}_j$ denotes the candidate set when it is reduced to $j$ arms. When $j > 2$, the algorithm samples arms in the candidate set $\mathcal{C}_j$ uniformly until a *discarding condition* is met. The algorithm then discards the empirically worst arm $\ell_j \in \mathcal{C}_j$, i.e., $\mathcal{C}_{j-1} \leftarrow \mathcal{C}_j \setminus \{\ell_j\}$. More precisely, in round $t$, if there are $j$ candidate arms remaining and if $\ell(t)$ denotes the empirically worst candidate arm, the discarding condition is $N_{\ell(t)}(t) > \max_{k \notin \mathcal{C}_j} N_k(t)$, $(\forall k \in \mathcal{C}_j, N_{\ell(t)}(t) = N_k(t))^2$, and

$$\text{for CR-C:} \quad \min_{k \in \mathcal{C}_j, k \neq \ell(t)} \hat{\mu}_k(t) - \hat{\mu}_{\ell(t)}(t) \geq G\left(\frac{\sum_{k \in \mathcal{C}_j} N_k(t)\overline{\log}j}{T - \sum_{k \notin \mathcal{C}_j} N_k(t)}\right), \tag{11}$$

$$\text{for CR-A:} \quad \frac{\sum_{k \in \mathcal{C}_j, k \neq \ell(t)} \hat{\mu}_k(t)}{j-1} - \hat{\mu}_{\ell(t)}(t) \geq G\left(\frac{\sum_{k \in \mathcal{C}_j} N_k(t)\overline{\log}j}{T - \sum_{k \notin \mathcal{C}_j} N_k(t)}\right), \tag{12}$$

where $G(\beta) = 1/\sqrt{\beta} - 1$ for all $\beta > 0$. The idea behind (11) is to keep the probability of discarding the best arm at most smaller than that of SR while using less budget. Note that (12) is easier to achieve than (11). CR-A is hence more aggressive than CR-C, and reduces the set of arms to $\mathcal{C}_2$ faster, but at the expense of a higher risk. After discarding $\ell_3$, CR will sample the arms in $\mathcal{C}_2$ evenly, and recommend the empirical best arm in $\mathcal{C}_2$ when the budget is exhausted. The pseudo-code of CR is presented in Algorithm 2.

---

**Algorithm 2:** CR-C and CR-A

---

**Input:** $\theta_0 \in (0, \frac{1}{\log K}) \cap \mathbb{Q}$ independent of $T$ (can be chosen as small as one wishes)

**initialization**

  |   $\mathcal{C}_K \leftarrow [K]$, $j \leftarrow K$, sample each arm $k \in [K]$ once, update $\{N_k(t)\}_{k \in \mathcal{C}_K}$ and $\hat{\boldsymbol{\mu}}(t)$;

**for** $t = K+1, \ldots, \lfloor \theta_0 T \rfloor$ **do**

  |   sample $A_t \leftarrow \operatorname{argmin}_{k \in \mathcal{C}_j} N_k(t)$ (tie broken arbitrarily), update $\{N_k(t)\}_{k \in \mathcal{C}_j}$ and $\hat{\boldsymbol{\mu}}(t)$;

**end**

**for** $(t = \lfloor \theta_0 T \rfloor + 1, \ldots, T)$ **do**

  |   $\ell(t) \leftarrow \operatorname{argmin}_{k \in \mathcal{C}_j} \hat{\mu}_k(t)$ (tie broken arbitrarily);

  |   **if** $j > 2$, $N_{\ell(t)}(t) > \max_{k \notin \mathcal{C}_j} N_k(t)$, $(\forall k \in \mathcal{C}_j, N_{\ell(t)}(t) = N_k(t))$,

  |   *and* (11) *(resp.* (12)*) holds for* CR-C *(resp.* CR-A*)* **then**

  |    |   $\ell_j \leftarrow \ell(t)$, $\mathcal{C}_{j-1} \leftarrow \mathcal{C}_j \setminus \{\ell_j\}$, $j \leftarrow j-1$

  |   sample $A_t \leftarrow \operatorname{argmin}_{k \in \mathcal{C}_j} N_k(t)$ (tie broken arbitrarily), update $\{N_k(t)\}_{k \in \mathcal{C}_j}$ and $\hat{\boldsymbol{\mu}}(t)$;

**end**

$\ell_2 \leftarrow \ell(T)$; return $\hat{\imath} \leftarrow \operatorname{argmax}_{k \in \mathcal{C}_2} \hat{\mu}_k(T)$ (tie broken arbitrarily).

---

## 4.2 Analysis of CR-C and CR-A

As in Section 3.4, $\mu_1 > \mu_2 \geq \ldots \geq \mu_K$ is assumed wlog and we further define $\mu_{K+1} = 0$. We introduce the following instance-specific quantities needed to state our error probability upper bounds. For $j \in \{2, \ldots, K\}$, define

$$\psi_j = \frac{j-1}{j}\left(\mu_1 - \frac{\sum_{k=2}^{j} \mu_k}{j-1}\right)^2, \quad \bar{\psi}_j = \frac{j-1}{j}\left(\mu_1 - \frac{\sum_{k=2}^{j-1} \mu_k + \mu_{j+1}}{j-1}\right)^2, \quad \zeta_j = \mu_j - \mu_{j+1},$$

---

[2]This condition is only imposed to simplify our analysis. It may be removed.

$$\varphi_j = \frac{\sum_{k=1}^{j} \mu_k}{j} - \mu_{j+1}, \quad \text{and} \quad \bar{\xi}_j = \inf\left\{ \sum_{k=1}^{K} (\lambda_k - \mu_k)^2 : \boldsymbol{\lambda} \in [0,1]^K, \lambda_1 \leq \min_{k=2,\ldots,j-1,j+1} \lambda_k \right\}.$$

Here we remark $\bar{\xi}_j \geq \xi_j$ and $\bar{\psi}_j \geq \psi_j$. These inequalities are proven in Proposition 3 and Proposition 5 in Appendix D.

**Theorem 3.** *Let $\boldsymbol{\mu} \in [0,1]^K$. Under* CR-C, $\varliminf_{T\to\infty} \frac{1}{T} \log \frac{1}{\mathbb{P}_{\boldsymbol{\mu}}[\hat{\imath} \neq 1]}$ *is larger than*

$$2 \min_{j=2,\ldots,K} \left\{ \frac{\min\left\{ \max\left\{ \frac{\xi_j \overline{\log}(j+1)(1-\alpha_j)\mathbb{1}_{\{j \neq K\}}}{\overline{\log} j}, \xi_j \right\}, \bar{\xi}_j \right\}}{j \overline{\log} K} \right\},$$

*where $\alpha_j \in \mathbb{R}$ is the real number such that $\frac{2\xi_j(1-\alpha_j)}{j \overline{\log} j} = [((1+\zeta_j)\sqrt{\alpha_j} - \sqrt{\frac{1}{(j+1)\overline{\log}(j+1)}})_+]^2$.*

**Theorem 4.** *Let $\boldsymbol{\mu} \in [0,1]^K$. Under* CR-A, $\varliminf_{T\to\infty} \frac{1}{T} \log \frac{1}{\mathbb{P}_{\boldsymbol{\mu}}[\hat{\imath} \neq 1]}$ *is larger than*

$$2 \min_{j=2,\ldots,K} \left\{ \frac{\min\{\max\{\frac{\psi_j \overline{\log}(j+1)(1-\alpha_j)\mathbb{1}_{\{j \neq K\}}}{\overline{\log} j}, \psi_j\}, \bar{\psi}_j\}}{j \overline{\log} K} \right\},$$

*where $\alpha_j \in \mathbb{R}$ is the real number such that $\frac{\psi_j(1-\alpha_j)}{j \overline{\log} j} = \frac{j}{j+1}[((1+\varphi_j)\sqrt{\alpha_j} - \sqrt{\frac{1}{(j+1)\overline{\log}(j+1)}})_+]^2$.*

Note that Theorem 3 implies that CR-C enjoys better performance guarantees than SR, and hence has for now the best known error probability upper bounds.

*Proof sketch.* The complete proof of Theorems 3 and 4 are given in Appendices C.1 and C.2. We sketch that of Theorem 3. The proof consists in upper bounding $\mathbb{P}_{\boldsymbol{\mu}}[\ell_j = 1]$ for $j \in \{2, \ldots, K\}$. We focus here on the most challenging case where $j \in \{3, \ldots, K-1\}$ (the analysis is simpler when $j = K$, since the only possible allocation is uniform, and when $j = 2$, since the only possible round deciding $\ell_2$ is the last round).

To upper bound $\mathbb{P}_{\boldsymbol{\mu}}[\ell_j = 1]$ using Theorem 1, we will show that it is enough to study the large deviations of the process $\{\boldsymbol{\omega}(\theta T)\}_{T \geq 1}$ for any fixed $\theta \in [\theta_0, 1]$ and to define a set $\mathcal{S} \subseteq [0,1]^K$ under which $\ell_j = \ell(\theta T) = 1$. We first observe that $\ell_j = \ell(\theta T)$ restricts the possible values of $\boldsymbol{\omega}(\theta T)$: $\boldsymbol{\omega}(\theta T) \in \mathcal{X}_j := \{\boldsymbol{x} \in \Sigma : \exists \sigma \in [K]^2 \text{ s.t. } x_{\sigma(1)} = \ldots = x_{\sigma(j)} > x_{\sigma(j+1)} > \ldots > x_{\sigma(K)} > 0\}$. We can hence just derive the LDP satisfied by $\{\boldsymbol{\omega}(\theta T)\}_{T \geq 1}$ on $\mathcal{X}_j$. This is done in Appendix E, and we identify by $I_\theta$ a rate function leading to an LDP upper bound. By defining

$$\mathcal{X}_{j,i}(\theta) = \left\{ \boldsymbol{x} \in \mathcal{X}_j : \theta x_{\sigma(i)} i \overline{\log} i > 1 - \theta \sum_{k=i+1}^{K} x_{\sigma(k)} \right\}, \forall i \in \{j, \ldots, K\},$$

As it is shown in Appendix E.1 that $I_\theta(\boldsymbol{x}) = \infty$ if $\boldsymbol{x} \in \mathcal{X}_{j,i}(\theta)$ for $i \geq j$, we may further restrict to $\mathcal{X}_j \setminus \cup_{i=j}^{K} \mathcal{X}_{j,i}(\theta)$.

Next, we explain how to apply Theorem 1 to upper bound $\mathbb{P}_{\boldsymbol{\mu}}[\ell_j = 1]$. Let $\mathcal{J} = \{J \subseteq [K] : |J| = j, 1 \in J\}$ as defined in Section 3.4. For all $\beta, \theta \in (0,1]$ and $J \in \mathcal{J}$, we introduce the sets

$$\mathcal{S}_J(\beta) = \left\{ \boldsymbol{\lambda} \in [0,1]^K : \min_{k \in J, k \neq 1} \lambda_k - \lambda_1 \geq G(\beta) \right\},$$

$$\mathcal{Z}_J(\theta, \beta) = \left\{ \boldsymbol{z} \in \mathcal{X}_j \setminus \cup_{i=j}^{K} \mathcal{X}_{j,i}(\theta) : (\forall k \in J, \ z_k = \max_{k' \in [K]} z_{k'}), \ \frac{\theta \sum_{k \in J} z_k \overline{\log} j}{1 - \theta \sum_{k \notin J} z_k} = \beta \right\}.$$

Assume that in round $t$, $\ell_j = \ell(t) = 1, \mathcal{C}_j = J$ and let $\tau = \sum_{k \notin J} N_k(t) \leq t$ be the number of times arms outside $J$ are pulled. While $\boldsymbol{\omega}(t) \notin \mathcal{X}_{j,j}(t/T)$, we have $\beta = \frac{(t-\tau)\overline{\log} j}{T-\tau} \in (0,1]$. Using the criteria (11), we observe that

$$\sum_{t=K+1}^{T} \sum_{J \in \mathcal{J}} \mathbb{P}_{\boldsymbol{\mu}}\left[ \ell_j = \ell(t) = 1, \mathcal{C}_j = J, \boldsymbol{\omega}(t) \in \mathcal{X}_j \setminus \cup_{i=j}^{K} \mathcal{X}_{j,i}(\frac{t}{T}) \right]$$

$$\leq \sum_{t=K+1}^{T} \sum_{J \in \mathcal{J}} \sum_{\tau \leq t, \tau \in \mathbb{N}} \mathbb{P}_{\boldsymbol{\mu}}\left[ \hat{\boldsymbol{\mu}}(t) \in \mathcal{S}_J(\frac{(t-\tau)\overline{\log} j}{T-\tau}), \boldsymbol{\omega}(t) \in \mathcal{Z}_J(\frac{t}{T}, \frac{(t-\tau)\overline{\log} j}{T-\tau}) \right].$$

To upper bound the r.h.s. in the above inequality, we combine the results of Theorem 1 and a partitioning technique (presented in Appendix G). This gives:

$$\lim_{T\to\infty} \frac{1}{T} \log \frac{1}{\mathbb{P}_{\boldsymbol{\mu}}\left[\hat{\boldsymbol{\mu}}(\theta T) \in \mathcal{S}_J(\beta), \boldsymbol{\omega}(\theta T) \in \mathcal{Z}_J(\theta, \beta)\right]} \geq \theta \inf_{\boldsymbol{z} \in \mathrm{cl}(\mathcal{Z}_J(\theta,\beta))} \max\{F_{\mathcal{S}_J(\beta)}(\boldsymbol{z}), I_\theta(\boldsymbol{z})\}.$$

The proof is completed by providing lower bounds of $\theta F_{\mathcal{S}_J(\beta)}(\boldsymbol{z})$ and $\theta I_\theta(\boldsymbol{z})$ for a fixed $\boldsymbol{z} \in \mathcal{Z}_J(\theta, \beta)$ with various $J$. Such bounds are derived in Appendix D.1 and E.3.1, respectively. □

**Example 2.** To conclude this section, we just illustrate through a simple example the gain in terms of performance guarantees brought by CR compared to SR. Assume we have 50 Bernoulli arms with $\mu_1 = 0.95, \mu_2 = 0.85, \mu_3 = 0.2$, and $\mu_k = 0$ for $k = 4, \ldots, 50$. For SR, Theorem 2 states that with a budget of 5000 samples, the error probability of SR does not exceed $1.93 \times 10^{-3}$. With the same budget, Theorems 3 and 4 state that the error probabilities of CR-C and CR-A do not exceed $6.40 \times 10^{-4}$ and $6.36 \times 10^{-4}$, respectively.

We note that in general, we cannot say that one of our two algorithms, CR-C or CR-A, has better guarantees than the other. This is demonstrated in the problem instances presented in Appendix K.1 and K.4.

## 5   Numerical Experiments

We consider various problem instances to numerically evaluate the performance of CR. In these instances, we vary the number of arms from 5 to 55; we use Bernoulli distributed rewards, and vary the shape of the arm-to-reward mapping. For each instance, we compare CR to SR, SH, and UGapE.

Most of our numerical experiments are presented in Appendix K. Due to space constraints, we just provide an example of these results below. In this example, we have 55 arms with *convex* arm-to-reward mapping. The mapping has 10 steps, and the $m$-th step consists of $m$ arms with same average reward, equal to $\frac{3}{4} \cdot 3^{-\frac{m}{10}}$. Table 1 presents the error probabilities averaged over $40,000$ independent runs. Observe that CR-A performs better than CR-C (being aggressive when discarding arms has some benefits), and both versions of CR perform better than SR and all other algorithms.

Table 1: Error probability (in %).

|  |  | $T = 3,000$ | $T = 4,000$ | $T = 5,000$ |
|---|---|---|---|---|
| UGapE | (Gabillon et al., 2012) | 24.7 | 21.3 | 18.9 |
| SH | Karnin et al. (2013) | 10.2 | 5.9 | 3.2 |
| SR | Audibert et al. (2010) | 5.5 | 2.8 | 1.3 |
| CR-C | (this paper) | 7.1 | 2.6 | 1.1 |
| CR-A | (this paper) | **4.7** | **1.6** | **0.6** |

## 6   Conclusion

In this paper, we have established, in MAB problems, a connection between the LDP satisfied by the sampling process (under any adaptive algorithm) and that satisfied by the empirical average rewards of the various arms. This connection has allowed us to improve the performance analysis of existing best arm identification algorithms, and to devise and analyze new algorithms with an increased level of adaptiveness. We show that one of these algorithms CR-C has better performance guarantees than existing algorithms and that it performs also better in practice in most cases.

Future research directions include: (i) developing algorithms with further improved performance guarantees – can the discarding conditions of CR be further optimized? (ii) Enhancing the Large Deviation analysis of adaptive algorithms – under which conditions, can we establish a complete LDP of the process $\{Z(t)\}_{t\geq 1} = \{(\boldsymbol{\omega}(t), \hat{\boldsymbol{\mu}}(t))\}_{t\geq 1}$? Answering this question would constitute a strong step towards characterizing the minimal instance-specific error probability for best arm identification with fixed budget. (iii) Extending our approach to other pure exploration tasks: top-$m$ arm identification problems (Bubeck et al., 2013), best arm identification in structured bandits (Yang and Tan, 2022; Azizi et al., 2022), or best policy identification in reinforcement learning.

## Acknowledgements

The authors would like to express their gratitude to Guo-Jhen Wu for his invaluable discussion during the initial stages of this project. R.-C Tzeng's research is supported by the ERC Advanced Grant REBOUND (834862), A. Proutiere is supported by the Wallenberg AI, Autonomous Systems and Software Program (WASP) funded by the Knut and Alice Wallenberg Foundation, and Digital Futures.

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
