# Contents

# A   Notation

| Problem setting | |
|---|---|
| $K$ | Number of arms |
| $[m]$ for any $m \in \mathbb{N}$ | The set $\{1, 2 \ldots, m\}$ |
| $\nu_k$ | Reward distribution for arm $k$ |
| $X_k(t)$ | Random reward received from pulling arm $k$ in round $t$ |
| $\boldsymbol{\mu} \in \mathbb{R}^K$ | Vector of the expected rewards of the various arms |
| $\Lambda$ | Set of all possible parameters $\boldsymbol{\mu}$ |
| $1(\boldsymbol{\mu})$ | Best arm under parameter $\boldsymbol{\mu}$ |
| $T$ | Given budget |

| Quantities related to the error rate lower bound | |
|---|---|
| $\boldsymbol{\omega}$ | Vector of the proportions of arm draws |
| $\Sigma$ | Simplex |
| $\mathbb{E}_{\boldsymbol{\mu}}$ and $\mathbb{P}_{\boldsymbol{\mu}}$ | The expectation and probability measure corresponding to $\boldsymbol{\mu}$ |
| $\mathrm{Alt}(\boldsymbol{\mu})$ | Set of confusing parameters for $\boldsymbol{\mu}$ (whose best arm is not $1(\boldsymbol{\mu})$) |
| $d(\mu, \mu')$ | KL divergence between the distributions parametrized by $\mu$ and $\mu'$ |
| $\mathrm{kl}(a, b)$ | KL divergence between two Bernoulli distributions of means $a$ and $b$ |
| $\Psi(\boldsymbol{\lambda}, \boldsymbol{\omega})$ | $\sum_{k=1}^{K} \omega_k d(\lambda_k, \mu_k)$ |

| Notation used in large deviation theory | |
|---|---|
| $\mathrm{cl}(\mathcal{S})$ | The closure of $\mathcal{S}$ |
| $F_{\mathcal{S}}(\boldsymbol{\omega})$ | $\inf_{\boldsymbol{\lambda} \in \mathrm{cl}(\mathcal{S})} \Psi(\boldsymbol{\lambda}, \boldsymbol{\omega})$ |
| $I$ | Rate function for $\{\boldsymbol{\omega}(t)\}_{t \geq 1}$ |
| $B(y, \delta)$ | The open ball with center $y$ and radius $\delta$ |

| Notation used in the algorithms | |
|---|---|
| $N_k(t)$ | Number of pulls of arm $k$ up to $t$ |
| $\omega_k(t)$ | $N_k(t)/t$ |
| $A_t$ | The arm pulled in round $t$ |
| $\hat{\mu}_k(t)$ | $\sum_{s=1}^{t} X_k(s) \mathbb{1}\{A_s = k\}/N_k(t)$ |
| $\hat{\imath}$ | Recommended arm |

| Notation for SR, CR (assuming $\mu_1 > \mu_2 \geq \ldots \geq \mu_K$) | |
|---|---|
| $\mathcal{C}_j$ | Candidates set with size $j$ |
| $\ell_j$ | The arm discarded from $\mathcal{C}_j$ |
| $\ell(t)$ | Empirical worst arm at round $t$ |
| $\overline{\log} j$ | $\frac{1}{2} + \sum_{k=2}^{j} \frac{1}{k}$ |
| $G(\beta)$ | $\frac{1}{\sqrt{\beta}} - 1$ |
| $\mathcal{J}$ | $\{J \subseteq [K] : \lvert J \rvert = j, 1 \in J\}$ |
| $I_\theta$ | Rate function for $\{\boldsymbol{\omega}(\theta T)\}_{T \geq 1}$ |
| $\Gamma_j$ | $\min_{J \in \mathcal{J}} \inf \left\{ \sum_{k \in J} d(\lambda_k, \mu_k) : \boldsymbol{\lambda} \in \mathbb{R}^K, \lambda_1 \leq \min_{k \in J} \lambda_k \right\}$ |
| $\xi_j$ | $\inf_{\boldsymbol{\lambda} \in [0,1]^j} \left\{ \sum_{k=1}^{j} (\lambda_k - \mu_k)^2 : \lambda_1 \leq \min_{k=1,\ldots,j} \lambda_k \right\}$ |
| $\bar{\xi}_j$ | $\inf_{\boldsymbol{\lambda} \in [0,1]^K} \left\{ \sum_{k=1}^{K} (\lambda_k - \mu_k)^2 : \lambda_1 \leq \min_{k=2,\ldots,j-1,j+1} \lambda_k \right\}$ |
| $\psi_j$ | $\frac{j-1}{j} \left( \mu_1 - \frac{\sum_{k=2}^{j} \mu_k}{j-1} \right)^2$ |
| $\bar{\psi}_j$ | $\frac{j-1}{j} \left( \mu_1 - \frac{\sum_{k=2}^{j-1} \mu_k + \mu_{j+1}}{j-1} \right)^2$ |
| $\zeta_j$ | $\mu_j - \mu_{j+1}$ |
| $\varphi_j$ | $\sum_{k=1}^{j} \mu_k / j - \mu_{j+1}$ |

## B Technical lemmas towards the proof of Theorem 1

**Lemma 2.** *Let $\boldsymbol{\mu} \in \Lambda$, $t > \max\{K, e\}$ and $\beta \in (0, 1)$. There is a constant $c > 0$ (that depends on $K$ and $\beta$ only) s.t.*

$$\mathbb{E}_{\boldsymbol{\mu}}\left[e^{\beta X}\right] \leq c(\log t)^K,$$

*where $X = \sum_{k=1}^{K} N_k(t) d(\hat{\mu}_k(t), \mu_k)$.*

*Proof.* Let $M$ be the smallest positive integer s.t. (i) $\frac{\log M}{\beta} > K + 1$ and (ii) $(\log M)^{2K} < M^{\frac{1}{\beta}}$. We have:

$$
\begin{aligned}
\mathbb{E}_{\boldsymbol{\mu}}\left[e^{\beta X}\right] &\leq \sum_{n=0}^{\infty} \mathbb{P}_{\boldsymbol{\mu}}\left[e^{\beta X} \geq n\right] \\
&\leq M + \sum_{n \geq M} \mathbb{P}_{\boldsymbol{\mu}}\left[X \geq \frac{\log n}{\beta}\right] \\
&\leq M + \sum_{n \geq M} \frac{\left(2(\log n)^2 \log t\right)^K}{n^{\frac{1}{\beta}}} \frac{e^{K+1}}{\beta^{2K} K^K},
\end{aligned}
\tag{13}
$$

where the last inequality follows from repeatedly invoking Lemma 3 with $\delta = \frac{\log n}{\beta}$ (notice that $n \geq M$ satisfies the condition on $\delta$ of Lemma 3). Observe that the r.h.s. of (13) is a Bertrand series and it is convergent since $\beta < 1$. Unfamiliar reader can check the convergence analysis below. (ii) implies that the sequence will decrease after $n \geq M$, hence the sum in (13) is bounded as (up to a constant multiplicative factor):

$$
\begin{aligned}
\sum_{n \geq M} \frac{(\log n)^{2K}}{n^{\frac{1}{\beta}}} &\leq \int_{1}^{\infty} \frac{(\log x)^{2K}}{x^{\frac{1}{\beta}}} dx \\
&= \int_{0}^{\infty} y^{2K} e^{-\left(\frac{1}{\beta}-1\right)y} dy \\
&\leq \left(\frac{1}{\beta} - 1\right)^{-2K-1} \Gamma(2K + 1).
\end{aligned}
\tag{14}
$$

The constant $c$ can be deduced from (13) and (14). $\qquad \square$

The following lemma is Theorem 2 in Magureanu et al. (2014). It was originally stated for Bernoulli distributions, but as claimed in Garivier and Kaufmann (2016); Kaufmann and Koolen (2018), it is straightforward to generalize it to one-parameter exponential distributions.

**Lemma 3** (Magureanu et al. (2014))**.** *For all $\delta > (K + 1)$ and $t \in \mathbb{N}$, we have:*

$$\mathbb{P}_{\boldsymbol{\mu}}[X \geq \delta] \leq e^{-\delta} \left(\frac{\lceil \delta \log t \rceil \delta}{K}\right)^K e^{K+1}.$$

## C  Analysis of CR

In C.1, we give the proof for Theorem 3 and in C.2 that of Theorem 4. As mentioned in §3.4 and §4.2, in the following analysis, we will assume that $1 \geq \mu_1 > \mu_2 \geq \ldots \geq \mu_K \geq 0$.

### C.1  Performance analysis of CR-C

**Theorem 3.** *Let $\boldsymbol{\mu} \in [0,1]^K$. Under CR-C, $\underline{\lim}_{T\to\infty} \frac{1}{T}\log\frac{1}{\mathbb{P}_{\boldsymbol{\mu}}[i\neq 1]}$ is larger than*

$$2 \min_{j=2,\ldots,K} \left\{ \frac{\min\left\{ \max\left\{ \frac{\xi_j \overline{\log}(j+1)(1-\alpha_j)\mathbb{1}_{\{j\neq K\}}}{\overline{\log} j}, \xi_j \right\}, \bar{\xi}_j \right\}}{j\overline{\log}K} \right\},$$

*where $\alpha_j \in \mathbb{R}$ is the real number such that*

$$\frac{2\xi_j(1-\alpha_j)}{j\overline{\log}j} = \left[\left((1+\zeta_j)\sqrt{\alpha_j} - \sqrt{\frac{1}{(j+1)\overline{\log}(j+1)}}\right)_+\right]^2.$$

We upper bound $\mathbb{P}_{\boldsymbol{\mu}}[\ell_j = 1]$ for (i) $j = K$; (ii) $j = 2$; (iii) $j \in \{3,\ldots,K\}$. The upper bound for (i), presented in C.1.1, is the easiest to derive as the only possible allocation before one discards the first arm is uniform among all arms. The bound for (ii), presented in C.1.2, is the second easiest to derive as $\ell_2$ is decided only in the end, namely, in the $T$-th round. The upper bound for (iii), presented in C.1.3, is more involved since we have to consider all possible allocations and rounds.

#### C.1.1  Upper bound of $\mathbb{P}_{\boldsymbol{\mu}}[\ell_K = 1]$

**Lemma 4.** *Let $\boldsymbol{\mu} \in [0,1]^K$. Under CR-C,*

$$\lim_{T\to\infty} \frac{1}{T}\log\frac{1}{\mathbb{P}_{\boldsymbol{\mu}}[\ell_K = 1]} \geq \frac{2\xi_K}{K\overline{\log}K}.$$

*Proof.* Without loss of generality, let us assume $\theta_0 T > K$. Observe that

$$\mathbb{P}_{\boldsymbol{\mu}}[\ell_K = 1] = \sum_{t\geq\theta_0 T}^{T} \mathbb{P}_{\boldsymbol{\mu}}[\ell_K = \ell(t) = 1]. \tag{15}$$

Since CR-C discards $\ell_K = \ell(t)$ at the round $t$ only when $N_{\ell(t)}(t) = N_k(t)$ for all $k \in [K]$, it suffices to consider $\boldsymbol{\omega}(t) \in \mathcal{X}_K = \{(1/K,\ldots,1/K)\}$. We further introduce

$$\mathcal{S}_\theta = \left\{ \boldsymbol{\lambda} \in \mathbb{R}^K : \lambda_1 \leq \min_{k\neq 1}\lambda_k - G(\theta\overline{\log}K) \right\}, \forall\theta \in [\theta_0, 1] \cap \mathbb{Q}.$$

With this notation, we can use the criteria of discarding the arm $\ell_K = \ell(t) = 1$ (see (11)) to get that:

$$\sum_{t\geq\theta_0 T}^{T} \mathbb{P}_{\boldsymbol{\mu}}[\ell_K = \ell(t) = 1] \leq \sum_{t\geq\theta_0 T}^{T} \mathbb{P}_{\boldsymbol{\mu}}\left[ \hat{\boldsymbol{\mu}}(t) \in \mathcal{S}_{\frac{t}{T}}, \boldsymbol{\omega}(t) \in \mathcal{X}_K \right]. \tag{16}$$

Applying Theorem 10 in Appendix G with $\tilde{\theta}_0 = \theta_0$

$$\mathcal{E} = \{\ell_K = 1\}, \mathcal{S}_{\theta,\gamma} = \mathcal{S}_\theta, W_{\theta,\gamma} = \mathcal{X}_K, \forall\gamma,$$

yields that

$$\lim_{T\to\infty} \frac{1}{T}\log\frac{1}{\mathbb{P}_{\boldsymbol{\mu}}[\ell_K = 1]} \geq \inf_{\theta,\gamma\in[\theta_0,1]\cap\mathbb{Q}} \inf_{\boldsymbol{\omega}\in\mathrm{cl}(W_{\theta,\gamma})} \theta\max\{F_{\mathcal{S}_{\theta,\gamma}}(\boldsymbol{\omega}), I_\theta(\boldsymbol{\omega})\}$$

$$= \inf_{\theta\in[\theta_0,1]\cap\mathbb{Q}} \theta\{F_{\mathcal{S}_\theta}(1/K,\ldots,1/K), I_\theta(1/K,\ldots,1/K)\}. \tag{17}$$

Theorem 10 can be indeed applied since, in view of Theorem 5, $\{\boldsymbol{\omega}(\theta T)\}_{T\geq 1}$ satisfies LDP upper bound (3) with rate function $I_\theta$. In the above derivation, by convention, we let $\inf_{\boldsymbol{\lambda}\in\emptyset} f(\boldsymbol{\lambda}) =$

$\infty$. Next, from Theorem 5 (a) in Appendix E.1, we know that $I_\theta(1/K, \ldots, 1/K) = \infty$ if $\theta > 1/\overline{\log}K$. Thus, the minimization problem on r.h.s. of (17) can be further lower bounded by $\inf_{\theta \in [\theta_0, 1/\overline{\log}K] \cap \mathbb{Q}} \theta F_{\mathcal{S}_\theta}(1/K, \ldots, 1/K)$. From the definition of $F_{\mathcal{S}_\theta}$, we have

$$
\begin{aligned}
\theta F_{\mathcal{S}_\theta}(1/K, \ldots, 1/K) &= \frac{\theta}{K} \inf \left\{ \sum_{k=1}^K d(\lambda_k, \mu_k) : \lambda_1 \leq \min_{k \neq 1} \lambda_k - G(\theta \overline{\log} K) \right\} \\
&\geq \frac{2\theta}{K} \inf \left\{ \sum_{k=1}^K (\lambda_k - \mu_k)^2 : \lambda_1 \leq \min_{k \neq 1} \lambda_k - G(\theta \overline{\log} K) \right\} \geq \frac{2\xi_K}{K \overline{\log} K},
\end{aligned}
\tag{18}
$$

where the first inequality is from Assumption 1, and the last inequality follows from Proposition 4 in Appendix D.1 with $\beta = \theta \overline{\log} K$. The proof is completed combining (17) and (18).

$\square$

### C.1.2 Upper bound of $\mathbb{P}_{\boldsymbol{\mu}}[\ell_2 = 1]$

**Lemma 5.** *Let* $\boldsymbol{\mu} \in [0, 1]^K$. *Under* CR–C,

$$
\lim_{T \to \infty} \frac{1}{T} \log \frac{1}{\mathbb{P}_{\boldsymbol{\mu}}[\ell_2 = 1]} \geq \frac{\min\{\max\{4\xi_2(1 - \alpha_2), 3\xi_2\}, 3\bar{\xi}_2\}}{3\overline{\log}K},
$$

*where* $\alpha_2 \in \mathbb{R}$ *is the real number such that*

$$
\xi_2(1 - \alpha_2) = \left[ \left( (1 + \zeta_2)\sqrt{\alpha_2} - \frac{1}{2} \right)_+ \right]^2.
\tag{19}
$$

*Proof.* There are two arms remaining in the last phase. Hence, it suffices to consider the estimate and allocation in the last round. The possible allocation $\boldsymbol{\omega}(T)$ belongs to the set

$$
\mathcal{X}_2 = \left\{ \boldsymbol{x} \in \Sigma : \exists \sigma : [K] \mapsto [K] \text{ s.t. } x_{\sigma(1)} = x_{\sigma(2)} > x_{\sigma(3)} > \ldots > x_{\sigma(K)} > 0 \right\}.
$$

As we defined $\mathcal{J}$ in Section 3.4, we introduce $\mathcal{D} = \{D \subseteq [K] : |D| = 2, 1 \in D\}$, and

$$
\mathcal{X}_D = \left\{ \boldsymbol{x} \in \mathcal{X}_2 : \min_{k \in D} x_k > \max_{k' \notin D} x_{k'} \right\}.
$$

The set $\cup_{D \in \mathcal{D}} \mathcal{X}_D$ is a subset of $\mathcal{X}_2$, and is relevant when we consider events where the best arm 1 is discarded in the last elimination phase. Since $\ell_2$ is decided as the empirical worst arm in $\mathcal{C}_2$,

$$
\begin{aligned}
\mathbb{P}_{\boldsymbol{\mu}}[\ell_2 = 1] &\leq \sum_{D \in \mathcal{D}} \mathbb{P}_{\boldsymbol{\mu}}[\hat{\boldsymbol{\mu}}(T) \in \mathcal{S}_D, \boldsymbol{\omega}(T) \in \mathcal{X}_D] \\
&\leq (K - 1) \max_{D \in \mathcal{D}} \mathbb{P}_{\boldsymbol{\mu}}[\hat{\boldsymbol{\mu}}(T) \in \mathcal{S}_D, \boldsymbol{\omega}(T) \in \mathcal{X}_D],
\end{aligned}
\tag{20}
$$

where

$$
\mathcal{S}_D = \left\{ \boldsymbol{\lambda} \in [0, 1]^K : \min_{k \in [D]} \lambda_k \geq \lambda_1 \right\}.
$$

Rearranging (20) yields that

$$
\begin{aligned}
\lim_{T \to \infty} \frac{1}{T} \log \frac{1}{\mathbb{P}_{\boldsymbol{\mu}}[\ell_2 = 1]} &\geq \lim_{T \to \infty} \min_{D \in \mathcal{D}} \frac{1}{T} \log \frac{1}{\mathbb{P}_{\boldsymbol{\mu}}[\hat{\boldsymbol{\mu}}(T) \in \mathcal{S}_D, \boldsymbol{\omega}(T) \in \mathcal{X}_D]} \\
&\geq \min_{D \in \mathcal{D}} \lim_{T \to \infty} \frac{1}{T} \log \frac{1}{\mathbb{P}_{\boldsymbol{\mu}}[\hat{\boldsymbol{\mu}}(T) \in \mathcal{S}_D, \boldsymbol{\omega}(T) \in \mathcal{X}_D]} \\
&\geq \min_{D \in \mathcal{D}} \inf_{\boldsymbol{\omega} \in \mathrm{cl}(\mathcal{X}_D)} \max\{F_{\mathcal{S}_D}(\boldsymbol{\omega}), I_1(\boldsymbol{\omega})\},
\end{aligned}
\tag{21}
$$

where $I_1$ denotes the rate function for which $\{\boldsymbol{\omega}(T)\}_{T \geq 1}$ satisfies an LDP upper bound (3), and the last inequality follows from Theorem 1 with $\mathcal{S} = \mathcal{S}_D, \overline{W} = \mathcal{X}_D$.

Further introduce for all $i \in \{2, \dots, K\}$,

$$\mathcal{X}_{2,i}(1) = \left\{ \boldsymbol{x} \in \mathcal{X}_2 : x_{\sigma(i)} i \overline{\log} i > 1 - \sum_{k=i+1}^{K} x_{\sigma(k)} \right\},$$

where in this definition, $\sigma$ refers to the permutation used in the definition of $\mathcal{X}_2$. In Theorem 5 (a) in Appendix E.1, we show that $I_1(\boldsymbol{\omega}) = \infty$ for all $\boldsymbol{\omega} \in \cup_{i=2}^{K} \mathcal{X}_{2,i}(1)$. Hence any $\boldsymbol{\omega} \in \cup_{i=2}^{K} \mathcal{X}_{2,i}(1)$ cannot be the minimizer on the r.h.s. of (21). In the following, we define

$$\mathcal{Z}_D(1,1) = \mathcal{X}_D \setminus \cup_{i=2}^{K} \mathcal{X}_{2,i}(1).$$

Here the argument $(1,1)$ in $\mathcal{Z}_D(1,1)$ is to be consistent with our notation in Appendix E, but we abbreviate it as $\mathcal{Z}_D$ for short below. To get a lower bound of (21), we consider two cases: (a) $D \neq [2]$; (b) $D = [2]$.

**(a) The case where $D \neq [2]$.** Using Corollary 1 with $\beta = 1, j = 2$ in Appendix D.1 yields that:

$$\inf_{\boldsymbol{\omega} \in \mathrm{cl}(\mathcal{X}_D)} \max\{F_{\mathcal{S}_D}(\boldsymbol{\omega}), I_1(\boldsymbol{\omega})\} \geq \inf_{z \in \mathrm{cl}(\mathcal{Z}_D)} F_{\mathcal{S}_D}(z)$$

$$\geq \left(1 - \sum_{k \notin D} z_k\right) \inf \left\{ \sum_{k \in D} (\lambda_k - \mu_k)^2 : \boldsymbol{\lambda} \in [0,1]^K, \min_{k \in D} \lambda_k = \lambda_1 \right\} \tag{22}$$

$$\geq \left(1 - \sum_{k \notin D} z_k\right) \bar{\xi}_2$$

$$\geq \frac{\bar{\xi}_2}{\log K}, \tag{23}$$

where the last inequality directly comes from Proposition 7 with $\theta = 1$, $j = 2$, $i = 3$ in Appendix E.2.

**(b) The case where $D = [2]$.** Next, we will show that both $\frac{\xi_2}{\log K}$ and $\frac{4\xi_2(1-\alpha_2)}{3\log K}$ are lower bounds for $\inf_{\boldsymbol{\omega} \in \mathrm{cl}(\mathcal{X}_D)} \max\{F_{\mathcal{S}_D}(\boldsymbol{\omega}), I_1(\boldsymbol{\omega})\}$. The maximum of these hence becomes our lower bound. Together with the conclusion obtained in the case (a), we complete the proof of Lemma 5.

Lower bounding by $\frac{\xi_2}{\log K}$. Observe that (22) holds also for $D = [2]$, hence

$$\inf_{\boldsymbol{\omega} \in \mathrm{cl}(\mathcal{X}_{[2]})} \max\{F_{\mathcal{S}_{[2]}}(\boldsymbol{\omega}), I_1(\boldsymbol{\omega})\} \geq \inf_{\boldsymbol{\omega} \in \mathrm{cl}(\mathcal{Z}_{[2]})} F_{\mathcal{S}_{[2]}}(\boldsymbol{\omega})$$

$$\geq \left(1 - \sum_{k=3}^{K} z_k\right) \inf_{\boldsymbol{\lambda} \in \mathbb{R}^K, \lambda_2 > \lambda_1} \left\{ \sum_{k \in [2]} (\lambda_k - \mu_k)^2 \right\}$$

$$\geq \left(1 - \sum_{k=3}^{K} z_k\right) \xi_2 \tag{24}$$

$$\geq \frac{\xi_2}{\log K}, \tag{25}$$

where the second inequality is derived by Proposition 4 with $\theta = 1, \beta = 1, j = 2$ in Appendix D.1, and the last inequality follows from Proposition 7 with $j = 2, i = 3$ in Appendix E.3.

Lower bounding by $\frac{4\xi_2(1-\alpha_2)}{3\log K}$. One can derive another lower bound by using $I_1$. In Theorem 5 (b), $j = 2, \theta = \beta = 1$ in Appendix E.1, we show that $I_1$ is a valid lower semi-continuous rate function for an LDP upper bound (3) for the process $\{\boldsymbol{\omega}(T)\}_{T \geq 1}$. In Corollary 3 in Appendix E.3.1, we further show that $I_1(\boldsymbol{z}) \geq \underline{I}_1(\boldsymbol{z})$ for $\boldsymbol{z} \in \mathcal{Z}_{[2]}$, where

$$\underline{I}_1(\boldsymbol{z}) = \frac{4}{3\log K} \left[ \left( (1 + \zeta_2) \sqrt{\frac{z_{\sigma(3)}}{1 - \sum_{k=4}^{K} z_{\sigma(k)}}} - \frac{1}{2} \right)_+ \right]^2, \quad \forall \boldsymbol{z} \in \mathcal{Z}_{[2]}. \tag{26}$$

Instead of using (25), we lower bound $F_{S_{[2]}}(z)$ as:

$$F_{S_{[2]}}(z) \geq \left(1 - \sum_{k=3}^{K} z_{\sigma(k)}\right) \xi_2$$

$$= \xi_2 \left(1 - \sum_{k=4}^{K} z_{\sigma(k)}\right) \left(1 - \frac{z_{\sigma(3)}}{1 - \sum_{k=4}^{K} z_{\sigma(k)}}\right)$$

$$\geq \frac{4\xi_2}{3\overline{\log}K} \left(1 - \frac{z_{\sigma(3)}}{1 - \sum_{k=4}^{K} z_{\sigma(k)}}\right), \tag{27}$$

where the first inequality follows the derivation of (24) and the last inequality stems from Proposition 7 with $\theta = 1, j = 2, i = 4$ in Appendix E.3. Since $F_{S_{[2]}}(z)$ and $I_1(z)$ are lower bounded by the functions of $\alpha = \frac{z_{\sigma(3)}}{1 - \sum_{k=4}^{K} z_{\sigma(k)}}$ given in (26) and (27), we have:

$$\inf_{\boldsymbol{\omega} \in \mathrm{cl}(\mathcal{X}_{[2]})} \max\{F_{S_{[2]}}(\boldsymbol{\omega}), \underline{I}_1(\boldsymbol{\omega})\} \geq \frac{4}{3\overline{\log}K} \inf_{\alpha \in \mathbb{R}} \max\{\xi_2(1-\alpha), [((1+\zeta_2)\sqrt{\alpha} - \tfrac{1}{2})_+]^2\}$$

$$\geq \frac{4\xi_2(1-\alpha_2)}{3\overline{\log}K}, \tag{28}$$

where the last inequality is due to Lemma 23 in Appendix H and $\alpha_2$ is defined in (19). Hence, the maximum of the r.h.s. of (25) and (28) is a lower bound for $\inf_{\boldsymbol{\omega} \in \mathrm{cl}(\mathcal{X}_{[2]})} \max\{F_{S_{[2]}}(\boldsymbol{\omega}), \underline{I}_1(\boldsymbol{\omega})\}$

$\square$

### C.1.3   Upper bound for $\mathbb{P}_{\boldsymbol{\mu}}[\ell_j = 1]$ for $j \in \{3, \ldots, K-1\}$

**Lemma 6.** *Let* $\boldsymbol{\mu} \in [0,1]^K$, $j \in \{3, \ldots, K-1\}$. *Under* CR-C,

$$\lim_{T \to \infty} \frac{1}{T} \log \frac{1}{\mathbb{P}_{\boldsymbol{\mu}}[\ell_j = 1]} \geq \frac{2 \min\left\{\max\left\{\frac{\xi_j \overline{\log}(j+1)(1-\alpha_j)}{\overline{\log}j}, \xi_j\right\}, \bar{\xi}_j\right\}}{j\overline{\log}K},$$

*where* $\alpha_j \in \mathbb{R}$ *is the real number such that*

$$\frac{2\xi_j(1-\alpha_j)}{j\overline{\log}j} = \left[\left((1+\zeta_j)\sqrt{\alpha_j} - \sqrt{\frac{1}{(j+1)\overline{\log}(j+1)}}\right)_+\right]^2. \tag{29}$$

*Proof.* Without loss of generality, we assume $\theta_0 T > K$ and $\frac{\theta_0 T}{K} \in \mathbb{N}$.
Observe that $\mathbb{P}_{\boldsymbol{\mu}}[\ell_j = 1] = \sum_{J \in \mathcal{J}} \mathbb{P}_{\boldsymbol{\mu}}[\ell_j = 1, \mathcal{C}_j = J]$, which directly implies

$$\lim_{T \to \infty} \frac{1}{T} \log \frac{1}{\mathbb{P}_{\boldsymbol{\mu}}[\ell_j = 1]} \geq \min_{J \in \mathcal{J}} \lim_{T \to \infty} \frac{1}{T} \log \frac{1}{\mathbb{P}_{\boldsymbol{\mu}}[\ell_j = 1, \mathcal{C}_j = J]}. \tag{30}$$

There are $j$ arms remaining while discarding $\ell_j$. The possible allocation $\boldsymbol{\omega}(t)$ belongs to the set

$$\mathcal{X}_j = \left\{\boldsymbol{x} \in \Sigma : \exists \sigma : [K] \mapsto [K] \text{ s.t. } x_{\sigma(1)} = \ldots = x_{\sigma(j)} > x_{\sigma(j+1)} > \ldots > x_{\sigma(K)} > 0\right\}.$$

Suppose that in round $t$, $\mathcal{C}_j = J$ for some $J \in \mathcal{J}$ and let $\tau = \sum_{k \notin J} N_k(t) \geq \tilde{\theta}_0 T$, where $\tilde{\theta}_0 = (K-j)\theta_0/K$, be the number of times arms outside $J$ are pulled. While $\ell_j = \ell(t) = 1$, we must have $\hat{\boldsymbol{\mu}}(t) \in \mathcal{S}_J(\frac{(t-\tau)\overline{\log}j}{T-\tau})$, where

$$\mathcal{S}_J(\beta) = \left\{\boldsymbol{\lambda} \in [0,1]^K : \min_{k \in J, k \neq 1} \lambda_k - \lambda_1 \geq G(\beta)\right\}, \forall \beta > 0,$$

because $\frac{(t-\tau)\overline{\log}j}{T-\tau} = \frac{\sum_{k \in \mathcal{C}_j} N_k(t)\overline{\log}j}{T - \sum_{k \notin \mathcal{C}_j} N_k(t)}$ (recall the discarding condition (11)). Now further introduce $\forall \theta, \beta \in (0,1]$,

$$\mathcal{X}_J(\theta, \beta) = \left\{\boldsymbol{x} \in \mathcal{X}_j : (\forall k \in J, x_k = \max_{k' \in [K]} x_{k'}), \frac{\theta \sum_{k \in J} x_k \overline{\log}j}{1 - \theta \sum_{k \notin J} x_k} = \beta\right\}.$$

We then have:

$$\mathbb{P}_{\boldsymbol{\mu}}\left[\ell_j = 1, \mathcal{C}_j = J\right] \leq \sum_{t \geq \theta_0 T} \sum_{\tau \geq \tilde{\theta}_0 T} \mathbb{P}_{\boldsymbol{\mu}}\left[\hat{\boldsymbol{\mu}}(t) \in \mathcal{S}_J(\frac{(t-\tau)\overline{\log}j}{T-\tau}), \boldsymbol{\omega}(t) \in \mathcal{X}_J(\frac{t}{T}, \frac{(t-\tau)\overline{\log}j}{T-\tau})\right].$$

Applying Theorem 10 in Appendix G with $\mathcal{E} = \{\ell_j = 1, \mathcal{C}_j = J\}$,

$$\mathcal{S}_{\theta,\gamma} = \begin{cases} \mathcal{S}_J(\frac{(\theta-\gamma)\overline{\log}j}{1-\gamma}), & \text{if } G(\frac{(\theta-\gamma)\overline{\log}j}{1-\gamma}) \leq 1, \\ \mathcal{S}_J(G^{-1}(1)), & \text{otherwise,} \end{cases}, \text{ and } W_{\theta,\gamma} = \mathcal{X}_J(\theta, \frac{(\theta-\gamma)\overline{\log}j}{1-\gamma}),$$

(notice that $\beta = \frac{(t-\tau)\overline{\log}j}{T-\tau} = \frac{(\theta-\gamma)\overline{\log}j}{1-\gamma}$ and $\mathcal{S}_J(\beta) = \emptyset$ if $G(\beta) > 1$) yields that

$$\varlimsup_{T\to\infty} \frac{1}{T} \log \frac{1}{\mathbb{P}_{\boldsymbol{\mu}}\left[\ell_j = 1, \mathcal{C}_j = J\right]} \geq$$
$$\inf_{\theta \in [\theta_0,1] \cap \mathbb{Q}} \inf_{\gamma \in [\tilde{\theta}_0,1] \cap \mathbb{Q}} \inf_{\boldsymbol{x} \in \mathrm{cl}(\mathcal{X}_J(\theta, \frac{(\theta-\gamma)\overline{\log}j}{1-\gamma}))} \theta \max\{F_{\mathcal{S}_J(\frac{(\theta-\gamma)\overline{\log}j}{1-\gamma})}(\boldsymbol{x}), I_\theta(\boldsymbol{x})\}. \quad (31)$$

Further introduce for all $i \in \{j, \ldots, K\}$,

$$\mathcal{X}_{j,i}(\theta) = \left\{\boldsymbol{x} \in \mathcal{X}_j : \theta x_{\sigma(i)} i \overline{\log} i > 1 - \theta \sum_{k=i+1}^{K} x_{\sigma(k)}\right\}.$$

In Theorem 5 with (a) in Appendix E.1, we show that $I_\theta(\boldsymbol{\omega}) = \infty$ for all $\boldsymbol{\omega} \in \cup_{i=j}^{K} \mathcal{X}_{2,i}(\theta)$, hence any $\boldsymbol{\omega} \in \cup_{i=j}^{K} \mathcal{X}_{j,i}(\theta)$ will not be the minimizer on the r.h.s. of (31). In the following, we define

$$\mathcal{Z}_J(\theta, \beta) = \mathcal{X}_J(\theta, \beta) \setminus \cup_{i=j}^{K} \mathcal{X}_{j,i}(\theta).$$

Consider any $\boldsymbol{z} \in \mathcal{Z}_J(\theta, \beta)$, as $\boldsymbol{z} \notin \mathcal{X}_{j,j}(\theta)$, we have

$$\frac{(\theta-\gamma)\overline{\log}j}{1-\gamma} = \frac{\theta z_{\sigma(j)} j \overline{\log}j}{1 - \theta \sum_{k>j} z_{\sigma(k)}} < 1.$$

Therefore, after excluding the points in $\cup_{i=j}^{K} \mathcal{X}_{j,i}(\theta)$ and setting $\beta = \frac{(\theta-\gamma)\overline{\log}j}{1-\gamma}$, the r.h.s of (31) can be lower bounded by

$$\inf_{\theta, \beta \in (0,1] \cap \mathbb{Q}} \inf_{\boldsymbol{z} \in \mathrm{cl}(\mathcal{Z}_J(\theta,\beta))} \theta \max\{F_{\mathcal{S}_J(\beta)}(\boldsymbol{z}), I_\theta(\boldsymbol{z})\}. \quad (32)$$

For lower bounding (32), we consider two cases: (a) $J \neq [j]$; (b) $J = [2]$.

**(a) The case where $J \neq [j]$.** If $\boldsymbol{z} \in \mathcal{Z}_J(\theta, \beta)$,

$$\theta F_{\mathcal{S}_J(\beta)}(\boldsymbol{z}) = \theta \inf_{\boldsymbol{\lambda} \in \mathrm{cl}(S_J(\beta))} \Psi(\boldsymbol{\lambda}, \boldsymbol{z})$$

$$\geq 2 \inf\left\{\sum_{k \in J} \theta z_k (\lambda_k - \mu_k)^2 : \min_{k \in J, k \neq 1} \lambda_k - \lambda_1 \geq G(\beta)\right\}$$

$$= \frac{2}{j \overline{\log}j} \left(1 - \theta \sum_{k \notin J} z_k\right) \beta \inf\left\{\sum_{k \in J} (\lambda_k - \mu_k)^2 : \min_{k \in J, k \neq 1} \lambda_k - \lambda_1 \geq G(\beta)\right\} \quad (33)$$

$$\geq \frac{2\bar{\xi}_j}{j \overline{\log}j}(1 - \theta \sum_{k \notin J} z_k)$$

$$\geq \frac{2\bar{\xi}_j}{j \overline{\log}K}, \quad (34)$$

where the first inequality is due to Assumption 1; the second equality uses the fact that $\forall k \in J$, $\theta z_k = \frac{(1-\theta \sum_{k' \notin J} z_{k'})\beta}{j \overline{\log}j}$ (as $\boldsymbol{z} \in \mathcal{Z}_J(\theta, \beta)$); the third follows from Corollary 1 in Appendix D.1; the last one is a consequence of Proposition 7 with $i = j + 1$ in Appendix E.3.

**(b) The case where $J = [j]$.** We will show that both $\frac{2\xi_j}{j\overline{\log}K}$ and $\frac{2\xi_j\overline{\log}(j+1)(1-\alpha_j)}{j\overline{\log}j\overline{\log}K}$ are lower bounds for the r.h.s. of (32). The maximum of these becomes our lower bound. Together with the conclusion obtained in the case (a), we complete the proof of Lemma 6.

Lower bounding by $\frac{2\xi_j}{j\overline{\log}K}$. By Proposition 4 and Proposition 7, we can further lower bound the r.h.s. of (32) as

$$\theta F_{S_{[j]}(\beta)}(\boldsymbol{z}) \geq \frac{2}{j\overline{\log}j}\left(1 - \theta\sum_{k>j}z_k\right)\beta\inf\left\{\sum_{k\in[j]}(\lambda_k - \mu_k)^2 : \min_{k\in[j],k\neq 1}\lambda_k - \lambda_1 \geq G(\beta)\right\}$$

$$\geq \frac{2\xi_j}{j\overline{\log}j}\left(1 - \theta\sum_{k>j}z_k\right) \tag{35}$$

$$\geq \frac{2\xi_j}{j\overline{\log}K}, \tag{36}$$

where the first inequality corresponds to (33); the second inequality is due to Proposition 4 in Appendix D.1; the last inequality uses Proposition 7 with $i = j + 1$ in Appendix E.3.

Lower bounding by $\frac{2\xi_j\overline{\log}(j+1)(1-\alpha_j)}{j\overline{\log}j\overline{\log}K}$. One can derive another lower bound by using $I_\theta$. In Theorem 5 with (b) in Appendix E.1, we show that $I_\theta$ is a valid lower semi-continuous rate function for an LDP upper bound (3) for the process $\{\boldsymbol{\omega}(\theta T)\}_{T\geq 1}$. And in Corollary 3 in Appendix E.3.1, $I_\theta(\boldsymbol{z}) \geq \underline{I}_\theta(\boldsymbol{z})$ for $\boldsymbol{z} \in \mathcal{Z}_{[j]}(\theta, \beta)$, where

$$\underline{I}_\theta(\boldsymbol{z}) = \frac{\overline{\log}(j+1)}{\theta\overline{\log}K}\left[\left((1+\zeta_j)\sqrt{\frac{\theta z_{\sigma(j+1)}}{1 - \theta\sum_{k=j+2}^{K}z_{\sigma(k)}}} - \sqrt{\frac{1}{(j+1)\overline{\log}(j+1)}}\right)_+\right]^2. \tag{37}$$

Instead of using (36), we lower bound $F_{S_{[J]}(\beta)}(\boldsymbol{z})$ as:

$$\theta F_{S_{[j]}(\beta)}(\boldsymbol{z}) \geq \frac{2\xi_j}{j\overline{\log}j}\left(1 - \theta\sum_{k\notin[j]}z_k\right) = \frac{2\xi_j}{j\overline{\log}j}\left(1 - \theta\sum_{k=j+2}^{K}z_{\sigma(k)}\right)\left(1 - \frac{\theta z_{\sigma(j+1)}}{1 - \theta\sum_{k=j+2}^{K}z_{\sigma(k)}}\right)$$

$$\geq \frac{2\xi_j\overline{\log}(j+1)}{j\overline{\log}j\overline{\log}K}\left(1 - \frac{\theta z_{\sigma(j+1)}}{1 - \theta\sum_{k=j+2}^{K}z_{\sigma(k)}}\right), \tag{38}$$

where the first inequality is from (35) and the last inequality is due to Proposition 7 with $i = j + 2$ in Appendix E.3. Since $\theta F_{S_{[j]}(\beta)}(\boldsymbol{z})$ and $\theta I_\theta(\boldsymbol{z})$ are lower bounded by the functions of $\alpha = \frac{\theta z_{\sigma(j+1)}}{1 - \theta\sum_{k=j+2}^{K}z_{\sigma(k)}}$ given in (37) and (38), we have:

$$\inf_{\theta,\beta\in[0,1]\cap\mathbb{Q}}\theta\inf_{\boldsymbol{z}\in\mathrm{cl}(\mathcal{Z}_{[j]}(\theta,\beta))}\max\{F_{S_{[j]}(\beta)}(\boldsymbol{z}), \underline{I}_\theta(\boldsymbol{z})\} \geq$$

$$\frac{\overline{\log}(j+1)}{\overline{\log}K}\inf_{\alpha\in\mathbb{R}}\max\left\{\frac{2\xi_j(1-\alpha)}{j\overline{\log}j}, \left[\left((1+\zeta_j)\sqrt{\alpha} - \sqrt{\frac{1}{(j+1)\overline{\log}(j+1)}}\right)_+\right]^2\right\}. \tag{39}$$

By Lemma 23 in Appendix H, (39) is lower bounded by $\frac{2\xi_j\overline{\log}(j+1)(1-\alpha_j)}{j\overline{\log}j\overline{\log}K}$, where $\alpha_j$ is described in (29).

$\square$

## C.2  Performance analysis of CR-A

**Theorem 4.** *Let* $\boldsymbol{\mu} \in [0,1]^K$. *Under* CR-A, $\varliminf_{T\to\infty}\frac{1}{T}\log\frac{1}{\mathbb{P}_{\boldsymbol{\mu}}[\hat{i}\neq 1]}$ *is larger than*

$$2\min_{j=2,\ldots,K}\left\{\frac{\min\{\max\{\frac{\psi_j\overline{\log}(j+1)(1-\alpha_j)\mathbb{1}_{\{j\neq K\}}}{\overline{\log}j}, \psi_j\}, \bar{\psi}_j\}}{j\overline{\log}K}\right\},$$

where $\alpha_j \in \mathbb{R}$ is the real number such that

$$\frac{\psi_j(1 - \alpha_j)}{j\overline{\log}j} = \frac{j}{j+1}\left[\left((1 + \varphi_j)\sqrt{\alpha_j} - \sqrt{\frac{1}{(j+1)\overline{\log}(j+1)}}\right)_+\right]^2.$$

We upper bound $\mathbb{P}_{\boldsymbol{\mu}}[\ell_j = 1]$ for (i) $j = K$; (ii) $j = 2$; (iii) $j \in \{3, \dots, K\}$. The upper bound for (i), presented in C.2.1, is the easiest to derive as the only possible allocation before one discards the first arm is uniform among all arms. The bound for (ii), presented in C.2.2, is the second easiest to derive as $\ell_2$ is decided only in the end, namely, in the $T$-th round. The upper bound for (iii), presented in C.2.3, is more involved since we have to consider all possible allocations and rounds. Overall, the analysis is very similar to that of CR-C, and we just sketch the arguments below.

### C.2.1 Upper bound of $\mathbb{P}_{\boldsymbol{\mu}}[\ell_K = 1]$

**Lemma 7.** *Let $\boldsymbol{\mu} \in [0, 1]^K$. Under* CR-A,

$$\varliminf_{T\to\infty}\frac{1}{T}\log\frac{1}{\mathbb{P}_{\boldsymbol{\mu}}[\ell_K = 1]} \geq \frac{2\psi_K}{K\overline{\log}K}.$$

*Proof.* The proof follows the steps similar to those in the proof in Appendix C.1.1.
Applying Theorem (10) to

$$\sum_{t\geq\theta_0 T}^{T}\mathbb{P}_{\boldsymbol{\mu}}[\ell_K = \ell(t) = 1] \leq \sum_{t\geq\theta_0 T}^{T}\mathbb{P}_{\boldsymbol{\mu}}\left[\hat{\boldsymbol{\mu}}(t) \in \mathcal{S}_{\frac{t}{T}}, \boldsymbol{\omega}(t) \in \mathcal{X}_K\right], \tag{40}$$

with

$$\mathcal{S}_\theta = \left\{\boldsymbol{\lambda} \in [0, 1]^K : \lambda_1 \leq \frac{\sum_{k=2}^{K}\lambda_k}{K - 1} - G(\theta\overline{\log}K)\right\}, \forall\theta \in (0, \frac{1}{\overline{\log}K}],$$

we obtain

$$\varliminf_{T\to\infty}\frac{1}{T}\log\frac{1}{\mathbb{P}_{\boldsymbol{\mu}}[\ell_K = 1]} \geq \min_{\theta\in[\theta_0,1]\cap\mathbb{Q}}\theta\{F_{\mathcal{S}_\theta}(1/K, \dots, 1/K), I_\theta(1/K, \dots, 1/K)\}. \tag{41}$$

Next, from Theorem 6 (a) in Appendix E.1, we know that $I_\theta(1/K, \dots, 1/K) = \infty$ if $\theta > 1/\overline{\log}K$. Finally, Assumption 1 and Proposition 6 in Appendix D.2 yields that

$$\min_{\theta\in[\theta_0,1/\overline{\log}(K)]\cap\mathbb{Q}}\theta\{F_{\mathcal{S}_\theta}(1/K, \dots, 1/K), I_\theta(1/K, \dots, 1/K)\} \geq \frac{2\psi_K}{K\overline{\log}K}.$$

$\square$

### C.2.2 Upper bound of $\mathbb{P}_{\boldsymbol{\mu}}[\ell_2 = 1]$

**Lemma 8.** *Let $\boldsymbol{\mu} \in [0, 1]^K$. Under* CR-A,

$$\varliminf_{T\to\infty}\frac{1}{T}\log\frac{1}{\mathbb{P}_{\boldsymbol{\mu}}[\ell_2 = 1]} \geq \frac{\min\{\max\{4\psi_2(1 - \alpha_2), 3\psi_2\}, 3\overline{\psi}_2\}}{3\overline{\log}K},$$

*where $\alpha_2 \in \mathbb{R}$ is the real number such that*

$$3\psi_2(1 - \alpha_2) = 4\left[\left((1 + \varphi_2)\sqrt{\alpha_2} - \frac{1}{2}\right)_+\right]^2. \tag{42}$$

*Proof.* The proof follows the same steps as those of the proof in Appendix C.1.2.
Applying Theorem 1 with $\mathcal{S} = \mathcal{S}_D, W = \mathcal{X}_D$, where

$$\mathcal{S}_D = \left\{\boldsymbol{\lambda} \in [0, 1]^K : \min_{k\in[D]}\lambda_k \geq \lambda_1\right\},$$

we get

$$\varliminf_{T\to\infty} \frac{1}{T} \log \frac{1}{\mathbb{P}_{\boldsymbol{\mu}}[\ell_2 = 1]} \geq \min_{D\in\mathcal{D}} \inf_{\boldsymbol{\omega}\in\text{cl}(\mathcal{X}_D)} \max\{F_{\mathcal{S}_D}(\boldsymbol{\omega}), I_1(\boldsymbol{\omega})\}, \tag{43}$$

Then we exclude the points in $\cup_{i=2}^K \mathcal{X}_{2,i}(1)$ using Theorem 6 (a) in Appendix E.1 and we define

$$\mathcal{Z}_D(1,1) = \mathcal{X}_D \setminus \cup_{i=2}^K \mathcal{X}_{2,i}(1).$$

Consider two cases: (a) $D \neq [2]$; (b) $D = [2]$.

**(a) The case where $D \neq [2]$.** Using Corollary 2 with $\beta = 1, j = 2$ in Appendix D.2 and Proposition 7 with $\theta = 1, j = 2, i = 3$ in Appendix E.2 yields that

$$\inf_{\boldsymbol{\omega}\in\text{cl}(\mathcal{X}_D)} \max\{F_{\mathcal{S}_D}(\boldsymbol{\omega}), I_1(\boldsymbol{\omega})\} \geq \frac{\bar{\psi}_2}{\overline{\log}K}.$$

**(b) The case where $D = [2]$.** We show that both $\frac{\psi_2}{\overline{\log}K}$ and $\frac{4\psi_2(1-\alpha_2)}{3\overline{\log}K}$ are lower bounds for $\varliminf_{T\to\infty} \frac{1}{T} \log \frac{1}{\mathbb{P}_{\boldsymbol{\mu}}[\hat{\boldsymbol{\mu}}(T)\in\mathcal{S}_{[2]},\boldsymbol{\omega}(T)\in\mathcal{Z}_{[2]}]}$. The maximum of these hence becomes our lower bound. Together with the conclusion obtained in the case (a), we complete the proof of Lemma 8.

Lower bounding by $\frac{\psi_2}{\overline{\log}K}$. Applying Proposition 6 with $\theta = 1, \beta = 1, j = 2$ in Appendix D.2, and Proposition 7 with $j = 2, i = 3$ in Appendix E.3, we can obtain:

$$\inf_{\boldsymbol{\omega}\in\text{cl}(\mathcal{X}_{[2]})} \max\{F_{\mathcal{S}_{[2]}}(\boldsymbol{\omega}), I_1(\boldsymbol{\omega})\} \geq \frac{\psi_2}{\overline{\log}K}.$$

Lower bounding by $\frac{4\psi_2(1-\alpha_2)}{3\overline{\log}K}$. In Theorem 6 (b), $j = 2, \theta = \beta = 1$ in Appendix E.1, we show that $I_1$ is a valid rate function for an LDP upper bound (3) for the process $\{\boldsymbol{\omega}(T)\}_{T\geq 1}$. And Corollary 4 in Appendix E.3.2 show $I_1(\boldsymbol{z}) > \underline{I}_1(\boldsymbol{z})$ for $\boldsymbol{z} \in \mathcal{Z}_{[2]}$, where

$$\underline{I}_1(\boldsymbol{z}) = \frac{16}{9\overline{\log}K} \left[ \left( (1+\varphi_2)\sqrt{\frac{z_{\sigma(3)}}{1 - \sum_{k=4}^K z_{\sigma(k)}}} - \frac{1}{2} \right)_+ \right]^2, \forall \boldsymbol{z} \in \mathcal{Z}_{[2]}. \tag{44}$$

Also, we lower bound $F_{S_{[2]}}(\boldsymbol{z})$ by Proposition 7 with $\theta = 1, j = 2, i = 4$ in Appendix E.3:

$$F_{S_{[2]}}(\boldsymbol{z}) \geq \frac{4\psi_2}{3\overline{\log}K} \left( 1 - \frac{z_{\sigma(3)}}{1 - \sum_{k=4}^K z_{\sigma(k)}} \right). \tag{45}$$

Observe that $F_{S_{[2]}}(\boldsymbol{z})$ and $\underline{I}_1(\boldsymbol{z})$ are lower bounded by the functions of $\alpha = \frac{z_{\sigma(3)}}{1-\sum_{k=4}^K z_{\sigma(k)}}$ given in (44) and (45). Applying Lemma 23 in Appendix H yields that

$$\inf_{\boldsymbol{\omega}\in\text{cl}(\mathcal{X}_{[2]})} \max\{F_{\mathcal{S}_{[2]}}(\boldsymbol{\omega}), \underline{I}_1(\boldsymbol{\omega})\} \geq \frac{4\psi_2(1-\alpha_2)}{3\overline{\log}K}. \tag{46}$$

$\square$

### C.2.3 Upper bound for $\mathbb{P}_{\boldsymbol{\mu}}[\ell_j = 1]$ for $j \in \{3, \ldots, K-1\}$

**Lemma 9.** *Let* $\boldsymbol{\mu} \in [0,1]^K$, $j \in \{3, \ldots, K-1\}$. *Under* CR-A,

$$\varliminf_{T\to\infty} \frac{1}{T} \log \frac{1}{\mathbb{P}_{\boldsymbol{\mu}}[\ell_j = 1]} \geq \frac{2 \min\left\{ \max\left\{ \frac{\psi_j \overline{\log}(j+1)(1-\alpha_j)}{\overline{\log}j}, \psi_j \right\}, \bar{\psi}_j \right\}}{j\overline{\log}K},$$

*where* $\alpha_j \in \mathbb{R}$ *is the real number such that*

$$\frac{\psi_j(1-\alpha_j)}{j\overline{\log}j} = \frac{j}{j+1} \left[ \left( (1+\varphi_j)\sqrt{\alpha_j} - \sqrt{\frac{1}{(j+1)\overline{\log}(j+1)}} \right)_+ \right]^2. \tag{47}$$

*Proof.* We proceed as in Appendix C.1.3. We have:

$$\lim_{T\to\infty}\frac{1}{T}\log\frac{1}{\mathbb{P}_{\boldsymbol{\mu}}\left[\ell_j=1\right]}\geq\min_{J\in\mathcal{J}}\lim_{T\to\infty}\frac{1}{T}\log\frac{1}{\mathbb{P}_{\boldsymbol{\mu}}\left[\ell_j=1,\mathcal{C}_j=J\right]}. \tag{48}$$

We then introduce

$$\mathcal{S}_J(\beta)=\left\{\boldsymbol{\lambda}\in[0,1]^K:\frac{\sum_{k\in J,k\neq1}\lambda_k}{j-1}-\lambda_1\geq G(\beta)\right\}$$

$$\mathcal{X}_J(\theta,\beta)=\left\{\boldsymbol{z}\in\mathcal{X}_j:(\forall k\in J,\ x_k=\max_{k'\in[K]}x_{k'}),\ \frac{\theta\sum_{x\in J}z_k\overline{\log}j}{1-\theta\sum_{k\notin J}x_k}=\beta\right\}.$$

Theorem 10 yields that for each $J\in\mathcal{J}$,

$$\lim_{T\to\infty}\frac{1}{T}\log\frac{1}{\mathbb{P}_{\boldsymbol{\mu}}\left[\ell_j=1,\mathcal{C}_j=J\right]}\geq\inf_{\theta,\beta\in(0,1]\cap\mathbb{Q}}\theta\inf_{\boldsymbol{z}\in\mathrm{cl}(\mathcal{Z}_J(\theta,\beta))}\max\{F_{\mathcal{S}_J(\beta)}(\boldsymbol{z}),I_\theta(\boldsymbol{z})\}, \tag{49}$$

where $\mathcal{Z}_J(\theta,\beta)=\mathcal{X}_J(\theta,\beta)\setminus\cup_{i=j}^K\mathcal{X}_{j,i}(\theta)$.

(a) The case where $J\neq[j]$. Corollary 2 in Appendix D.2 and Proposition 7 with $i=j+1$ in Appendix E.3 yields:

$$\theta F_{\mathcal{S}_J(\beta)}(\boldsymbol{z})\geq\frac{2\bar{\psi}_j}{j\overline{\log}K}.$$

(b) The case where $J=[j]$. We show both $\frac{2\psi_j}{j\overline{\log}K}$ and $\frac{2\psi_j\overline{\log}(j+1)(1-\alpha_j)}{j\overline{\log}j\overline{\log}K}$ are lower bounds of (49). The maximum of these becomes our lower bound.

Lower bounding $\frac{2\psi_j}{j\overline{\log}K}$. By Proposition 6 and Proposition 7, Proposition 6 in Appendix D.2 and Proposition 7 with $i=j+1$ in Appendix E.3, we have

$$\theta F_{S_J(\beta)}(\boldsymbol{z})\geq\frac{2\psi_j}{j\overline{\log}K}. \tag{50}$$

Lower bounding by $\frac{2\psi_j\overline{\log}(j+1)(1-\alpha_j)}{j\overline{\log}j\overline{\log}K}$. A similar argument as above implies

$$(49)\geq\frac{2\overline{\log}(j+1)}{\overline{\log}K}\inf_{\alpha\in\mathbb{R}}\max\left\{\frac{\psi_j(1-\alpha)}{j\overline{\log}j},\frac{j}{j+1}\left[\left((1+\varphi_j)\sqrt{\alpha}-\sqrt{\frac{1}{(j+1)\overline{\log}(j+1)}}\right)_+\right]^2\right\}. \tag{51}$$

By Lemma 23 in Appendix H, (49) is lower bounded by $\frac{2\psi_j\overline{\log}(j+1)(1-\alpha_j)}{j\overline{\log}j\overline{\log}K}$.

$\square$

# D  Optimization Problems

This section provides results related to the various optimization problems we encounter in the paper. In D.1, we compute the $\xi_j$'s appearing in the performance guarantees of SR and CR-C, and prove other useful results. In D.2, we focus on computing the $\psi_j$'s, useful for the performance analysis of CR-A.

## D.1  Optimization problems for SR and CR-C

Let $j \in \{2, \ldots, K\}$ and let $\boldsymbol{\mu} \in \mathbb{R}^K$ such that $\mu_1 > \mu_2 \geq \ldots \geq \mu_K$. Denote by $\xi_j$ the optimal value of the following optimization problem:

$$\inf \left\{ \sum_{k=1}^{j} (\lambda_k - \mu_k)^2 : \boldsymbol{\lambda} \in [0,1]^K, \lambda_1 \leq \min_{k \neq 1} \lambda_k \right\}. \tag{52}$$

We first show Proposition 1, restated below for convenience, and deduce some related results.

**Proposition 1.** *We have:*

$$\xi_j = \begin{cases} \sum_{k=1,j} \left( \mu_k - \frac{\mu_1 + \mu_j}{2} \right)^2, & \text{if } \mu_{j-1} \geq \frac{\mu_1 + \mu_j}{2}, \\ \sum_{k=1,j-1,j} \left( \mu_k - \frac{\mu_1 + \mu_{j-1} + \mu_j}{3} \right)^2, & \text{if } \mu_{j-1} < \frac{\mu_1 + \mu_j}{2}, \mu_{j-2} \geq \frac{\mu_1 + \mu_{j-1} + \mu_j}{3}, \\ \vdots & \vdots \\ \sum_{k=1}^{j} \left( \mu_k - \frac{\sum_{i=1}^{j} \mu_i}{j} \right)^2, & \text{if } \mu_{j-1} < \frac{\mu_1 + \mu_j}{2}, \ldots, \mu_2 \leq \frac{\mu_1 + \mu_3 + \ldots + \mu_j}{j-1}. \end{cases}$$

*Proof.* The objective function and the functions defining the constraints in (52) are all convex. There exists $\boldsymbol{\lambda} \in \mathbb{R}^K$ s.t. all the constraints are strict (Slater condition). Hence we can identify the solution of (52) by just verifying the KKT conditions. The Lagrangian of the problem is

$$\mathcal{L}_{\boldsymbol{\mu}}(\boldsymbol{\lambda}, \eta_2, \ldots, \eta_j) = \frac{1}{2} \sum_{k=1}^{j} (\lambda_k - \mu_k)^2 + \sum_{k=2}^{j} \eta_k (\lambda_1 - \lambda_k), \text{ for } (\boldsymbol{\lambda}, \boldsymbol{\eta}) \in \mathbb{R}^K \times \mathbb{R}_{\geq 0}^{j-1}.$$

Let $(\boldsymbol{\lambda}^\star, \boldsymbol{\eta}^\star)$ be a saddle point of $\mathcal{L}$. It satisfies KKT conditions:

$$\lambda_1^\star \leq \lambda_k^\star, \text{ for } k = 2, \ldots, j, \qquad \text{(Primal Feasibility)}$$
$$\eta_k^\star \geq 0, \text{ for } k = 2, \ldots, j, \qquad \text{(Dual Feasibility)}$$

$$\lambda_1^\star - \mu_1 + \sum_{k=2}^{j} \eta_k^\star = 0; \lambda_k^\star - \mu_k - \eta_k^\star = 0, \text{ for } k = 2, \ldots, j, \qquad \text{(Stationarity)}$$

$$\eta_k^\star (\lambda_1^\star - \lambda_k^\star) = 0, \text{ for } k = 2, \ldots, j. \qquad \text{(Complementarity)}$$

In case $\mu_{j-1} \geq \frac{\mu_1 + \mu_j}{2}$, one can easily see the point $(\boldsymbol{\lambda}^\star, \boldsymbol{\eta}^\star)$ defined as $\lambda_1^\star = \lambda_j^\star = \frac{\mu_1 + \mu_j}{2}$, $\lambda_k^\star = \mu_k, \forall k \notin \{1, j\}$, and $\eta_k^\star = 0, \forall k = 2, \ldots, j-1, \eta_j^\star = \frac{\mu_1 - \mu_j}{2}$ satisfies the KKT conditions listed above. As for the other cases, one can also easily find the solutions in a similar manner. $\square$

We are now interested in quantifying the impact of $\boldsymbol{\mu}$ on the value of $\xi_j$. We investigate this impact in the following two propositions.

**Proposition 2.** *Assume that $\xi_j = \sum_{b \in B} (\mu_b - A)^2$ for some $B \subseteq [j]$ and for $A = \frac{\sum_{b \in B} \mu_b}{|B|}$. Let $S$ be such that $S_1 = \sum_{b \in B, b \neq 1} S_b$, where $S_b \geq 0$ for all $b \neq 1$. Consider another parameter $\boldsymbol{\mu}'$ defined as $\mu_1' = \mu_1 + S_1$ and $\mu_b' = \mu_b - S_b$ for all $b \in B, b \neq 1$. Then (i) $\frac{\sum_{b \in B} \mu_b'}{|B|} = A$; (ii)*

$$\inf \left\{ \sum_{k \in B} (\lambda_k' - \mu_k')^2 : \boldsymbol{\lambda}' \in [0,1]^K, \lambda_1' \leq \min_{b \in B} \lambda_b' \right\} = \sum_{k \in B} (\mu_b' - A)^2.$$

*Proof.* (i) is trivial. We now prove (ii). Using Proposition 1 and the fact that $\xi_j = \sum_{b \in B}(\mu_b - A)^2$, we get that

$$\forall b \neq 1, b \in B, \ \mu_b < \frac{\mu_1 + \sum_{k>b}\mu_k}{j - b + 1}. \tag{53}$$

Also, as $S_b \geq 0$,

$$\mu'_b = \mu_b - S_b \leq \mu_b < \frac{\mu_1 + \sum_{k>b}\mu_k}{j - b + 1}$$

$$= \frac{1}{j - b + 1}\left(\mu_1 - S_1 + \sum_{k>b}\mu_k + \sum_{k>b}S_k\right)$$

$$= \frac{\mu'_1 + \sum_{k>b}\mu'_k}{j - b + 1},$$

where the second inequality is due to (53). By (i) and Proposition 1 again, we conclude the proof. $\square$

**Proposition 3.** *Consider the optimization problem (52) instantiated with another $\mu' \in [0, 1]^K$ which satisfies that $\mu'_1 \geq \mu_1$ and $\mu'_k \leq \mu_k$ for all $k = 2, \ldots, j$, and denote its value by $\xi'_j$. Then $\xi'_j \geq \xi_j$.*

*Proof.* Consider the Lagrangians of the two optimization problems: $\mathcal{L}_{\boldsymbol{\mu}}$ and $\mathcal{L}_{\boldsymbol{\mu}'}$. The corresponding Lagrange dual functions are: $g_{\boldsymbol{\mu}}(\boldsymbol{\eta}) = \min_{\boldsymbol{\lambda} \in \mathbb{R}^K} \mathcal{L}_{\boldsymbol{\mu}}(\boldsymbol{\lambda}, \boldsymbol{\eta})$ and $g_{\boldsymbol{\mu}'}(\boldsymbol{\eta}) = \min_{\boldsymbol{\lambda} \in \mathbb{R}^K} \mathcal{L}_{\boldsymbol{\mu}'}(\boldsymbol{\lambda}, \boldsymbol{\eta})$ and one can easily verify that

$$g_{\boldsymbol{\mu}}(\boldsymbol{\eta}) = \frac{1}{2}\left[(\sum_{k=2}^{j}\eta_k)^2 + \sum_{k=2}^{j}\eta_k^2\right] + \sum_{k=2}^{j}\eta_k(\mu_1 - \mu_k - \sum_{i \neq k}\eta_i),$$

$$g_{\boldsymbol{\mu}'}(\boldsymbol{\eta}) = \frac{1}{2}\left[(\sum_{k=2}^{j}\eta_k)^2 + \sum_{k=2}^{j}\eta_k^2\right] + \sum_{k=2}^{j}\eta_k(\mu'_1 - \mu'_k - \sum_{i \neq k}\eta_i).$$

Recall $\mu_1 - \mu_k \leq \mu'_1 - \mu'_k$ and $\boldsymbol{\eta} \in \mathbb{R}_{\geq 0}^{j-1}$, hence $g_{\boldsymbol{\mu}}(\boldsymbol{\eta}) \leq g_{\boldsymbol{\mu}'}(\boldsymbol{\eta})$ for all $\boldsymbol{\eta} \in \mathbb{R}_{\geq 0}^{j-1}$. For (52), Slater condition holds clearly, hence strong duality follows (see e.g. (Boyd et al., 2004) Chapter 5.5.3). Thus, $\xi_j = \max_{\boldsymbol{\eta} \in \mathbb{R}_+^{j-1}} g_{\boldsymbol{\mu}}(\boldsymbol{\eta}) \leq \max_{\boldsymbol{\eta} \in \mathbb{R}_+^{j-1}} g_{\boldsymbol{\mu}'}(\boldsymbol{\eta}) = \xi'_j$. $\square$

The following result relates the function $G$ to $\xi_j$ and is instrumental in the proof of Theorem 3.

**Proposition 4.** $\forall \beta \in (0, 1]$, $\boldsymbol{\mu} \in [0, 1]^K$, *and* $2 \leq j \leq K$, *one has*

$$\beta \inf\left\{\sum_{k=1}^{j}(\lambda_k - \mu_k)^2 : \boldsymbol{\lambda} \in [0, 1]^K, \lambda_1 \leq \min_{k=2,\ldots,j}\lambda_k - G(\beta)\right\} \geq \xi_j. \tag{54}$$

*Proof.* Let $B \subseteq [j]$ s.t. $\xi_j = \sum_{b \in B}(\mu_b - A)^2$, where $A = \frac{\sum_{b \in B}\mu_b}{|B|}$. Using the fact that $\sum_{k \notin B}(\mu_k - \lambda_k)^2 \geq 0$ for all $\boldsymbol{\lambda} \in \mathbb{R}^K$, one can deduce that

$$\text{l.h.s. of (54)} \geq \beta \inf\left\{\sum_{b \in B}(\lambda_b - \mu_b)^2 : \lambda_1 \leq \lambda_b - G(\beta), \forall b \in B, b \neq 1\right\}$$

$$\geq \beta \inf\left\{\sum_{b \in B}(\lambda_b - \mu_b)^2 : \lambda_1 \leq \lambda_b - (\mu_1 - \mu_b)G(\beta), \forall b \in B, b \neq 1\right\}$$

$$= \beta \inf\left\{\sum_{b \in B}(\lambda_b - \mu_b)^2 : \lambda_1 + (\mu_1 - A)G(\beta) \leq \lambda_k - (A - \mu_b)G(\beta), \forall b \in B, b \neq 1\right\} \tag{55}$$

where the second inequality comes from $1 \geq \mu_1 - \mu_k$. Now introduce $\boldsymbol{\lambda}'$ and $\boldsymbol{\mu}'$ as

$$\lambda'_1 = \lambda_1 + (\mu_1 - A)G(\beta), \ \lambda'_b = \lambda_b - (A - \mu_b)G(\beta), \forall b \neq 1, b \in B;$$

$$\mu'_1 = \mu_1 - (\mu_1 - A)G(\beta), \ \mu'_b = \mu_b + (A - \mu_b)G(\beta), \forall b \neq 1, b \in B.$$

These allow us to write the r.h.s. of (55) as the value of the following optimization problem:

$$\beta \inf \left\{ \sum_{b \in B} (\lambda'_b - \mu'_b)^2 : \lambda'_1 \leq \min_{b \in B} \lambda'_b \right\}. \tag{56}$$

Applying Proposition 2 with $S_b = (A - \mu_b)G(\beta)$ for all $b \in B, b \neq 1$ yields that the value of (56) is

$$\beta \sum_{b \in B} (\mu'_b - A)^2 = \beta \left( (\mu_1 + (\mu_1 - A)G(\beta) - A)^2 + \sum_{b \in B, b \neq 1} (A - \mu_b + (A - \mu_b)G(\beta))^2 \right). \tag{57}$$

Recall that $G(\beta) = 1/\sqrt{\beta} - 1$. Hence, (57) is larger than $\sum_{b \in B} (\mu_b - A)^2 = \xi_j$. $\qquad\square$

In Proposition 4, the top-$j$ arms only are considered. We can prove similar results for any $J \in \mathcal{J}$, by combining Proposition 3 to the arguments of the previous proof.

**Corollary 1.** $\forall \beta \in (0, 1]$, $\boldsymbol{\mu} \in [0, 1]^K$, $2 \leq j \leq K$, and $J \in \mathcal{J}$, $J \neq [j]$, one has

$$\beta \inf \left\{ \sum_{k=1}^{K} (\lambda_k - \mu_k)^2 : \boldsymbol{\lambda} \in [0, 1]^K, \lambda_1 \leq \min_{k \in J, k \neq 1} \lambda_k - G(\beta) \right\} \geq \bar{\xi}_j. \tag{58}$$

*Proof.* Let $J \in \mathcal{J}$, $J \neq [j]$ be fixed, we denote the indexes in $J$ by $\{\tilde{1}, \tilde{2}, \ldots, \tilde{j}\}$ such that $\tilde{1} < \tilde{2} < \ldots < \tilde{j}$. One can repeat the argument in the proof of Proposition 4 to obtain that the l.h.s. of (58) is larger than

$$\inf \left\{ \sum_{k=1}^{K} (\lambda_k - \mu_k)^2 : \boldsymbol{\lambda} \in [0, 1]^K, \lambda_1 \leq \min_{k \in J, k \neq 1} \lambda_k \right\}. \tag{59}$$

Since every $J \in \mathcal{J}$ includes 1, $\tilde{1} = 1$. Also, since $J \neq [j]$, we have $\tilde{2} \geq 2, \ldots, \tilde{j} \geq j + 1$. Because we assume that $\mu_1 > \mu_2 \geq \ldots \geq \mu_K$, Proposition 3 yields that the value of (59) is larger than $\bar{\xi}_j$. $\qquad\square$

## D.2 Optimization problem for `CR-A`

The following proposition is the analogue of Proposition 3 for $\psi_j$.

**Proposition 5.** *Let* $\boldsymbol{\mu}' \in [0, 1]^K$ *such that* $\mu'_1 \geq \mu_1$ *and* $\mu'_k \leq \mu_k$ *for all* $k = 2, \ldots, j$. *Define* $\psi'_j = \frac{j-1}{j}(\mu'_1 - \frac{\sum_{k=2}^{j} \mu'_k}{j-1})^2$. *Then,* $\psi'_j \geq \psi_j$.

*Proof.* The result simply follows from the following inequality:

$$\psi'_j = \frac{j-1}{j}(\mu'_1 - \frac{\sum_{k=2}^{j} \mu'_k}{j-1})^2 \geq \frac{j-1}{j}(\mu_1 - \frac{\sum_{k=2}^{j} \mu'_k}{j-1})^2 \geq \frac{j-1}{j}(\mu_1 - \frac{\sum_{k=2}^{j} \mu_k}{j-1})^2 = \psi_j.$$

$\qquad\square$

We use the following result in the proof of Theorem 4.

**Proposition 6.** $\forall \beta \in (0, 1], \boldsymbol{\mu} \in [0, 1]^K$ *with* $\mu_1 > \mu_2 \geq \ldots \geq \mu_K$, *and* $2 \leq j \leq K$, *one has*

$$\beta \inf \left\{ \sum_{k=1}^{j} (\lambda_k - \mu_k)^2 : \boldsymbol{\lambda} \in [0, 1]^K, \lambda_1 \leq \frac{\sum_{k=2,\ldots,j} \lambda_k}{j-1} - G(\beta) \right\} \geq \psi_j, \tag{60}$$

*where* $\psi_j = \frac{j-1}{j}(\mu_1 - \frac{\sum_{k=2}^{j} \mu_k}{j-1})^2$, $\forall j \in \{2, \ldots, K\}$, *as introduced in Section 4.2.*

*Proof.* The Lagrangian of the optimization problem (60) is:

$$\mathcal{L}(\boldsymbol{\lambda}, \eta) = \frac{\beta}{2} \sum_{k=1}^{j} (\lambda_k - \mu_k)^2 + \eta(\lambda_1 - \frac{\sum_{k=2}^{j} \lambda_k}{j-1} + G(\beta)), \text{ for } (\boldsymbol{\lambda}, \eta) \in \mathbb{R}^j \times \mathbb{R}_{\geq 0}.$$

Denote the saddle point of $\mathcal{L}$ by $(\boldsymbol{\lambda}^\star, \eta^\star)$. The KKT conditions are satisfied:

$$\lambda_1^\star \leq \frac{\sum_{k=2}^{j} \lambda_k^\star}{j-1} - G(\beta) \text{ and } \eta^\star \geq 0, \qquad \text{(Feasibility)}$$

$$\beta(\lambda_1^\star - \mu_1) + \eta^\star = 0, \text{ and } \beta(\lambda_k^\star - \mu_k) - \frac{\eta^\star}{j-1} = 0, \forall k \neq 1, \qquad \text{(Stationarity)}$$

$$\eta^\star \left( \lambda_1^\star - \frac{\sum_{k=2}^{j} \lambda_k^\star}{j-1} + G(\beta) \right) = 0. \qquad \text{(Complementarity)}$$

One can simply verify that if $\eta^\star = 0$, stationarity and feasibility cannot hold simultaneously. Thus $\eta^\star > 0$ and complementarity yield that $\lambda_1^\star - \sum_{k=2}^{j} \lambda_k^\star/(j-1) + G(\beta) = 0$. In conjunction with stationarity, we have

$$\eta^\star = \frac{\beta(j-1)}{j} \left( \mu_1 - \frac{\sum_{k=2}^{j} \mu_k}{j-1} + G(\beta) \right),$$

and hence the value of (60) is

$$\frac{(j-1)\beta}{j} \left( \mu_1 - \frac{\sum_{k=2}^{j} \mu_k}{j-1} + G(\beta) \right)^2. \qquad (61)$$

Recall that $G(\beta) = 1/\sqrt{\beta} - 1$ and $\boldsymbol{\mu} \in [0,1]^K$. We deduce that $G(\beta) \geq (1/\sqrt{\beta} - 1)(\mu_1 - \frac{\sum_{k=2}^{j} \mu_k}{j-1})$, which is equivalent to $\mu_1 - \frac{\sum_{k=2}^{j} \mu_k}{j-1} + G(\beta) \geq \frac{1}{\sqrt{\beta}}(\mu_1 - \frac{\sum_{k=2}^{j} \mu_k}{j-1})$ and hence (61) is larger than $\frac{j-1}{j}(\mu_1 - \frac{\sum_{k=2}^{j} \mu_k}{j-1})^2$. $\qquad \square$

As we obtained Corollary 1 in Appendix D.1, combining Proposition 5 and the proof of Proposition 6 yields the following corollary.

**Corollary 2.** $\forall \beta \in (0,1]$, $\boldsymbol{\mu} \in [0,1]^K$ with $\mu_1 > \mu_2 \geq \ldots \geq \mu_K$, $2 \leq j \leq K$, and $J \in \mathcal{J}$, $J \neq [j]$, *one has*

$$\beta \inf \left\{ \sum_{k \in [K]} (\lambda_k - \mu_k)^2 : \boldsymbol{\lambda} \in [0,1]^K, \lambda_1 \leq \frac{\sum_{k \in J, k \neq 1} \lambda_k}{j-1} - G(\beta) \right\} \geq \bar{\psi}_j. \qquad (62)$$

*Proof.* Let $J \in \mathcal{J}$, $J \neq [j]$ be fixed. We denote the indexes in $J$ by $\{\tilde{1}, \tilde{2}, \ldots, \tilde{j}\}$ such that $\tilde{1} < \tilde{2} < \ldots < \tilde{j}$. One can repeat the arguments of the proof of Proposition 6 to obtain that the l.h.s. of (62) is larger than

$$\frac{j-1}{j}(\mu_{\tilde{1}} - \frac{\sum_{k=2}^{j} \mu_{\tilde{k}}}{j-1})^2. \qquad (63)$$

Note that every $J \in \mathcal{J}$ containing 1 satisfies $\tilde{1} = 1$, and that since $J \neq [j]$, we have $\tilde{2} \geq 2, \ldots, \tilde{j} \geq j+1$. As we assume that $\mu_1 > \mu_2 \geq \ldots \geq \mu_K$, Proposition 5 yields that the value of (63) is larger than $\bar{\xi}_j$. $\qquad \square$

# E   LDP for the sampling process under CR

In this section, we are interested in deriving an LDP for the process $\{\boldsymbol{\omega}(\theta T)\}_{T \geq 1}$ for a fixed $\theta \in (0, 1] \cap \mathbb{Q}$ under CR. More precisely, we look for a function $\bar{I}_\theta(\cdot)$ which satisfies an LDP upper bound (3) on $\mathcal{X}_j$ for some fixed $j \in \{1, \ldots, K\}$, where

$$\mathcal{X}_j = \left\{ \boldsymbol{x} \in \Sigma : \exists \sigma : [K] \mapsto [K] \text{ s.t. } x_{\sigma(1)} = \ldots = x_{\sigma(j)} > x_{\sigma(j+1)} > \ldots > x_{\sigma(K)} > 0 \right\}. \quad (64)$$

For convenience, we define $x_{\sigma(K+1)} = 0$. For any $j$, we also define

$$\mathcal{X}_{j,i}(\theta) = \left\{ \boldsymbol{x} \in \mathcal{X}_j : \theta x_{\sigma(i)} i \overline{\log} i > 1 - \theta \sum_{k=i+1}^{K} x_{\sigma(k)} \right\}, \quad \forall i \in \{j, \ldots, K\}, \quad (65)$$

where the permutation $\sigma$ depends on $\boldsymbol{x}$ as in the definition of $\mathcal{X}_j$ (64).

It is important to remark that when $\theta T$ is not an integer, $\boldsymbol{\omega}(\theta T)$ is not defined. Hence in the following, when we write $\varliminf_{T \to \infty} f(\mathbb{P}_{\boldsymbol{\mu}}[\boldsymbol{\omega}(\theta T) \in F])$, we actually mean $\varliminf_{T \to \infty : \theta T \in \mathbb{N}} f(\mathbb{P}_{\boldsymbol{\mu}}[\boldsymbol{\omega}(\theta T) \in F])$.

Deriving an LDP upper bound (3) is not easy in general, and to this aim, we first introduce a useful sufficient condition in E.1.

## E.1   A sufficient condition towards an LDP upper bound (3)

The following condition will be useful in our analysis, in particular in this section. This condition is similar to those presented in Chapter 2 in Varadhan (2016). We say that $\{Y(t)\}_{t \geq 1}$ satisfies an *LDP local upper bound with rate function $I$ at point $\boldsymbol{y} \in \mathcal{Y}$* if:

$$\lim_{\delta \to 0} \varliminf_{t \to \infty} \frac{1}{t} \log \frac{1}{\mathbb{P}[Y(t) \in B(\boldsymbol{y}, \delta)]} \geq I(\boldsymbol{y}), \quad (66)$$

where $B(\boldsymbol{y}, \delta)$ is the open ball with center $\boldsymbol{y}$ and radius $\delta$.

**Lemma 10.** *Suppose $\mathcal{Y}$ is compact and $\{Y(t)\}_{t \geq 1}$ satisfies an LDP local upper bound (66) with a lower semi-continuous function $I$ at all $\boldsymbol{y} \in \mathcal{Y}$, then $\{Y(t)\}_{t \geq 1}$ satisfies an LDP upper bound (3).*

*Proof.* Let $C \subseteq \mathcal{Y}$ be a closed (and hence compact) set, and $s = \inf_{\boldsymbol{y} \in C} I(\boldsymbol{y})$. We prove $\varliminf_{t \to \infty} \frac{1}{t} \log \frac{1}{\mathbb{P}[Y(t) \in C]} \geq s$ if (i) $s = \infty$ and if (ii) $s < \infty$ separately.

**(i) If $s = \infty$.** Let $M > 0$ and $\boldsymbol{y} \in C$. As $I(\boldsymbol{y}) = \infty$, and since $I$ is lower semi-continuous, there exists $\delta_{\boldsymbol{y}} > 0$ s.t.

$$\varliminf_{t \to \infty} \frac{1}{t} \log \frac{1}{\mathbb{P}[Y(t) \in B(\boldsymbol{y}, \delta_{\boldsymbol{y}})]} \geq M. \quad (67)$$

Now observe that $C \subseteq \cup_{\boldsymbol{y} \in C} B(\boldsymbol{y}, \delta_{\boldsymbol{y}})$. The compactness of $C$ implies that we can find $N \in \mathbb{N}$, and $\{\boldsymbol{y}_1, \ldots, \boldsymbol{y}_N\}$ such that $C \subseteq \cup_{i=1}^{N} B(\boldsymbol{y}_i, \delta_{\boldsymbol{y}_i})$, which directly yields that

$$\mathbb{P}[Y(t) \in C] \leq \sum_{i=1}^{N} \mathbb{P}[Y(t) \in B(\boldsymbol{y}_i, \delta_{\boldsymbol{y}_i})] \leq N \max_{i \in [N]} \mathbb{P}[Y(t) \in B(\boldsymbol{y}_i, \delta_{\boldsymbol{y}_i})]. \quad (68)$$

Using a simple rearrangement in (67) and (68), we then have $\varliminf_{t \to \infty} \frac{1}{t} \log \frac{1}{\mathbb{P}[Y(t) \in C]} \geq M$. As $M$ can be taken arbitrarily large, the proof is completed.

**(ii) If $s < \infty$.** Let $\epsilon \in (0, s/2)$ and $\boldsymbol{y} \in C$. As $I(\boldsymbol{y}) \geq s$ and since $I$ is lower semi-continuous, there exists $\delta_{\boldsymbol{y}} > 0$ such that

$$\varliminf_{t \to \infty} \frac{1}{t} \log \frac{1}{\mathbb{P}[Y(t) \in B(\boldsymbol{y}, \delta_{\boldsymbol{y}})]} \geq s - \epsilon. \quad (69)$$

Now observe that $C \subseteq \cup_{\boldsymbol{y} \in C} B(\boldsymbol{y}, \delta_{\boldsymbol{y}})$. The compactness of $C$ implies that we can find $N \in \mathbb{N}$, $\{\boldsymbol{y}_1, \ldots, \boldsymbol{y}_N\}$ such that $C \subseteq \cup_{i=1}^{N} B(\boldsymbol{y}_i, \delta_{\boldsymbol{y}_i})$, which directly yields that

$$\mathbb{P}[Y(t) \in C] \leq \sum_{i=1}^{N} \mathbb{P}[Y(t) \in B(\boldsymbol{y}_i, \delta_{\boldsymbol{y}_i})] \leq N \max_{i \in [N]} \mathbb{P}[Y(t) \in B(\boldsymbol{y}_i, \delta_{\boldsymbol{y}_i})]. \quad (70)$$

Using a simple rearrangement in (69) and (70), we then have $\varliminf_{t\to\infty} \frac{1}{t} \log \frac{1}{\mathbb{P}[Y(t)\in C]} \geq s - \epsilon$. As $\epsilon$ can be taken arbitrarily small, the proof is completed. $\qquad\square$

We apply Lemma 10 to the process $\{\boldsymbol{\omega}(\theta T)\}_{T\geq 1}$. The latter has values in $\Sigma$, a compact set. To derive an LDP upper bound for this process (such an LDP upper bound is required to apply Theorem 1), we just need to establish at all points in $\Sigma$ a local LDP upper bound.

The following two theorems state that $\{\boldsymbol{\omega}(\theta T)\}_{T\geq 1}$ under CR-C and CR-A satisfies a local LDP upper bound.

**Theorem 5.** *[Local LDP upper bound for* CR-C *] For $\theta \in (0,1] \cap \mathbb{Q}$, we define $I_\theta$ as follows.*
*(a) If $\boldsymbol{x} \in \mathcal{X}_1 \cup (\cup_{j=2}^K \cup_{i=j}^K \mathcal{X}_{j,i}(\theta))$, then $I_\theta(\boldsymbol{x}) = \infty$;*
*(b) If $\exists j \in \{2,\dots,K\}$ such that $\boldsymbol{x} \in \mathcal{X}_j \setminus \cup_{j=2}^K \cup_{i=j}^K \mathcal{X}_{j,i}(\theta)$, then*

$$I_\theta(\boldsymbol{x}) = \max_{p=j,\dots,K-1} 2x_{\sigma(p+1)} \inf_{\boldsymbol{\lambda}\in\mathcal{S}_p(\boldsymbol{x})} \sum_{k=1}^{p+1} (\lambda_{\sigma(k)} - \mu_{\sigma(k)})^2,$$

*where $\mathcal{S}_p(\boldsymbol{x})$ is defined in (89);*
*(c) If $\mathcal{V} = \cup_{k=1}^K \mathcal{X}_k$, and $\boldsymbol{x} \in \mathrm{cl}(\mathcal{V}) \setminus \mathcal{V}$, then*

$$I_\theta(\boldsymbol{x}) = \inf\{ \varliminf_{s\to\infty} I_\theta(\boldsymbol{x}^{(s)}) : \{\boldsymbol{x}^{(s)}\}_{s\in\mathbb{N}} \subset \mathcal{V}, \boldsymbol{x}^{(s)} \to \boldsymbol{x} \text{ as } s \to \infty\};$$

*Then the process $\{\boldsymbol{\omega}(\theta T)\}_{T\geq 1}$ under* CR-C *satisfies an LDP upper bound (3) with the rate function $I_\theta$, and $I_\theta$ is lower semi-continuous.*

*Proof.* In view of Lemma 10, the theorem holds if we are able to show that $\{\boldsymbol{\omega}(\theta T)\}_{T\geq 1}$ satisfies a local LDP upper bound with $I_\theta$ and if $I_\theta$ is lower semi-continuous. The first part is established below in Lemma 12, Lemma 13, Lemma 14, and Theorem 7.
For the second part, we first verify the lower semi-continuity of $I_\theta$ restricted to $\cup_{j=1}^K \mathcal{X}_j$, and then apply Lemma 11 with $f = I_\theta$ to establish the lower semi-continuity of $I_\theta$ in $\Sigma$. Let $\boldsymbol{x} \in \cup_{j=1}^K \mathcal{X}_j$. If $\boldsymbol{x} \in \mathcal{X}_1$, lower semi-continuity directly follows from the fact $\mathcal{X}_1$ is open and $I_\theta(\boldsymbol{x}) = \infty$ for $\boldsymbol{x} \in \mathcal{X}_1$. We then consider $\boldsymbol{x} \in \mathcal{X}_j$ for some $j = 2,\dots,K$. By definition, there is $\sigma : [K] \mapsto [K]$ such that $x_{\sigma(1)} = \dots = x_{\sigma(j)} > x_{\sigma(j+1)} > \dots > x_{\sigma(K)}$. By taking $\delta < \min_{i\geq j}\{x_{\sigma(i)} - x_{\sigma(i+1)}\}/2$, we have $\boldsymbol{x}' \in \cup_{q=1}^j \mathcal{X}_q$ if $\|\boldsymbol{x}' - \boldsymbol{x}\|_\infty < \delta$, and $I_\theta(\boldsymbol{x}') \geq \max_{p=j,\dots,K-1} 2x'_{\sigma(p+1)} \inf_{\boldsymbol{\lambda}\in\mathcal{S}_p(\boldsymbol{x}')} \sum_{k=1}^{p+1} (\lambda_{\sigma(k)} - \mu_{\sigma(k)})^2$ as a consequence. Now as verified in Lemma 17 in Appendix F, the mapping $\boldsymbol{x} \mapsto 2x_{\sigma(p+1)} \inf_{\boldsymbol{\lambda}\in\mathcal{S}_p(\boldsymbol{x})} \sum_{k=1}^{p+1} (\lambda_{\sigma(k)} - \mu_{\sigma(k)})^2$ is continuous, we hence deduce that $I_\theta$ is lower semi-continuity at $\boldsymbol{x}$. $\qquad\square$

**Theorem 6.** *[Local LDP upper bound for* CR-A *] For $\theta \in (0,1] \cap \mathbb{Q}$, we define $I_\theta$ as follows.*
*(a) If $\boldsymbol{x} \in \mathcal{X}_1 \cup (\cup_{j=2}^K \cup_{i=j}^K \mathcal{X}_{j,i}(\theta))$, then $I_\theta(\boldsymbol{x}) = \infty$;*
*(b) If $\exists j \in \{2,\dots,K\}$ such that $\boldsymbol{x} \in \mathcal{X}_j \setminus \cup_{j=2}^K \cup_{i=j}^K \mathcal{X}_{j,i}(\theta)$, then*

$$I_\theta(\boldsymbol{x}) = \max_{p=j,\dots,K-1} 2x_{\sigma(p+1)} \inf_{\boldsymbol{\lambda}\in\mathcal{S}_p(\boldsymbol{x})} \sum_{k=1}^{p+1} (\lambda_{\sigma(k)} - \mu_{\sigma(k)})^2,$$

*where $\mathcal{S}_p(\boldsymbol{x})$ is defined in (96);*
*(c) If $\mathcal{V} = \cup_{k=1}^K \mathcal{X}_k$, and $\boldsymbol{x} \in \mathrm{cl}(\mathcal{V}) \setminus \mathcal{V}$, then*

$$I_\theta(\boldsymbol{x}) = \inf\{ \varliminf_{s\to\infty} I_\theta(\boldsymbol{x}^{(s)}) : \{\boldsymbol{x}^{(s)}\}_{s\in\mathbb{N}} \subset \mathcal{V}, \boldsymbol{x}^{(s)} \to \boldsymbol{x} \text{ as } s \to \infty\};$$

*Then the process $\{\boldsymbol{\omega}(\theta T)\}_{T\geq 1}$ under* CR-A *satisfies an LDP upper bound (3) with the rate function $I_\theta$, and $I_\theta$ is lower semi-continuous.*

*Proof.* In view of Lemma 10, the theorem is deduced if we are able to show that $\{\boldsymbol{\omega}(\theta T)\}_{T\geq 1}$ satisfies a local LDP upper bound with $I_\theta$. This is established below in Lemma 12, Lemma 13, Lemma 14, and Theorem 8.

We then verify the lower semi-continuity of $I_\theta$ restricted to $\cup_{j=1}^K \mathcal{X}_j$, and then applying Lemma 11 with $f = I_\theta$ yields the lower semi-continuity of $I_\theta$ in $\Sigma$. Let $\boldsymbol{x} \in \cup_{j=1}^K \mathcal{X}_j$. If $\boldsymbol{x} \in \mathcal{X}_1$, lower semi-continuity directly follows from the fact $\mathcal{X}_1$ is open and $I_\theta(\boldsymbol{x}) = \infty$ for $\boldsymbol{x} \in \mathcal{X}_1$. We then consider $\boldsymbol{x} \in \mathcal{X}_j$ for some $j = 2, \ldots, K$. By definition, there is $\sigma : [K] \mapsto [K]$ such that $x_{\sigma(1)} = \ldots = x_{\sigma(j)} > x_{\sigma(j+1)} > \ldots > x_{\sigma(K)}$. By taking $\delta < \min_{i \geq j}\{x_{\sigma(i)} - x_{\sigma(i+1)}\}/2$, we have $\boldsymbol{x}' \in \cup_{q=1}^j \mathcal{X}_q$ if $\|\boldsymbol{x}' - \boldsymbol{x}\|_\infty < \delta$ and $I_\theta(\boldsymbol{x}') \geq \max_{p=j,\ldots,K-1} 2x'_{\sigma(p+1)} \inf_{\boldsymbol{\lambda} \in \mathcal{S}_p(\boldsymbol{x}')} \sum_{k=1}^{p+1}(\lambda_{\sigma(k)} - \mu_{\sigma(k)})^2$, as a consequence. Now as verified in Lemma 18 in Appendix F, the mapping $\boldsymbol{x} \mapsto 2x_{\sigma(p+1)} \inf_{\boldsymbol{\lambda} \in \mathcal{S}_p(\boldsymbol{x})} \sum_{k=1}^{p+1}(\lambda_{\sigma(k)} - \mu_{\sigma(k)})^2$ is continuous, we hence deduce that $I_\theta$ is lower semi-continuity at $\boldsymbol{x}$.

$\square$

**Lemma 11.** *Suppose $\mathcal{V} \subseteq \Sigma$ is the set such that $\mathrm{cl}(\mathcal{V}) = \Sigma$, and $f : \mathcal{V} \to \mathbb{R}$ is a lower semi-continous function. If we extend $f$ to $\Sigma$ by defining*

$$\bar{f}(\boldsymbol{\omega}) = \begin{cases} f(\boldsymbol{\omega}), & \text{if } \boldsymbol{\omega} \in \mathcal{V}, \\ \inf\{\underline{\lim}_{s \to \infty} f(\boldsymbol{\omega}^{(s)}) : \{\boldsymbol{\omega}^{(s)}\}_{s \in \mathbb{N}} \subset \mathcal{V}, \boldsymbol{\omega}^{(s)} \to \boldsymbol{\omega} \text{ as } s \to \infty\}, & \text{otherwise,} \end{cases}$$

*then $\bar{f}$ is a lower semi-continous function in $\Sigma$.*

*Proof.* By the definition of $\bar{f}$ and the fact $\mathrm{cl}(\mathcal{V}) = \Sigma$,

$$\forall \varepsilon > 0, \forall \delta > 0, \forall \boldsymbol{\omega} \in \Sigma, \exists \boldsymbol{x} \in \mathcal{V} \text{ such that } f(\boldsymbol{x}) < \bar{f}(\boldsymbol{\omega}) + \epsilon \text{ and } \|\boldsymbol{x} - \boldsymbol{\omega}\|_\infty < \delta. \quad (71)$$

Next suppose on the contrary, $\bar{f}$ is not lower semi-continous at some $\boldsymbol{\omega} \in \Sigma$, that is, $\exists \{\boldsymbol{\omega}^{(s)}\} \subset \Sigma$ such that $\boldsymbol{\omega}^{(s)} \to \boldsymbol{\omega}$ as $s \to \infty$ and $\lim_{s \to \infty} \bar{f}(\boldsymbol{\omega}^{(s)}) < \bar{f}(\boldsymbol{\omega})$. Let $\eta = \bar{f}(\boldsymbol{\omega}) - \lim_{s \to \infty} \bar{f}(\boldsymbol{\omega}^{(s)}) > 0$. For each $s \in \mathbb{N}$, (71) implies that there is $\boldsymbol{x}^{(s)} \in \mathcal{V}$ such that $\|\boldsymbol{x}^{(s)} - \boldsymbol{\omega}^{(s)}\|_\infty < 1/s$ and $f(\boldsymbol{x}^{(s)}) < \bar{f}(\boldsymbol{\omega}^{(s)}) + \eta/2$. Hence,

$$\underline{\lim}_{s \to \infty} f(\boldsymbol{x}^{(s)}) \leq \underline{\lim}_{s \to \infty} \bar{f}(\boldsymbol{\omega}^{(s)}) + \frac{\eta}{2} < \bar{f}(\boldsymbol{\omega}),$$

which contradicts the lower semi-continuity of $f$ if $\boldsymbol{\omega} \in \mathcal{V}$ and the definition of $\bar{f}$ if $\boldsymbol{\omega} \notin \mathcal{V}$.

$\square$

### E.2   Local LDP upper bound on $\cup_{i=j}^K \mathcal{X}_{j,i}(\theta)$

Let $\theta \in (0, 1] \cap \mathbb{Q}$ and $j \in \{2, \ldots, K\}$. Here, we first prove the result on $\mathcal{X}_{j,j}(\theta)$ in Lemma 13 and that on $\mathcal{X}_{j,i}(\theta)$ for any $i > j$ in Lemma 14. Note the results in this subsection are valid for both `CR-C` and `CR-A`.

**Lemma 12.** *Let $\theta \in (0, 1] \cap \mathbb{Q}$, $\boldsymbol{x} \in \mathcal{X}_1$, the process $\{\boldsymbol{\omega}(\theta T)\}_{T \geq 1}$ satisfies an LDP local upper bound (66) with $I_\theta(\boldsymbol{x}) = \infty$ at $\boldsymbol{x} \in \mathcal{X}_1$.*

*Proof.* For $\boldsymbol{x} \in \mathcal{X}_1$, there exists $\sigma : [K] \mapsto [K]$ such that $x_{\sigma(1)} > x_{\sigma(2)} > \ldots > x_{\sigma(K)}$. Let $\delta < \min_{k=1,\ldots,K-1}\{x_{\sigma(k)} - x_{\sigma(k+1)}\}$ and $T > \frac{2}{\theta\delta}$ such that $\theta T \in \mathbb{N}$, we show that $\mathbb{P}_{\boldsymbol{\mu}}[\boldsymbol{\omega}(\theta T) \in B(\boldsymbol{x}, \delta)] = 0$, which directly completes the proof.

If $\boldsymbol{\omega}(\theta T) \in B(\boldsymbol{x}, \delta)$, then we have $\omega_{\sigma(1)}(\theta T) > \omega_{\sigma(2)}(\theta T) > \ldots > \omega_{\sigma(K)}(\theta T)$ and

$$\min_{k=1,\ldots,K-1}\{N_{\sigma(k)}(\theta T) - N_{\sigma(k+1)}(\theta T)\} = \theta T \min_{k=1,\ldots,K-1}\{\omega_{\sigma(k)}(\theta T) - \omega_{\sigma(k+1)}(\theta T)\} > 2. \quad (72)$$

Because `CR` always pulls arms in the candidate set in a round-robin manner, (72) will never happen. Hence, $\mathbb{P}_{\boldsymbol{\mu}}[\boldsymbol{\omega}(\theta T) \in B(\boldsymbol{x}, \delta)] = 0$. $\square$

**Lemma 13.** *Let $j \in \{2, \ldots, K\}$. The process $\{\boldsymbol{\omega}(\theta T)\}_{T \geq 1}$ satisfies an LDP local upper bound (66) with $I_\theta(\boldsymbol{x}) = \infty$ at $\boldsymbol{x} \in \mathcal{X}_{j,j}(\theta)$.*

*Proof.* Let $\boldsymbol{x} \in \mathcal{X}_{j,j}(\theta)$. We show that there exists $\delta_{\theta,\boldsymbol{x}} > 0$ and $T_{\theta,\boldsymbol{x}} \in \mathbb{N}$ s.t. if $T \geq T_{\theta,\boldsymbol{x}}$ and $\delta < \delta_{\theta,\boldsymbol{x}}$, then $\boldsymbol{\omega}(\theta T) \notin B(\boldsymbol{x}, \delta)$ almost surely. As a consequence, $\mathbb{P}_{\boldsymbol{\mu}}[\boldsymbol{\omega}(\theta T) \in B(\boldsymbol{x}, \delta)] = 0$, and $I_\theta(\boldsymbol{x}) = \infty$. We decompose the proof into three steps.

*1. Defining $\delta_{\theta,\boldsymbol{x}}$ and $T_{\theta,\boldsymbol{x}}$.* We introduce the two functions, $f_1, f_2 : [0,1] \times \Sigma \mapsto \mathbb{R}$:

$$f_1(\theta', \boldsymbol{x}') = \theta' \sum_{k=1}^{j} x'_{\sigma(k)} \overline{\log} j - \theta' \sum_{k=j+1}^{K} x'_{\sigma(k)} - 1,$$

$$f_2(\theta', \boldsymbol{x}') = \min_{k \leq j} x'_{\sigma(k)} - \max_{k \geq j+1} x'_{\sigma(k)}.$$

Since $\boldsymbol{x} \in \mathcal{X}_{j,j}(\theta)$, we have $c = \min\{f_1(\theta, \boldsymbol{x}), f_2(\theta, \boldsymbol{x})\} > 0$. Leveraging the fact that $f_1, f_2$ are continuous, we can find $\delta_{\theta,\boldsymbol{x}} \in (0, \frac{1}{3j})$ s.t.

$$\text{if } |\theta' - \theta| < 3j\delta_{\theta,\boldsymbol{x}} \text{ and } \|\boldsymbol{x}' - \boldsymbol{x}\|_\infty < 7j\delta_{\theta,\boldsymbol{x}}, \text{ then } \min\{f_1(\theta', \boldsymbol{x}'), f_2(\theta', \boldsymbol{x}')\} > \frac{c}{2}. \quad (73)$$

We also define $T_{\theta,\boldsymbol{x}} = \max\{\lceil \frac{4}{\theta c} \rceil, \lceil \frac{1}{\delta_{\theta,\boldsymbol{x}}} \rceil\}$.

*2. We prove that for $\delta < \delta_{\theta,\boldsymbol{x}}$ and $T > T_{\theta,\boldsymbol{x}}$, $\boldsymbol{\omega}(\theta T) \notin B(\boldsymbol{x}, \delta)$ a.s..* We proceed by contradiction. Assume $\boldsymbol{\omega}(\theta T) \in B(\boldsymbol{x}, \delta)$. From (73), we have $f_2(\theta, \boldsymbol{\omega}(\theta T)) > c/2$. It then directly yields that

$$\min_{k \leq j} N_{\sigma(k)}(\theta T) - \max_{k \geq j+1} N_{\sigma(k)}(\theta T) = \theta T \left[ \min_{k \leq j} \omega_{\sigma(k)}(\theta T) - \max_{k \geq j+1} \omega_{\sigma(k)}(\theta T) \right]$$

$$= \theta T f_2(\theta, \boldsymbol{\omega}(\theta T))$$

$$> \frac{\theta c T}{2} > \frac{\theta c T_{\theta,\boldsymbol{x}}}{2} \geq 2, \quad (74)$$

where the last inequality follows from $T_{\theta,\boldsymbol{x}} > \frac{4}{\theta c}$. Observe that CR always pulls the arms in the candidate set in a round-robin manner (the maximal difference of pulling amounts among the candidate set is at most 1), and CR stops pulling an arm $k$ after $k$ is removed from the candidate set. Thus, from (74), we deduce several facts: (i) $\mathcal{C}_j = \{\sigma(1), \ldots, \sigma(j)\}$; (ii) before the round $\tau = j \min_{k \leq j} N_{\sigma(k)}(\theta T) + \sum_{k > j} N_{\sigma(k)}(\theta T)$, the arm $\ell_j$ to be removed has not yet been decided; (iii)

$$N_{\sigma(k)}(\tau - j) = \begin{cases} \min_{k \leq j} N_{\sigma(k)}(\theta T) - 1, & \text{if } k \leq j, \\ N_{\sigma(k)}(\tau - j) = N_{\sigma(k)}(\theta T), & \text{if } k > j. \end{cases}$$

However, we will show in the next step that in the round $\tau - j$, the condition for discarding an arm in $\mathcal{C}_j$ is triggered. In other words, $\ell_j = \ell(\tau - j)$ is removed from $\mathcal{C}_j$ in the round $\tau - j$, which contradicts (ii).

*3. The condition for discarding an arm in round $\tau - j$ is triggered.* First, from (iii) in Step 2,

$$N_{\sigma(1)}(\tau - j) = N_{\sigma(2)}(\tau - j) = \ldots = N_{\sigma(j)}(\tau - j). \quad (75)$$

Then, using (iii) in Step 2 again yields that

$$\min_{k \leq j} N_{\sigma(k)}(\tau - j) - \max_{k \geq j+1} N_{\sigma(k)}(\tau - j) = \min_{k \leq j} N_{\sigma(k)}(\theta T) - 1 - \max_{k \geq j+1} N_{\sigma(k)}(\theta T) > 1, \quad (76)$$

where the last inequality comes from (74). Finally, observe that

$$\theta - \frac{\tau - j}{T} = \frac{\sum_{k \in [j]} N_{\sigma(k)}(\theta T) - j \min_{k \in [j]} N_{\sigma(k)}(\theta T)}{T} + \frac{j}{T}$$

$$\leq j \left[ \max_{k \in [j]} \omega_{\sigma(k)}(\theta T) - x_{\sigma(j)} + x_{\sigma(j)} - \min_{k \in [j]} \omega_{\sigma(k)}(\theta T) \right] + j\delta_{\theta,\boldsymbol{x}}$$

$$\leq j(2\delta + \delta_{\theta,\boldsymbol{x}}) < 3j\delta_{\theta,\boldsymbol{x}}, \quad (77)$$

where the first inequality is due to $T \geq 1/\delta_{\theta,\boldsymbol{x}}$; the second inequality follows from $\boldsymbol{\omega}(\theta T) \in B(\boldsymbol{x}, \delta)$; the last inequality is obtained using $\delta < \delta_{\theta,\boldsymbol{x}}$. Combining Lemma 15 (see below) with $\delta = 3j\delta_{\theta,\boldsymbol{x}}$ and (77) yields that $\|\boldsymbol{\omega}(\tau - j) - \boldsymbol{\omega}(\theta T)\|_\infty \leq 6j\delta_{\theta,\boldsymbol{x}}$, hence

$$\|\boldsymbol{\omega}(\tau - j) - \boldsymbol{x}\|_\infty \leq \|\boldsymbol{\omega}(\tau - j) - \boldsymbol{\omega}(\theta T)\|_\infty + \|\boldsymbol{\omega}(\theta T) - \boldsymbol{x}\|_\infty \leq 7j\delta_{\theta,\boldsymbol{x}}. \tag{78}$$

From (73)-(77)-(78), we get $f_1(\frac{\tau - j}{T}, \boldsymbol{\omega}(\tau - j)) > \frac{c}{2}$. Thus,

$$G\left(\frac{\sum_{k=1}^{j} N_{\sigma(k)}(\tau - j)\overline{\log}j}{T - \sum_{k=j+1}^{K} N_{\sigma(k)}(\tau - j)}\right) = G\left(\frac{\frac{\tau-j}{T}\sum_{k=1}^{j} \omega_{\sigma(k)}(\tau - j)\overline{\log}j}{1 - \frac{\tau-j}{T}\sum_{k=j+1}^{K} \omega_{\sigma(k)}(\tau - j)}\right) < G(1) = 0, \tag{79}$$

where the inequality is directly derived from $f_1(\frac{\tau - j}{T}, \boldsymbol{\omega}(\tau - j)) > 0$ and the fact that $G(\beta) = 1/\sqrt{\beta} - 1$ is a strictly decreasing function. Note that (75)-(76)-(79) trigger the condition of discarding $\ell(\tau - j)$ in the round $\tau - j$. $\qquad\square$

**Lemma 14.** *Let $j \in \{2, \ldots, K - 1\}$ and $i > j$. The process $\{\boldsymbol{\omega}(\theta T)\}_{T \geq 1}$ satisfies an LDP local upper bound (66) with $I_\theta(\boldsymbol{x}) = \infty$ at $\boldsymbol{x} \in \mathcal{X}_{j,i}(\theta)$.*

*Proof.* Let $\boldsymbol{x} \in \mathcal{X}_{j,i}(\theta)$ and let $\sigma$ be the permutation described in (64) for $\boldsymbol{x}$. We show that there exists $\delta_{\theta,\boldsymbol{x}} > 0$ s.t. for all $\delta < \delta_{\theta,\boldsymbol{x}}$,

$$\lim_{T \to \infty} \frac{1}{T} \log \frac{1}{\mathbb{P}_{\boldsymbol{\mu}}[\boldsymbol{\omega}(\theta T) \in B(\boldsymbol{x}, \delta)]} = \infty. \tag{80}$$

We decompose the proof into three steps.

*1. Defining $\delta_{\theta,\boldsymbol{x}}$, $T_{\boldsymbol{x}}$, and a random time $\tau$.* We introduce two functions: for all $\boldsymbol{x}' \in \Sigma$:

$$f_1(\boldsymbol{x}') = \theta \min_{k=1,\ldots,i} x'_{\sigma(k)} i\overline{\log}i - \theta \sum_{k=i+1}^{K} x'_{\sigma(k)} - 1,$$

$$f_2(\boldsymbol{x}') = \min_{k=1,\ldots,i} x'_{\sigma(k)} - \max_{k=i+1,\ldots,K} x'_{\sigma(k)}.$$

Because $f_1(\boldsymbol{x}) > 0$, $f_2(\boldsymbol{x}) > (x_{\sigma(i)} - x_{\sigma(i+1)})/2$, and both $f_1, f_2$ are continuous, we can find a positive $\delta_{\theta,\boldsymbol{x}} > 0$ s.t.

$$\forall \boldsymbol{x}' \in B(\boldsymbol{x}, \delta_{\theta,\boldsymbol{x}}), \ f_1(\boldsymbol{x}') > 0, \ f_2(\boldsymbol{x}') > \frac{x_{\sigma(i)} - x_{\sigma(i+1)}}{2}. \tag{81}$$

In the following, we fix $\delta < \delta_{\theta,\boldsymbol{x}}$. We define $T_{\boldsymbol{x}}$ as: $T_{\boldsymbol{x}} = \lceil \frac{2}{x_{\sigma(i)} - x_{\sigma(i+1)}} \rceil$. Finally, we introduce the random time $\tau = i\min_{k \leq i} N_{\sigma(k)}(\theta T) + \sum_{k > i} N_{\sigma(k)}(\theta T)$ and two fixed rounds, $\tau_{\min} = \lfloor (ix_{\sigma(i)} + \sum_{k > i} x_{\sigma(k)} - K\delta)T \rfloor$ and $\tau_{\max} = \lceil (ix_{\sigma(i)} + \sum_{k > i} x_{\sigma(k)} + K\delta)T \rceil$.

*2. If $T > T_{\theta,\boldsymbol{x}}$ and $\boldsymbol{\omega}(\theta T) \in B(\boldsymbol{x}, \delta)$, then (i) $\tau \in \{\tau_{\min}, \ldots, \tau_{\max}\}$ and (ii) $\boldsymbol{\omega}(\tau) \in \mathcal{X}_{i,i}(\frac{\tau}{T})$.* (i) is trivial based on the definition of $B(\boldsymbol{x}, \delta)$. To show (ii), we observe that

$$\min_{k \leq i} N_{\sigma(k)}(\theta T) - \max_{k \geq i+1} N_{\sigma(k)}(\theta T) = Tf_2(\boldsymbol{\omega}(\theta T)) > \frac{2}{x_{\sigma(i)} - x_{\sigma(i+1)}} \frac{x_{\sigma(i)} - x_{\sigma(i+1)}}{2} = 1, \tag{82}$$

where the inequality simply comes from (81) and $T > T_{\boldsymbol{x}}$. Since CR always pulls the arms in the candidate set in a round-robin manner (the maximal difference of pulling amounts among the candidate set is at most 1), (82) implies CR discards one arm in $\{\sigma(i + 1), \ldots, \sigma(K)\}$ in the round $\tau$, and $\boldsymbol{\omega}(\tau)$ is:

$$\omega_{\sigma(k)}(\tau) = \begin{cases} \min_{k' \leq i} N_{\sigma(k')}(\theta T)/\tau, & \text{if } k \leq i, \\ N_{\sigma(k)(\theta T)}/\tau, & \text{otherwise.} \end{cases} \tag{83}$$

Note that (83) yields that $\boldsymbol{\omega}(\tau) \in \mathcal{X}_i$. Moreover,

$$\frac{\tau}{T}\omega_{\sigma(i)}(\tau)i\overline{\log}i - \frac{\tau}{T}\sum_{k=i+1}^{K} \omega_{\sigma(k)}(\tau) = \theta \min_{k \leq i} \omega_{\sigma(k)}(\theta T)i\overline{\log}i - \theta \sum_{k=i+1}^{K} \omega_{\sigma(k)}(\theta T)$$

$$= f_1(\boldsymbol{\omega}(\theta T)) + 1 > 1,$$

where the inequality directly follows from (81) and the fact that $\boldsymbol{\omega}(\theta T) \in B(\boldsymbol{x}, \delta_{\theta, \boldsymbol{x}})$. Hence $\boldsymbol{\omega}(\tau) \in \mathcal{X}_{i,i}(\frac{\tau}{T})$.

*3. We show (80).* Thanks to (i) and (ii) in Step 2, we have, for $T > T_{\boldsymbol{x}}$,

$$\mathbb{P}_{\boldsymbol{\mu}}\left[\boldsymbol{\omega}(\theta T) \in \mathcal{B}(\boldsymbol{x}, \delta)\right] \leq \sum_{\tau = \tau_{\min}}^{\tau_{\max}} \mathbb{P}_{\boldsymbol{\mu}}\left[\boldsymbol{\omega}(\tau) \in \mathcal{X}_{i,i}(\frac{\tau}{T})\right] \leq 2K\delta T \max_{\theta' \in (0,1]} \mathbb{P}_{\boldsymbol{\mu}}\left[\boldsymbol{\omega}(\theta'T) \in \mathcal{X}_{i,i}(\theta')\right].$$

Thus, a simple rearrangement of the above inequality yields that

$$\lim_{T\to\infty} \frac{1}{T} \log \frac{1}{\mathbb{P}_{\boldsymbol{\mu}}[\boldsymbol{\omega}(\theta T) \in B(\boldsymbol{x}, \delta)]} \geq \inf_{\theta' \in (0,1]} \lim_{T\to\infty} \frac{1}{T} \log \frac{1}{\mathbb{P}_{\boldsymbol{\mu}}[\boldsymbol{\omega}(\theta'T) \in \mathcal{X}_{i,i}(\theta')]}$$
$$\geq \inf_{\theta' \in (0,1]} \inf_{\boldsymbol{x}' \in \mathcal{X}_{i,i}(\theta')} I_{\theta'}(\boldsymbol{x}') = \infty,$$

where the last inequality stems from Lemma 13. $\qquad\square$

**Lemma 15.** *Let* $\theta \in (0,1] \cap \mathbb{Q}$, *and* $\delta \in (0,1)$. *If we consider* $\theta' \in [\theta - \theta\delta, \theta]$ *and* $T \in \mathbb{N}$ *such that* $\theta T, \theta'T \in \mathbb{N}$, *then*

$$\|\boldsymbol{\omega}(\theta T) - \boldsymbol{\omega}(\theta'T)\|_\infty \leq 2\delta. \tag{84}$$

*Proof.* Observe that for any $k \in [K]$,

$$|\omega_k(\theta T) - \omega_k(\theta'T)| = \left|\frac{N_k(\theta T)}{\theta T} - \frac{N_k(\theta'T)}{\theta'T}\right| \leq \left|\frac{N_k(\theta T)}{\theta T} - \frac{N_k(\theta'T)}{\theta T}\right| + \left|\frac{N_k(\theta'T)}{\theta T} - \frac{N_k(\theta'T)}{\theta'T}\right|$$
$$\leq \frac{\theta - \theta'}{\theta} + \theta'\left(\frac{1}{\theta'} - \frac{1}{\theta}\right)$$
$$= 1 - \frac{\theta'}{\theta} + 1 - \frac{\theta'}{\theta} \leq 2\delta,$$

where the first inequality uses the triangle inequality; the second inequality simply comes from $N_k(\theta T) - N_k(\theta'T) \leq (\theta - \theta')T$ and $N_k(\theta'T) \leq \theta'T$; the third inequality is a consequence of $\theta' \geq \theta(1 - \delta)$. Hence (84) follows. $\qquad\square$

**Lemma 16.** *Let* $\theta \in (0,1] \cap \mathbb{Q}$, *and* $\delta \in (0,1)$. *If we consider* $\theta' \in [\theta - \theta\delta, \theta]$ *and* $T \in \mathbb{N}$ *such that* $\theta T, \theta'T \in \mathbb{N}$, *then*

$$\|\hat{\boldsymbol{\mu}}(\theta T) - \hat{\boldsymbol{\mu}}(\theta'T)\|_\infty \leq 2\delta. \tag{85}$$

*Proof.* Observe that for any $k \in [K]$,

$$|\hat{\mu}_k(\theta T) - \hat{\mu}_k(\theta'T)| = \left|\frac{\sum_{t \leq \theta T} X_k(t)}{\theta T} - \frac{\sum_{t \leq \theta'T} X_k(t)}{\theta'T}\right|$$
$$\leq \left|\frac{\sum_{t \leq \theta T} X_k(t)}{\theta T} - \frac{\sum_{t \leq \theta'T} X_k(t)}{\theta T}\right| + \left|\frac{\sum_{t \leq \theta'T} X_k(t)}{\theta T} - \frac{\sum_{t \leq \theta'T} X_k(t)}{\theta'T}\right|$$
$$\leq \frac{\theta - \theta'}{\theta} + \theta'\left(\frac{1}{\theta'} - \frac{1}{\theta}\right)$$
$$= 1 - \frac{\theta'}{\theta} + 1 - \frac{\theta'}{\theta} \leq 2\delta,$$

where the first inequality uses the triangle inequality; the second inequality simply comes from $\sum_{\theta'T < t \leq \theta T} X_k(t) \leq (\theta - \theta')T$ and $\sum_{t \leq \theta T} X_k(t) \leq \theta'T$ (as Assumption 1 ensures that $X_k(t) \in [0,1]$); the third inequality is a consequence of $\theta' \geq \theta(1 - \delta)$. Hence (85) follows. $\qquad\square$

## E.3 Local LDP upper bound on $\mathcal{X}_j \setminus \cup_{i=j}^K \mathcal{X}_{j,i}(\theta)$

Fix $\theta \in (0,1] \cap \mathbb{Q}$. We will establish local LDP upper bounds on $\mathcal{X}_j \setminus \cup_{i=j}^K \mathcal{X}_{j,i}(\theta)$ for the process $\{\boldsymbol{\omega}(\theta T)\}_{T \geq 1}$ under CR-C and CR-A. The upper bound for CR-C is presented in E.3.1 and that for CR-A in E.3.2. We first present a useful proposition repeatedly used in E.3.1, E.3.2, and the main proofs for Theorem 3 and Theorem 4 in C.

One important property for $\boldsymbol{x} \in \mathcal{X}_j \setminus \cup_{i=j}^K \mathcal{X}_{j,i}(\theta)$ is that the remaining budget for the empirical top-$j$ arms is lower bounded by a constant. This observation is summarized in Proposition 7.

**Proposition 7.** *If $\boldsymbol{x} \in \mathcal{X}_j \setminus \cup_{i=j}^K \mathcal{X}_{j,i}(\theta)$, then $\forall i \in \{j+1, \ldots, K\}$,*

$$1 - \theta \sum_{k=i}^K x_{\sigma(k)} \geq \frac{\overline{\log}(i-1)}{\overline{\log}K}.$$

*Proof.* Notice that the statement of the proposition is equivalent to (86): $\forall i \in \{j+1, \ldots, K\}$,

$$\theta \sum_{k=i}^K x_{\sigma(k)} \leq \frac{1}{\overline{\log}K} \sum_{k=i}^K \frac{1}{k}. \tag{86}$$

The inequalities (86) will be proved by induction.

*1. We show (86) for $i = K$.* Since $\boldsymbol{x} \notin \mathcal{X}_{j,K}(\theta)$,

$$1 \geq \theta K \overline{\log} K x_{\sigma(K)}. \tag{87}$$

Dividing by $K\overline{\log}K$ on both sides of (87) directly yields (86) with $i = K$. Now we assume (86) is valid for some $i+1 \in \{j+2, \ldots, K\}$, and we show (86) for $i$.

*2. We show (86) for $i$ assuming that (86) holds for $i+1$.* As $\boldsymbol{x} \notin \mathcal{X}_{j,i}(\theta)$, we get:

$$1 - \theta \sum_{k=i+1}^K x_{\sigma(k)} \geq \theta i \overline{\log} i\, x_{\sigma(i)}.$$

Dividing the above inequality by $i\overline{\log}i$ and adding $\theta \sum_{k=i+1}^K x_{\sigma(k)}$ to the both sides, we obtain that

$$\theta \sum_{k=i}^K x_{\sigma(k)} \leq \frac{1}{i\overline{\log}i} + (1 - \frac{1}{i\overline{\log}i})\theta \sum_{k=i+1}^K x_{\sigma(k)}$$

$$\leq \frac{1}{i\overline{\log}i} + (1 - \frac{1}{i\overline{\log}i})\frac{1}{\overline{\log}K} \sum_{k=i+1}^K \frac{1}{k}$$

$$= \frac{1}{\overline{\log}K} \sum_{k=i+1}^K \frac{1}{k} + \frac{1}{i\overline{\log}i}\left[1 - \frac{1}{\overline{\log}K} \sum_{k=i+1}^K \frac{1}{k}\right],$$

where the second inequality stems from our inductive hypothesis (86) for $i+1$. As $\overline{\log}K - \overline{\log}i = \sum_{k=i+1}^K \frac{1}{k}$, the r.h.s on the above inequality is equal to

$$\frac{1}{\overline{\log}K} \sum_{k=i+1}^K \frac{1}{k} + \frac{1}{i\overline{\log}i}\left[1 - \frac{1}{\overline{\log}K}\left(\overline{\log}K - \overline{\log}i\right)\right] = \frac{1}{\overline{\log}K} \sum_{k=i}^K \frac{1}{k},$$

hence (86) is proved. $\square$

### E.3.1 The allocation process under CR-C

We show that $I_\theta$ presented in Theorem 7 below is a valid rate function for a local LDP upper bound (66) for the process $\{\boldsymbol{\omega}(\theta T)\}_{T \geq 1}$. This function is however too complicated to use, and we will present a simpler rate function in Corollary 3.

**Theorem 7.** *Let $j \in \{2, \ldots, K-1\}$, $\theta \in (0, 1]$, $\boldsymbol{x} \in \mathcal{X}_j \setminus \cup_{i=j}^K \mathcal{X}_{j,i}(\theta)$, define*

$$I_\theta(\boldsymbol{x}) = \max_{p=j,\ldots,K-1} 2x_{\sigma(p+1)} \inf_{\boldsymbol{\lambda} \in \mathcal{S}_p(\boldsymbol{x})} \sum_{k=1}^{p+1} (\lambda_{\sigma(k)} - \mu_{\sigma(k)})^2, \tag{88}$$

*where $\sigma$ is the permutation described in (64) for $\boldsymbol{z}$ and*

$$\mathcal{S}_p(\boldsymbol{x}) = \left\{ \boldsymbol{\lambda} \in [0, 1]^K : \lambda_{\sigma(p+1)} \leq \min_{k \leq p} \lambda_{\sigma(k)} - G\left(\frac{\theta x_{\sigma(p+1)}(p+1)\overline{\log}(p+1)}{1 - \theta \sum_{k=p+2}^K x_{\sigma(k)}}\right)\right\}. \tag{89}$$

*Then the process $\{\boldsymbol{\omega}(\theta T)\}_{T \geq 1}$ under* CR-C *satisfies a local LDP upper bound (66) with $I_\theta$ at $\boldsymbol{x}$.*

*Proof.* For simplicity, $\sigma$ is assumed to be the identity map. Let $p \in \{j, \ldots, K-1\}$. As $\boldsymbol{x} \in \mathcal{X}_j$, we have $x_p - x_{p+1} > 0$. Let $T > K$, and $\delta < (x_p - x_{p+1})/2$ be some positive number. We will derive an upper bound for $\mathbb{P}_{\boldsymbol{\mu}}[\boldsymbol{\omega}(\theta T) \in B(\boldsymbol{x}, \delta)]$, and then compute its rate by driving $T \to \infty$ and $\delta \to 0$.

Observe that of course:

$$\mathbb{P}_{\boldsymbol{\mu}}[\boldsymbol{\omega}(\theta T) \in B(\boldsymbol{x}, \delta)] \leq \mathbb{P}_{\boldsymbol{\mu}}\left[\cup_{\boldsymbol{y} \in B(\boldsymbol{x}, \delta)}\{\boldsymbol{\omega}(\theta T) = \boldsymbol{y}\}\right]. \tag{90}$$

*1. Obtaining a necessary condition for $\boldsymbol{\omega}(\theta T) = \boldsymbol{y}$.* For any $\boldsymbol{y} \in B(\boldsymbol{x}, \delta)$, we introduce $\theta(\boldsymbol{y})$ and $\boldsymbol{z}(\boldsymbol{y})$ as:

$$\theta(\boldsymbol{y}) = \theta - \theta \sum_{k=1}^p (y_k - y_{p+1}), \text{ and } z_k(\boldsymbol{y}) = \begin{cases} \theta y_k/\theta(\boldsymbol{y}), & \text{if } k \geq p+2, \\ \theta y_{p+1}/\theta(\boldsymbol{y}), & \text{if } k \leq p+1. \end{cases}$$

Following directly from the above definitions, we obtain that

$$\theta(\boldsymbol{y}) \sum_{k=1}^{p+1} z_k(\boldsymbol{y}) = \theta(p+1)y_{p+1}, \text{ and } \theta(\boldsymbol{y})z_k(\boldsymbol{y}) = \theta y_k, \forall k = p+2, \ldots K. \tag{91}$$

From the choice of $\delta$, $y_{p+1}$ is the smallest real number in $\{y_1, \ldots, y_{p+1}\}$, so $\ell_{p+1} = p+1$. Moreover, $\theta(\boldsymbol{y})T$ is the round CR-C decides $\ell_{p+1} = \ell(\theta(\boldsymbol{y})T) = p+1$, and $\boldsymbol{\omega}(\theta(\boldsymbol{y})T) = \boldsymbol{z}(\boldsymbol{y})$. Due to the condition for discarding $p+1$ (see (11)), we have $\hat{\boldsymbol{\mu}}(\theta(\boldsymbol{y})T) \in \mathcal{S}_p(\boldsymbol{y})$. Consequently, we have:

$$\{\boldsymbol{\omega}(\theta T) = \boldsymbol{y}\} \subset \{\hat{\boldsymbol{\mu}}(\theta(\boldsymbol{y})T) \in \mathcal{S}_p(\boldsymbol{y}), \boldsymbol{\omega}(\theta(\boldsymbol{y})T) = \boldsymbol{z}(\boldsymbol{y})\}. \tag{92}$$

*2. Reducing $\cup_{\boldsymbol{y} \in B(\boldsymbol{x}, \delta)}\{\boldsymbol{\omega}(\theta T) = \boldsymbol{y}\}$ to a single set.* To do this reduction, we use the results of Step 1 and Lemmas 15 and 16.

Let $\boldsymbol{y}_0 \in \arg\max_{\boldsymbol{y} \in B(\boldsymbol{x}, \delta)} \theta(\boldsymbol{y})$. Using the continuity of the function $\theta(\boldsymbol{y})$, there exists $\eta(\delta)$ such that $\eta(\delta)$ tends to 0 as $\delta \to 0$, and for all $\boldsymbol{y} \in B(\boldsymbol{x}, \delta)$, $\theta(\boldsymbol{y}) \in [\theta(\boldsymbol{y}_0)(1 - \eta(\delta)), \theta(\boldsymbol{y}_0)]$. By Lemmas 15 and 16, we obtain that:

$$\|\hat{\mu}(\theta(\boldsymbol{y})) - \hat{\mu}(\theta(\boldsymbol{y}_0))\|_\infty \leq 2\eta(\delta),$$
$$\|\boldsymbol{\omega}(\theta(\boldsymbol{y})T) - \boldsymbol{\omega}(\theta(\boldsymbol{y}_0)T)\|_\infty \leq 2\eta(\delta).$$

Now define the following sets:

$$\bar{S}_{\delta,p} = \cup_{\boldsymbol{y} \in B(\boldsymbol{x}, \delta)}\{\bar{s} : \exists s \in \mathcal{S}_p(\boldsymbol{y}) : \|s - \bar{s}\|_\infty \leq 2\eta(\delta)\},$$
$$\bar{W}_\delta = \cup_{\boldsymbol{y} \in B(\boldsymbol{x}, \delta)}\{\bar{w} \in \Sigma : \|\bar{w} - z(\boldsymbol{y})\|_\infty \leq 2\eta(\delta)\}.$$

By construction, for all $\boldsymbol{y} \in B(\boldsymbol{x}, \delta)$, we have that if $\hat{\boldsymbol{\mu}}(\theta(\boldsymbol{y})T) \in \mathcal{S}_p(\boldsymbol{y})$, then $\hat{\boldsymbol{\mu}}(\theta(\boldsymbol{y}_0)T) \in \bar{S}_{\delta,p}$, and similarly $\boldsymbol{\omega}(\theta(\boldsymbol{y})T) = z(\boldsymbol{y})$ implies that $\boldsymbol{\omega}(\theta(\boldsymbol{y}_0)T) \in \bar{W}_\delta$.

*3. Using Theorem 1.* Putting the results of the first two steps together, we get:

$$\mathbb{P}_{\boldsymbol{\mu}}[\boldsymbol{\omega}(\theta T) \in B(\boldsymbol{x}, \delta)] \leq \mathbb{P}_{\boldsymbol{\mu}}\left[\hat{\boldsymbol{\mu}}(\theta(\boldsymbol{y}_0)T) \in \bar{S}_{\delta,p}, \boldsymbol{\omega}(\theta(\boldsymbol{y}_0)T) \in \bar{W}_\delta\right]$$

By applying (c) in Section 3.2 with $\mathcal{S} = \bar{S}_{\delta,p}$ and $W = \bar{W}_\delta$, it follows that

$$\lim_{T \to \infty} \frac{1}{T} \log \frac{1}{\mathbb{P}_{\boldsymbol{\mu}} \left[ \boldsymbol{\omega}(\theta T) \in B(\boldsymbol{x}, \delta) \right]} \geq \theta(\boldsymbol{y}_0) \inf_{\boldsymbol{\omega} \in \bar{W}_\delta} F_{\bar{S}_{\delta,p}}(\boldsymbol{\omega}).$$

When $\delta$ tends to 0, the r.h.s. simply converges to $\theta(\boldsymbol{x}) F_{\mathcal{S}_p(\boldsymbol{x})}(z(\boldsymbol{x}))$. The latter is $\inf_{\boldsymbol{\lambda} \in \mathcal{S}_p(\boldsymbol{x})} 2 \sum_{k=1}^{p+1} x_{p+1}(\lambda_k - \mu_k)^2$ (see Lemma 17 in Appendix F.1 for continuity arguments). As the above proof holds for any $p \in \{j, \ldots, K-1\}$, we complete the proof. $\square$

Next, as in Appendix C, we define the subset of $\mathcal{X}_j \setminus \cup_{i=j}^K \mathcal{X}_{j,i}(\theta)$:

$$\mathcal{Z}_{[j]}(\theta, \beta) = \left\{ \boldsymbol{z} \in \mathcal{X}_j \setminus \cup_{i=j}^K \mathcal{X}_{j,i}(\theta) : \sigma(k) = k, \forall k \leq j \text{ and } \frac{\theta z_j j \overline{\log} j}{1 - \theta \sum_{k>j} z_k} = \beta \right\}$$

for all $\beta \in (0, 1]$. Note that the permutation $\sigma$ in the above definition corresponds to that used for point $\boldsymbol{z}$ as in the definition of $\mathcal{X}_j$ (64): it is such that $z_{\sigma(1)} = \ldots = z_{\sigma(j)} > z_{\sigma(j+1)} > \ldots > z_{\sigma(K)} > 0$.

**Corollary 3.** *Let* $j \in \{2, \ldots, K-1\}$, $\theta, \beta \in (0, 1]$, $\boldsymbol{z} \in \mathcal{Z}_{[j]}(\theta, \beta)$. *Define*

$$\underline{I}_\theta(\boldsymbol{z}) = \frac{\overline{\log}(j+1)}{\theta \overline{\log} K} \left[ \left( (1 + \zeta_j) \sqrt{\frac{\theta z_{\sigma(j+1)}}{1 - \theta \sum_{k=j+2}^K z_{\sigma(k)}}} - \sqrt{\frac{1}{(j+1) \overline{\log}(j+1)}} \right)_+ \right]^2,$$

*where* $\sigma$ *is the permutation described in (64) for* $\boldsymbol{z}$, *then* $\underline{I}_\theta(\boldsymbol{z}) \leq I_\theta(\boldsymbol{z})$.

*Proof.* Observe that $I_\theta(\boldsymbol{z})$ is larger than the value of the following optimization problem:

$$\min_{\boldsymbol{\lambda} \in \mathbb{R}^K} 2 z_{\sigma(j+1)} \left( (\lambda_j - \mu_j)^2 + (\lambda_{\sigma(j+1)} - \mu_{\sigma(j+1)})^2 \right), \tag{93}$$

$$\text{subject to } \lambda_{\sigma(j+1)} \leq \lambda_j - G(\tilde{\beta}),$$

where

$$\tilde{\beta} = \frac{\theta z_{\sigma(j+1)}(j+1) \overline{\log}(j+1)}{1 - \theta \sum_{k=j+2}^K z_{\sigma(k)}}.$$

One can simply verify that the optimal value of (93) is

$$z_{\sigma(j+1)} [(\mu_j - \mu_{\sigma(j+1)} - G(\tilde{\beta}))_+]^2 \geq z_{\sigma(j+1)} [(1 + \zeta_j - \frac{1}{\sqrt{\tilde{\beta}}})_+]^2, \tag{94}$$

where the inequality follows from the fact that $\mu_{j+1} \geq \mu_{\sigma(j+1)}$ and $G(\tilde{\beta}) = 1/\sqrt{\tilde{\beta}} - 1$. Substituting the value of $\tilde{\beta}$ yields that the r.h.s. of (94) is equal to

$$\frac{1}{\theta} \left( 1 - \theta \sum_{k=j+2}^K z_{\sigma(k)} \right) \left[ \left( (1 + \zeta_j) \sqrt{\frac{\theta z_{\sigma(j+1)}}{1 - \theta \sum_{k=j+2}^K z_{\sigma(k)}}} - \sqrt{\frac{1}{(j+1) \overline{\log}(j+1)}} \right)_+ \right]^2.$$

As $\boldsymbol{z} \in \mathcal{X}_j \setminus \cup_{i=j}^K \mathcal{X}_{j,i}(\theta)$, Proposition 7 with $i = j + 2$ directly completes the proof. $\square$

### E.3.2 The allocation process under `CR-A`

One can use similar arguments as those used in the proof of Theorem 7 to derive the analogous rate function for the allocation process under `CR-A`.

**Theorem 8.** *Let* $j \in \{2, \ldots, K-1\}$, $\theta \in (0, 1]$, $\boldsymbol{x} \in \mathcal{X}_j \setminus \cup_{i=j}^K \mathcal{X}_{j,i}(\theta)$, *define*

$$I_\theta(\boldsymbol{x}) = \max_{p=j,\ldots,K-1} 2 x_{\sigma(p+1)} \inf_{\boldsymbol{\lambda} \in \mathcal{S}_p(\boldsymbol{x})} \sum_{k=1}^{p+1} (\lambda_{\sigma(k)} - \mu_{\sigma(k)})^2, \tag{95}$$

*where $\sigma$ is the permutation described in (64) for $z$ and*

$$\mathcal{S}_p(\boldsymbol{x}) = \left\{ \boldsymbol{\lambda} \in [0,1]^K : \lambda_{\sigma(p+1)} \leq \frac{\sum_{k=1}^p \lambda_{\sigma(k)}}{p} - G\left(\frac{\theta x_{\sigma(p+1)}(p+1)\overline{\log}(p+1)}{1 - \theta \sum_{k=p+2}^K x_{\sigma(k)}}\right) \right\}. \quad (96)$$

*Then the process $\{\boldsymbol{\omega}(\theta T)\}_{T \geq 1}$ under* CR-A *satisfies a local LDP upper bound (66) with $\bar{I}_\theta$ at $\boldsymbol{x}$.*

The proof is omitted as it is almost the same as that of Theorem 7. We can also obtain the analogous version of Corollary 3 as shown below:

**Corollary 4.** *Let $j \in \{2, \ldots, K\}$, $\theta, \beta \in (0,1]$, $\boldsymbol{z} \in \mathcal{Z}_{[j]}(\theta, \beta)$, we define*

$$\underline{I}_\theta(\boldsymbol{z}) = \frac{2j\overline{\log}(j+1)}{\theta(j+1)\overline{\log}K} \left[ \left( (1 + \varphi_j)\sqrt{\frac{\theta z_{\sigma(j+1)}}{1 - \theta \sum_{k=j+2}^K z_{\sigma(k)}}} - \sqrt{\frac{1}{(j+1)\overline{\log}(j+1)}} \right)_+ \right]^2, \quad (97)$$

*where $\sigma$ is the permutation described in (64) for $z$, then $\underline{I}_\theta(\boldsymbol{z}) \leq I_\theta(\boldsymbol{z})$.*

*Proof.* We can simply solve the optimization problem similar to (60) as in the proof Proposition 6 in Appendix D.2 and get that $I_\theta(\boldsymbol{z})$ is greater than the l.h.s. of the following inequality.

$$\frac{2j z_{\sigma(j+1)}}{j+1} \left[ \left( \frac{\sum_{k=1}^j \mu_k}{j} - \mu_{\sigma(j+1)} - G(\tilde{\beta}) \right)_+ \right]^2 \geq \frac{2j z_{\sigma(j+1)}}{j+1} \left[ \left( (1 + \varphi_j) - \frac{1}{\sqrt{\tilde{\beta}}} \right)_+ \right]^2, \quad (98)$$

where

$$\tilde{\beta} = \frac{\theta z_{\sigma(j+1)}(j+1)\overline{\log}(j+1)}{1 - \theta \sum_{k=j+2}^K z_{\sigma(k)}}.$$

(98) is due to $\mu_{j+1} \geq \mu_{\sigma(j+1)}$ (hence $\sum_{k=1}^j \mu_k/j - \mu_{\sigma(j+1)} \geq \varphi_j$) and $G(\tilde{\beta}) = \frac{1}{\sqrt{\tilde{\beta}}} - 1$. Substituting the value of $\tilde{\beta}$ yields that the r.h.s. of (98) equals to

$$\frac{2j}{\theta(j+1)} \left( 1 - \theta \sum_{k=j+2}^K z_{\sigma(k)} \right) \left[ \left( (1 + \varphi_j)\sqrt{\frac{\theta z_{\sigma(j+1)}}{1 - \theta \sum_{k=j+2}^K z_{\sigma(k)}}} - \sqrt{\frac{1}{(j+1)\overline{\log}(j+1)}} \right)_+ \right]^2.$$

As $\boldsymbol{z} \in \mathcal{X}_j \setminus \cup_{i=j}^K \mathcal{X}_{j,i}(\theta)$, Proposition 7 with $i = j+2$ directly completes the proof.

$\square$

## F    Continuity arguments

We introduce some definitions and results taken from (Berge, 1997), and also used recently in (Wang et al., 2021; Degenne and Koolen, 2019; Combes et al., 2017) in the bandit literature.

Suppose $\mathbb{X}$ and $\mathbb{Y}$ are Hausdorff topological spaces. Let $u : \mathbb{X} \times \mathbb{Y} \to \mathbb{R}$ be a function and $\Phi : \mathbb{X} \rightrightarrows \mathbb{S}(\mathbb{Y})$ be a set-valued function, where $\mathbb{S}(\mathbb{Y})$ is the set of non-empty subsets of $\mathbb{Y}$. Besides, we introduce $\mathbb{K}(\mathbb{X}) = \{F \in \mathbb{S}(\mathbb{X}) : F \text{ is compact}\}$. We are interested in a minimization problem of the form: for $x \in \mathbb{X}$,

$$v(x) = \inf_{y \in \Phi(x)} u(x, y).$$

We define the set of solutions of this problem as $\Phi^*(x) = \{y \in \Phi(x) \ : \ u(x, y) = v(x)\}$.

**Theorem 9** ((Berge, 1997)). *Assume that*

- $\Phi : \mathbb{X} \rightrightarrows \mathbb{K}(\mathbb{X})$ *is continuous (i.e., both lower and upper hemicontinous),*

- $u : \mathbb{X} \times \mathbb{Y} \to \mathbb{R}$ *is continuous.*

*Then the function $v : \mathbb{X} \to \mathbb{R}$ is continuous and the solution multifunction $\Phi^* : \mathbb{X} \to \mathbb{S}(\mathbb{Y})$ is upper hemicontinuous, with values that are nonempty and compact.*

### F.1    Verifying the continuity in Theorem 7

We verify the continuity argument used in the proof of Theorem 7.

**Lemma 17.** *The function*

$$\inf_{\boldsymbol{\lambda} \in \mathcal{S}_p(\boldsymbol{x})} \sum_{k=1}^{p+1} x_{p+1}(\lambda_k - \mu_k)^2,$$

$$\text{where } \mathcal{S}_p(\boldsymbol{x}) = \left\{ \boldsymbol{\lambda} \in [0,1]^K : \lambda_{p+1} \leq \min_{k \leq p} \lambda_k - G\left(\frac{\theta x_{p+1}(p+1)\overline{\log}(p+1)}{1 - \theta \sum_{k=j+2}^K x_k}\right) \right\},$$

*is continuous for all $\boldsymbol{x} \in \Sigma$.*

*Proof.* We apply Theorem 9 with:

- $\mathbb{X} = \Sigma$,
- $\mathbb{Y} = [0,1]^K$,
- $\Phi(\boldsymbol{x}) = \mathcal{S}_p(\boldsymbol{x})$,
- $u(\boldsymbol{x}, \boldsymbol{\lambda}) = \sum_{k=1}^K x_{p+1}(\lambda_k - \mu_k)^2$.

As the objective function is obviously continuous, it remains to show that $\mathcal{S}_p(\cdot)$ is a continuous correspondence. By simply using Lemma 19 with $f(\boldsymbol{\lambda}) = \lambda_{p+1} - \min_{k \leq p} \lambda_k$ and $g(\boldsymbol{x}) = -G\left(\frac{\theta x_{p+1}(p+1)\overline{\log}(p+1)}{1-\theta \sum_{k=p+2}^K x_k}\right)$, we can complete the proof. $\square$

It is straightforward to get a similar guarantee for the function involved in `CR-A`: we hence omit the proof of the following lemma.

**Lemma 18.** *The function*

$$\inf_{\boldsymbol{\lambda} \in \mathcal{S}_p(\boldsymbol{x})} \sum_{k=1}^{p+1} x_{p+1}(\lambda_k - \mu_k)^2,$$

$$\text{where } \mathcal{S}_p(\boldsymbol{x}) = \left\{ \boldsymbol{\lambda} \in [0,1]^K : \lambda_{p+1} \leq \frac{\sum_{k \leq p} \lambda_k}{p} - G\left(\frac{\theta x_{p+1}(p+1)\overline{\log}(p+1)}{1 - \theta \sum_{k=j+2}^K x_k}\right) \right\},$$

*is continuous for all $\boldsymbol{x} \in \Sigma$.*

**Lemma 19.** *Let $g : \Sigma \mapsto \mathbb{R}$ be a continuous mapping and $f : [0,1]^K \mapsto \mathbb{R}$ be a lower semicontinuous mapping which further satisfies that $\forall \boldsymbol{\lambda} \in \mathbb{R}^K, \delta > 0$, there exists $\boldsymbol{\lambda}' \in \mathbb{R}^K$ s.t. $\left\| \boldsymbol{\lambda} - \boldsymbol{\lambda}' \right\|_\infty \leq \delta$ and $f(\boldsymbol{\lambda}') < f(\boldsymbol{\lambda})$. Then $\forall \boldsymbol{x} \in \Sigma$,*

$$\mathcal{S}(\boldsymbol{x}) = \left\{ \boldsymbol{\lambda} \in [0,1]^K : f(\boldsymbol{\lambda}) \leq g(\boldsymbol{x}) \right\},$$

*is a continuous correspondence.*

*Proof.* (i) *Upper hemicontinuity*: Suppose $\{\boldsymbol{x}_n\}_{n \in \mathbb{N}} \subset \Sigma$ converges to $\boldsymbol{x}^\star \in \Sigma$ and $\{\boldsymbol{\lambda}_n\}_{n \in \mathbb{N}} \subset \mathbb{R}^K$ converges to $\boldsymbol{\lambda}^\star$ s.t. $\boldsymbol{\lambda}_n \in \mathcal{S}(\boldsymbol{x}_n)$ for all $n \in \mathbb{N}$. We will show that $\boldsymbol{\lambda}^\star \in \mathcal{S}(\boldsymbol{x}^\star)$. Since $g$ is upper semicontinuous, and $\boldsymbol{x}_n \to \boldsymbol{x}^\star$ as $n \to \infty$, for any $\epsilon > 0, \exists N_\epsilon \in \mathbb{N}$ s.t. $g(\boldsymbol{x}_n) \leq g(\boldsymbol{x}^\star) + \epsilon$ for all $n \geq N_\epsilon$. As $\boldsymbol{\lambda}_n \in \mathcal{S}(\boldsymbol{x}_n)$, we deduce that

$$f(\boldsymbol{\lambda}_n) \leq g(\boldsymbol{x}_n) \leq g(\boldsymbol{x}^\star) + \epsilon, \ \forall n \geq N_\epsilon.$$

Now the lower semicontinuity of $f$ implies $f(\boldsymbol{\lambda}^\star) \leq \underline{\lim}_{n \to \infty} f(\boldsymbol{\lambda}_n) \leq g(\boldsymbol{x}^\star) + \epsilon$. As $\epsilon$ can be taken arbitrarily, $\boldsymbol{\lambda}^\star \in \mathcal{S}(\boldsymbol{x}^\star)$.

(ii) *Lower hemicontinuity*: Suppose $\{\boldsymbol{x}_n\}_{n \in \mathbb{N}} \subset \Sigma$ converges to $\boldsymbol{x}^\star \in \Sigma$, we aim to show that for all $\boldsymbol{\lambda}^\star \in \mathcal{S}(\boldsymbol{x}^\star)$ (or equivalently $f(\boldsymbol{\lambda}^\star) \leq g(\boldsymbol{x}^\star)$), there exist a subsequence $\{\boldsymbol{x}_{n_k}\}_{k \in \mathbb{N}} \subseteq \{\boldsymbol{x}_n\}_{n \in \mathbb{N}}$ and a sequence $\{\boldsymbol{\lambda}_k\}_{k \in \mathbb{N}}$ s.t. $\boldsymbol{\lambda}_k \in \mathcal{S}(\boldsymbol{x}_{n_k})$ and $\boldsymbol{\lambda}_k \to \boldsymbol{\lambda}^\star$ as $k \to \infty$. By assumption on $f$, for any integer $k$, there exists $\boldsymbol{\lambda}_k$ s.t. $\|\boldsymbol{\lambda}_k - \boldsymbol{\lambda}^\star\|_\infty < 1/k$ and $f(\boldsymbol{\lambda}_k) < f(\boldsymbol{\lambda}^\star)$. Also, $g(\boldsymbol{x}_n) \to g(\boldsymbol{x}^\star)$ as $n \to \infty$ implies there exists a finite $n$ s.t. $g(\boldsymbol{x}_n) \geq f(\boldsymbol{\lambda}_k)$ due to the lower semicontinouity of $g$. Hence we can always find a subsequence $\{n_k\}$ s.t. $g(\boldsymbol{x}_{n_k}) \geq f(\boldsymbol{\lambda}_k)$, which is equivalent to $\boldsymbol{\lambda}_k \in \mathcal{S}(\boldsymbol{x}_{n_k})$. $\qquad\square$

**Lemma 20.** *When $\mathcal{S}$ is a bounded set in $\mathbb{R}^K$, $F_{\mathcal{S}}(\cdot)$ is a continuous function.*

*Proof.* This is a direct application of Berge's maximum theorem. $\qquad\square$

# G A partitioning technique for large deviations

In this section, we establish a theorem that is instrumental in the large deviation analysis of our algorithms.

**Theorem 10.** *Let $\theta_0, \tilde{\theta}_0 \in (0,1)$. Let $(\mathcal{S}_{\theta,\gamma})_{\theta \in [\theta_0,1], \gamma \in [\tilde{\theta}_0,1]}$ and $(W_{\theta,\gamma})_{\theta \in [\theta_0,1], \gamma \in [\tilde{\theta}_0,1]}$ two collections of Borel sets in $[0,1]^K$ and $\Sigma$, respectively. We assume that*

*Suppose (i) for any $T \in \mathbb{N}$,*

$$\mathbb{P}_{\boldsymbol{\mu}}[\mathcal{E}] \leq \sum_{t \geq \theta_0 T}^{T} \sum_{\tau \geq \tilde{\theta}_0 T}^{T} \mathbb{P}_{\boldsymbol{\mu}}[\hat{\boldsymbol{\mu}}(t) \in \mathcal{S}_{\frac{t}{T}, \frac{\tau}{T}}, \boldsymbol{\omega}(t) \in W_{\frac{t}{T}, \frac{\tau}{T}}],$$

*(ii) for any $\theta \in [\theta_0, 1] \cap \mathbb{Q}$, $\{\boldsymbol{\omega}(\theta T)\}_{T \geq 1}$ satisfies LDP upper bound (3) with $I_\theta$, where $I_\theta$ is lower semi-continuous in $\Sigma$.*
*(iii) $\forall \theta, \gamma, \mathcal{S}_{\theta,\gamma} \neq \emptyset$. For all $\delta > 0$, there exists $\eta > 0$ such that if $\max\{|\theta' - \theta|, |\gamma' - \gamma|\} < \eta$, then*

$$\text{dist}(\mathcal{S}_{\theta,\gamma}, \mathcal{S}_{\theta',\gamma'}) = \max\left\{\sup_{s \in \mathcal{S}_{\theta,\gamma}} \inf_{s' \in \mathcal{S}_{\theta',\gamma'}} \|s - s'\|_\infty, \sup_{s' \in \mathcal{S}_{\theta',\gamma'}} \inf_{s \in \mathcal{S}_{\theta,\gamma}} \|s' - s\|_\infty\right\}.$$

*Under Assumption 1, we have*

$$\varliminf_{T \to \infty} \frac{1}{T} \log \frac{1}{\mathbb{P}_{\boldsymbol{\mu}}[\mathcal{E}]} \geq \inf_{\theta \in [\theta_0,1] \cap \mathbb{Q}} \inf_{\gamma \in [\tilde{\theta}_0,1] \cap \mathbb{Q}} \inf_{\boldsymbol{\omega} \in \text{cl}(W_{\theta,\gamma})} \theta \max\{F_{\mathcal{S}_{\theta,\gamma}}(\boldsymbol{\omega}), I_\theta(\boldsymbol{\omega})\}. \tag{99}$$

*Proof.* Without loss of generality, we can assume $\theta_0 = \tilde{\theta}_0$. If $\theta_0 < \tilde{\theta}_0$, we further define $\mathcal{S}_{\theta,\gamma} = \mathcal{S}_{\theta,\tilde{\theta}_0}$ and $W_{\theta,\gamma} = W_{\theta,\tilde{\theta}_0}$ for $\theta_0 \leq \gamma < \tilde{\theta}_0$. And we handle the case $\theta_0 > \tilde{\theta}_0$ similarly.

The main idea behind the proof is to partition the set of instants $t \in \{\theta_0 T, \ldots, T\}$ into a finite collection of instant sets such that if $t$ lies within one of these sets then we may bound $\mathbb{P}_{\boldsymbol{\mu}}[\hat{\boldsymbol{\mu}}(t) \in \mathcal{S}_{\frac{t}{T}, \frac{\tau}{T}}, \boldsymbol{\omega}(t) \in W_{\frac{t}{T}, \frac{\tau}{T}}]$ uniformly. From this partition, we can rewrite the upper bound $\mathbb{P}_{\boldsymbol{\mu}}[\mathcal{E}]$ as a finite sum. This sum is further upper bounded by a maximum over each of its terms. We conclude by taking the limit as $T$ grows large – since we deal with the maximum over a finite number of terms, the limit and the maximum can be exchanged.

Step 1. Partition of $[\theta_0, 1]$. To build this partition, we leverage the results of Lemmas 15 and 16. Let $\delta > 0$. We construct $N_\delta$ points $\theta_1, \ldots, \theta_{N_\delta}$ as follows: $\theta_{N_\delta} = 1$ and

$$\theta_n = (1 - \frac{\delta}{2})^{-n}\theta_0, \qquad \forall n = 1, \ldots, N_\delta - 1.$$

$N_\delta$ is chosen as $\min\{p \in \mathbb{N} : \theta_0(1 - \frac{\delta}{2})^{-p} \geq 1\}$. Now observe by construction that:

$$\cup_{n=1}^{N_\delta} [\theta_{n-1}, \theta_n] = [\theta_0, 1], \tag{100}$$

$$\forall n, (\theta \in [\theta_{n-1}, \theta_n]) \implies (\theta \in [\theta_n(1 - \frac{\delta}{2}), \theta_n]). \tag{101}$$

Step 2. Uniform upper bounds of $\mathbb{P}_{\boldsymbol{\mu}}[\hat{\boldsymbol{\mu}}(t) \in \mathcal{S}_{\frac{t}{T}, \frac{\tau}{T}}, \boldsymbol{\omega}(t) \in W_{\frac{t}{T}, \frac{\tau}{T}}]$. We define the following sets: for all $n, m = 1, \ldots, N_\delta$,

$$\bar{S}_{n,m}^\delta = \cup_{\theta \in [\theta_{n-1}, \theta_n] \cap \mathbb{Q}} \cup_{\gamma \in [\theta_{m-1}, \theta_m] \cap \mathbb{Q}} \{\bar{s} \in [0,1]^K : \exists s \in \mathcal{S}_{\theta,\gamma} : \|\bar{s} - s\|_\infty \leq \delta\},$$

$$\bar{W}_{n,m}^\delta = \cup_{\theta \in [\theta_{n-1}, \theta_n] \cap \mathbb{Q}} \cup_{\gamma \in [\theta_{m-1}, \theta_m] \cap \mathbb{Q}} \{\bar{w} \in \Sigma : \exists w \in W_{\theta,\gamma} : \|\bar{w} - w\|_\infty \leq \delta\}.$$

Let $\theta = t/T, \gamma = \tau/T$, and assume that $\theta \in [\theta_{n-1}, \theta_n], \gamma \in [\theta_{m-1}, \theta_m]$. Then $\hat{\mu}(t) \in \mathcal{S}_{\frac{t}{T}, \frac{\tau}{T}}$ implies that $\hat{\mu}(\theta_n) \in \bar{S}_{n,m}^\delta$ from Lemma 16. Similarly, $\boldsymbol{\omega}(t) \in W_{\frac{t}{T}, \frac{\tau}{T}}$ implies that $\boldsymbol{\omega}(\theta_n) \in \bar{W}_{n,m}^\delta$ from Lemma 15. We conclude that: for all $t, \tau$ such that $t/T \in [\theta_{n-1}T, \theta_n T]$ and $\tau/T \in [\theta_{m-1}T, \theta_m T]$,

$$\mathbb{P}_{\boldsymbol{\mu}}[\hat{\boldsymbol{\mu}}(t) \in \mathcal{S}_{\frac{t}{T}, \frac{\tau}{T}}, \boldsymbol{\omega}(t) \in W_{\frac{t}{T}, \frac{\tau}{T}}] \leq \mathbb{P}_{\boldsymbol{\mu}}[\hat{\boldsymbol{\mu}}(\theta_n T) \in \bar{S}_n^\delta, \boldsymbol{\omega}(\theta_n T) \in \bar{W}_{n,m}^\delta].$$

Step 3. Upper bound on $\mathbb{P}_{\boldsymbol{\mu}}[\mathcal{E}]$. We denote by $p_n$ the number of instants $t$ such that $\frac{t}{T} \in [\theta_{n-1}T, \theta_n T]$. From the above inequality, we conclude that:

$$\mathbb{P}_{\boldsymbol{\mu}}[\mathcal{E}] \leq \sum_{n=1}^{N_\delta} \sum_{m=1}^{N_\delta} p_n p_m \mathbb{P}_{\boldsymbol{\mu}}[\hat{\boldsymbol{\mu}}(\theta_n T) \in \bar{S}_{n,m}^\delta, \boldsymbol{\omega}(\theta_n) \in \bar{W}_{n,m}^\delta],$$

$$\leq (\sum_{n=1}^{N_\delta} p_n)(\sum_{m=1}^{N_\delta} p_m) \max_{n,m \in \{1,\dots,N_\delta\}} \mathbb{P}_{\boldsymbol{\mu}}[\hat{\boldsymbol{\mu}}(\theta_n T) \in \bar{S}_{n,m}^\delta, \boldsymbol{\omega}(\theta_n T) \in \bar{W}_{n,m}^\delta].$$

We note that $\sum_{n=1}^{N_\delta} p_n$ is roughly equal to $T$, and always smaller than $T + 2N_\delta$. Taking the logarithm, dividing by $-T$, and the liminf of the above inequality, we get:

$$\varliminf_{T \to \infty} \frac{1}{T} \log \frac{1}{\mathbb{P}_{\boldsymbol{\mu}}[\mathcal{E}]} \geq \varliminf_{T \to \infty} \frac{1}{T} \min_{n,m \in \{1,\dots,N_\delta\}} \log \frac{1}{\mathbb{P}_{\boldsymbol{\mu}}[\hat{\boldsymbol{\mu}}(\theta_n T) \in \bar{S}_{n,m}^\delta, \boldsymbol{\omega}(\theta_n T) \in \bar{W}_{n,m}^\delta]},$$

$$= \min_{n \in \{1,\dots,N_\delta\}} \varliminf_{T \to \infty} \frac{1}{T} \log \frac{1}{\mathbb{P}_{\boldsymbol{\mu}}[\hat{\boldsymbol{\mu}}(\theta_n T) \in \bar{S}_{n,m}^\delta, \boldsymbol{\omega}(\theta_n T) \in \bar{W}_{n,m}^\delta]},$$

$$\geq \min_{n,m \in \{1,\dots,N_\delta\}} \theta_n \inf_{\boldsymbol{\omega} \in \mathrm{cl}(\bar{W}_{n,m}^\delta)} \max\{F_{\bar{S}_{n,m}^\delta}(\boldsymbol{\omega}), I_{\theta_n}(\boldsymbol{\omega})\},$$

where the last inequality follows from Theorem 1 with $\mathcal{S} = \bar{S}_{n,m}^\delta$ and $W = \bar{W}_{n,m}^\delta$.

Step 4. Continuity arguments. The last step consists in proving that:

$$\limsup_{\delta \to 0} \min_{n,m \in \{1,\dots,N_\delta\}} \inf_{\boldsymbol{\omega} \in \mathrm{cl}(\bar{W}_{n,m}^\delta)} \theta_n \max\{F_{\bar{S}_{n,m}^\delta}(\boldsymbol{\omega}), I_{\theta_n}(\boldsymbol{\omega})\}$$

$$\geq \inf_{\theta,\gamma \in [\theta_0,1] \cap \mathbb{Q}} \inf_{\boldsymbol{\omega} \in \mathrm{cl}(W_{\theta,\gamma})} \theta \max\{F_{\mathcal{S}_{\theta,\gamma}}(\boldsymbol{\omega}), I_\theta(\boldsymbol{\omega})\}.$$

We first state two uniform continuity results, proved in Lemma 21: For any $\epsilon > 0$, $\theta, \gamma \in [\theta_0, 1]$, there exists $\delta > 0$ such that

$$\forall \boldsymbol{\omega}, \forall n, m = 1, \dots, N_\delta, \quad F_{\bar{S}_{n,m}^\delta}(\boldsymbol{\omega}) \geq F_{\mathcal{S}_{\theta_n, \theta_m}}(\boldsymbol{\omega}) - \epsilon, \tag{102}$$

$$\forall \boldsymbol{\omega}, \boldsymbol{\omega}' : \|\boldsymbol{\omega} - \boldsymbol{\omega}'\|_\infty \leq \delta, \quad F_{\mathcal{S}_{\theta,\gamma}}(\boldsymbol{\omega}') \geq F_{\mathcal{S}_{\theta,\gamma}}(\boldsymbol{\omega}) - \epsilon, \quad I_\theta(\boldsymbol{\omega}') \geq I_\theta(\boldsymbol{\omega}) - \epsilon. \tag{103}$$

(102) is the consequence of (iii) and Lemma 21. (103) immediately follows from the compactness of $\Sigma$, lower semi-continuity of $I_\theta$, and $F_{\mathcal{S}_{\theta,\gamma}}$ (see Lemma 20 and (ii)). We fix such a $\delta$, and consider any convergent sequence $(n_k, m_k, \boldsymbol{\omega}_k)_k$ with values in $\{1, \dots, N_\delta\}^2 \times \Sigma$ such that if $n_k = n, m_k = m$ then $\boldsymbol{\omega}_k \in \bar{W}_{n,m}^\delta$. Denote by $(n, m, \boldsymbol{\omega}_0)$ its limit. We let:

$$g^\star = \lim_{k \to \infty} \theta_{n_k} \max(F_{\bar{S}_{n_k,m_k}^\delta}(\boldsymbol{\omega}_k), I_{\theta_{n_k}}(\boldsymbol{\omega}_k)).$$

Then, we have:

$$g^\star \geq \theta_n \max(F_{\bar{S}_{n,m}^\delta}(\boldsymbol{\omega}_0), I_{\theta_n}(\boldsymbol{\omega}_0)) - \epsilon$$

$$\geq \theta_n \max(F_{\mathcal{S}_{\theta_n, \theta_m}}(\boldsymbol{\omega}_0), I_{\theta_n}(\boldsymbol{\omega}_0)) - 2\epsilon$$

$$\geq \theta_n \max(F_{\mathcal{S}_{\theta_n, \theta_m}}(\boldsymbol{\omega}), I_{\theta_n}(\boldsymbol{\omega})) - 3\epsilon,$$

for some $\theta \in [\theta_{n-1}, \theta_n]$, $\gamma \in [\theta_{m-1}, \theta_m]$, $\boldsymbol{\omega} \in \mathrm{cl}(W_{\theta,\gamma})$. The first inequality is due to (103), the second to (102), and the third to the fact that $\omega_0 \in \bar{W}_{n,m}^\delta$ and (103). We conclude that:

$$g^\star \geq \inf_{\theta,\gamma \in [\theta_0,1] \cap \mathbb{Q}} \inf_{\boldsymbol{\omega} \in \mathrm{cl}(W_{\theta,\gamma})} \theta \max\{F_{\mathcal{S}_{\theta,\gamma}}(\boldsymbol{\omega}), I_\theta(\boldsymbol{\omega})\} - 3\epsilon.$$

$\square$

**Lemma 21.** *Assume* $(\mathcal{S}_{\theta,\gamma})_{\theta \in [\theta_0,1], \gamma \in [\tilde{\theta}_0,1]}$ *is a collection of nonempty sets in* $[0,1]^K$ *that satisfies* $\forall \delta > 0$*, there exists* $\eta > 0$ *such that if* $\max\{|\theta' - \theta|, |\gamma' - \gamma|\} < \eta$*, then*

$$\mathrm{dist}(\mathcal{S}_{\theta,\gamma}, \mathcal{S}_{\theta',\gamma'}) = \max \left\{ \sup_{s \in \mathcal{S}_{\theta,\gamma}} \inf_{s' \in \mathcal{S}_{\theta',\gamma'}} \|s - s'\|_\infty, \sup_{s' \in \mathcal{S}_{\theta',\gamma'}} \inf_{s \in \mathcal{S}_{\theta,\gamma}} \|s' - s\|_\infty \right\}.$$

*Then (102) holds.*

*Proof.* Recall that $F_{\mathcal{S}_\theta}(\boldsymbol{\omega}) = \inf_{\boldsymbol{\lambda} \in \mathrm{cl}(\mathcal{S}_\theta)} \Psi(\boldsymbol{\lambda}, \boldsymbol{\omega}) = \inf_{\boldsymbol{\lambda} \in \mathrm{cl}(\mathcal{S}_\theta)} \sum_k \omega_k d(\lambda_k, \mu_k)$. $\Psi$ is uniformly continuous in $[0,1]^K \times \Sigma$, and hence:

$$\forall \epsilon > 0, \exists \bar{\delta} : \forall \boldsymbol{\lambda}, \boldsymbol{\lambda}', \|\boldsymbol{\lambda} - \boldsymbol{\lambda}'\|_\infty \leq 2\bar{\delta} \Rightarrow |\Psi(\boldsymbol{\lambda}, \boldsymbol{\omega}) - \Psi(\boldsymbol{\lambda}', \boldsymbol{\omega})| \leq \epsilon.$$

Based on the assumption in the lemma, there exists $\eta > 0$ such that if $\max\{|\theta' - \theta|, |\gamma' - \gamma|\} < \eta$, then

$$\mathrm{dist}(\mathcal{S}_{\theta,\gamma}, \mathcal{S}_{\theta',\gamma'}) < \bar{\delta}.$$

Select $\delta < \min(\bar{\delta}, 2\eta)$. Let $n, m \in \{1, \ldots, N_\delta\}$. For $(\theta, \gamma) \in [\theta_{n-1}, \theta_n] \times [\theta_{m-1}, \theta_m]$, we have $\max(|\theta - \theta_n|, |\gamma - \theta_m|) \leq \delta/2 < \eta$. This implies that:

$$\mathrm{dist}(\mathcal{S}_{\theta,\gamma}, \mathcal{S}_{\theta_n,\theta_m}) < \bar{\delta}.$$

And hence since $\mathcal{S}_{\theta_n,\theta_m} \subset \bar{S}_{n,m}^\delta$:

$$\mathrm{dist}(\mathcal{S}_{\theta,\gamma}, \bar{S}_{n,m}^\delta) < \bar{\delta}.$$

We conclude that $\forall \boldsymbol{\omega}, \forall \boldsymbol{\lambda} \in \mathcal{S}_{\theta,\gamma}, \exists \boldsymbol{\lambda}' \in \bar{S}_{n,m}^\delta$,

$$\Psi(\boldsymbol{\lambda}, \boldsymbol{\omega}) \geq \Psi(\boldsymbol{\lambda}', \boldsymbol{\omega}) - \epsilon.$$

This completes the proof.

$\square$

# H  A simple min-max problem

The two following results are used in Appendix C.

**Lemma 22.** *Let* $b_1, c_1, b_2, c_2 > 0$. *Introduce* $f(x) = -b_1 x + c_1$ *and* $g(x) = b_2 x + c_2$, $\forall x \in \mathbb{R}$. *Then*

$$\inf_{x \in \mathbb{R}} \max\{f(x), g(x)\} \geq f(x_0),$$

*where* $x_0$ *is the real number s.t.* $x_0 \geq 0$, $f(x_0) = g(x_0)$

*Proof.* As $g$ is an increasing function and $f$ is a decreasing function, we deduce that for all $x \geq x_0$, $\max\{f(x), g(x)\} \geq g(x) \geq g(x_0) = f(x_0)$. Similarly for all $x \leq x_0$, $\max\{f(x), g(x)\} \geq f(x) \geq f(x_0)$.

$\square$

**Lemma 23.** *Let* $b_1, c_1, b_2, c_2 > 0$. *Introduce* $f(x) = -b_1 x + c_1$ *and* $g(x) = [(c_2\sqrt{x} - b_2)_+]^2$ *for* $x \in \mathbb{R}_+$. *Then*

$$\inf_{x \in \mathbb{R}_+} \max\{f(x), g(x)\} \geq f(x_0),$$

*where* $x_0$ *is the unique real number s.t.* $x_0 > 0$ *and* $f(x_0) = g(x_0)$.

*Proof.* We first prove the uniqueness of $x_0$. Observe that $g$ is an increasing function, whereas $f$ is a strictly decreasing function. From the definition, we can have $f(0) = c_1 > 0 = g(0)$, hence there exists an unique point $x_0 > 0$ s.t. $f(x_0) = g(x_0)$. Leveraging the convexity of $g$, there exists a linear function $\underline{g}$ s.t. $g(x) \geq \underline{g}(x)$ and $g(x_0) = \underline{g}(x_0)$. The proof then follows from the fact that

$$\inf_{x \in \mathbb{R}_+} \max\{f(x), g(x)\} \geq \inf_{x \in \mathbb{R}} \max\{f(x), \underline{g}(x)\} \geq f(x_0),$$

where the last inequality is the application of Lemma 22.  $\square$

# I  The LDP conjecture and its consequence

In this section, we restate the conjecture mentioned in Section 3.2, and we discuss how it relates to the conjectured lower bound (1).

**Conjecture 1.** *Assume that under some adaptive sampling algorithm, $\{\boldsymbol{\omega}(t)\}_{t\geq 1}$ satisfies an LDP with rate function L. Then we have: for any non-empty subset $\mathcal{S}$ of $\mathbb{R}^K$ and any subset W of $\Sigma$,*

$$\lim_{t\to\infty} \frac{1}{t} \log \frac{1}{\mathbb{P}_{\boldsymbol{\mu}}\left[\hat{\boldsymbol{\mu}}(t) \in \mathrm{cl}(\mathcal{S}), \boldsymbol{\omega}(t) \in W\right]} = \inf_{\boldsymbol{\omega}\in W} \max\left\{F_{\mathcal{S}}(\boldsymbol{\omega}), L(\boldsymbol{\omega})\right\}.$$

For simplicity, we assume that $\Lambda = \{\boldsymbol{\mu} \in \mathbb{R}^K : \mu_{1(\boldsymbol{\mu})} > \mu_k, \forall k \neq 1(\boldsymbol{\mu})\}$ and all the reward distributions are Gaussian. Introduce the notation:

$$\Psi_{\boldsymbol{\mu}}^{\star} = \max_{\boldsymbol{\omega}\in\Sigma} \inf_{\boldsymbol{\lambda}\in\mathrm{Alt}(\boldsymbol{\mu})} \Psi_{\boldsymbol{\mu}}(\boldsymbol{\lambda}, \boldsymbol{\omega}),$$

$$\text{and } \boldsymbol{\omega}^{\star}(\boldsymbol{\mu}) = \operatorname*{argmax}_{\boldsymbol{\omega}\in\Sigma} \inf_{\boldsymbol{\lambda}\in\mathrm{Alt}(\boldsymbol{\mu})} \Psi_{\boldsymbol{\mu}}(\boldsymbol{\lambda}, \boldsymbol{\omega}),$$

where $\Psi_{\boldsymbol{\mu}}(\boldsymbol{\lambda}, \boldsymbol{\omega}) = \sum_{k=1}^{K} \omega_k d(\lambda_k, \mu_k)$. Notice that the KL-divergence is symmetric in its arguments if the distributions are Gaussian. Hence the conjectured lower bound (1) is exactly the same as that in the fixed confidence setting. The solution $\boldsymbol{\omega}^{\star}(\boldsymbol{\mu})$ to the corresponding optimization problem is unique (see Theorem 5 in Garivier and Kaufmann (2016)).

We consider the set of algorithms returning the best empirical arm $\hat{\imath} = 1(\hat{\boldsymbol{\mu}}(T))$ and with error probability matching the conjectured lower bound (1). If such an algorithm exists, under the assumption that Conjecture 1 is true, the rate function for the corresponding process $\{\boldsymbol{\omega}(T)\}_{T\geq 1}$ must satisfy:

$$\inf_{\boldsymbol{\omega}\in\Sigma} \max\left\{ \left(\inf_{\boldsymbol{\lambda}\in\mathrm{Alt}(\boldsymbol{\mu})} \Psi_{\boldsymbol{\mu}}(\boldsymbol{\lambda}, \boldsymbol{\omega})\right), L_{\boldsymbol{\mu}}(\boldsymbol{\omega}) \right\} \geq \Psi_{\boldsymbol{\mu}}^{\star}, \tag{104}$$

where $L_{\boldsymbol{\mu}}$ is the rate function of a complete LDP under $\boldsymbol{\mu}$ for the process $\{\boldsymbol{\omega}(T)\}_{T\geq 1}$. Lemma 24 below shows that (104) implies the process $\{\boldsymbol{\omega}(T)\}_{T\geq 1}$ convergences to the optimal allocation.

**Lemma 24.** *For $\boldsymbol{\mu} \in \Lambda$, if there is a strategy satisfying (104), then $L_{\boldsymbol{\mu}}(\boldsymbol{\omega}) \geq \Psi_{\boldsymbol{\mu}}^{\star}, \forall \boldsymbol{\omega} \neq \boldsymbol{\omega}^{\star}(\boldsymbol{\mu})$ and $L_{\boldsymbol{\mu}}(\boldsymbol{\omega}^{\star}(\boldsymbol{\mu})) = 0$.*

*Proof.* Assume that, on the contrary, there is $\boldsymbol{\omega}' \neq \boldsymbol{\omega}^{\star}(\boldsymbol{\mu})$ s.t. $L_{\boldsymbol{\mu}}(\boldsymbol{\omega}') < \Psi_{\boldsymbol{\mu}}^{\star}$. Together with $\inf_{\boldsymbol{\lambda}\in\mathrm{Alt}(\boldsymbol{\mu})} \Psi_{\boldsymbol{\mu}}(\boldsymbol{\lambda}, \boldsymbol{\omega}') < \Psi_{\boldsymbol{\mu}}^{\star}$, this implies that:

$$\inf_{\boldsymbol{\omega}\in\Sigma} \max\left\{ \left(\inf_{\boldsymbol{\lambda}\in\mathrm{Alt}(\boldsymbol{\mu})} \Psi_{\boldsymbol{\mu}}(\boldsymbol{\lambda}, \boldsymbol{\omega})\right), L_{\boldsymbol{\mu}}(\boldsymbol{\omega}) \right\} \leq \max\left\{ \inf_{\boldsymbol{\lambda}\in\mathrm{Alt}(\boldsymbol{\mu})} \Psi_{\boldsymbol{\mu}}(\boldsymbol{\lambda}, \boldsymbol{\omega}'), L_{\boldsymbol{\mu}}(\boldsymbol{\omega}') \right\} < \Psi_{\boldsymbol{\mu}}^{\star}.$$

This contradicts the assumption, (104), so we have $L_{\boldsymbol{\mu}}(\boldsymbol{\omega}) \geq \Psi_{\boldsymbol{\mu}}^{\star}, \forall \boldsymbol{\omega} \neq \boldsymbol{\omega}^{\star}(\boldsymbol{\mu})$. As for the optimal allocation, $\boldsymbol{\omega}^{\star}(\boldsymbol{\mu})$, the fact $\mathbb{P}_{\boldsymbol{\mu}}\left[\boldsymbol{\omega}(T) \in \Sigma\right] = 1$ implies that

$$0 = \lim_{T\to\infty} \frac{1}{T} \log \frac{1}{\mathbb{P}_{\boldsymbol{\mu}}\left[\boldsymbol{\omega}(T) \in \Sigma\right]} \geq \inf_{\boldsymbol{\omega}\in\Sigma} L_{\boldsymbol{\mu}}(\boldsymbol{\omega}),$$

where the last inequality stems from (4) in Definition 1. Since $L_{\boldsymbol{\mu}}(\boldsymbol{\omega}) \geq \Psi_{\boldsymbol{\mu}}^{\star}, \forall \boldsymbol{\omega} \neq \boldsymbol{\omega}^{\star}(\boldsymbol{\mu})$, we conclude that $L_{\boldsymbol{\mu}}(\boldsymbol{\omega}^{\star}(\boldsymbol{\mu})) = 0$. $\square$

So far, we have investigated the consequence of matching the lower bound (1) on a single instance (a single parameter $\boldsymbol{\mu}$). Of course, we wish to get an algorithm matching (1) for all instances. The following theorem shows that this is impossible even for two parameters.

**Theorem 11.** *Consider $\boldsymbol{\mu}, \boldsymbol{\pi} \in \Lambda$ s.t. $\boldsymbol{\omega}^{\star}(\boldsymbol{\mu}) \neq \boldsymbol{\omega}^{\star}(\boldsymbol{\pi})$ and $\max_{k\in[K]} d(\pi_k, \mu_k) < \Psi_{\boldsymbol{\mu}}^{\star}$, then there is no strategy satisfying (i) and (ii) simultaneously:*

*(i) (104) holds for $\boldsymbol{\pi}$*

*(ii) (104) holds for $\boldsymbol{\mu}$*

*Proof.* Assume that, on the contrary, there is such a strategy. By the assumption on $\boldsymbol{\mu}$ and $\boldsymbol{\pi}$, there will be an open set $\mathcal{O} \subset \Sigma$ s.t. $\boldsymbol{\omega}^\star(\boldsymbol{\pi}) \in \mathcal{O}$ but $\boldsymbol{\omega}^\star(\boldsymbol{\mu}) \notin \mathcal{O}$. On the one hand, $L_{\boldsymbol{\pi}}(\boldsymbol{\omega}^\star(\boldsymbol{\pi})) = 0$ by Lemma 24 and (i). Recalling the LDP lower bound (4) in Definition 1, $L_{\boldsymbol{\pi}}(\boldsymbol{\omega}^\star(\boldsymbol{\pi})) = 0$ directly implies that:

$$\lim_{T \to \infty} \mathbb{P}_{\boldsymbol{\pi}}[\boldsymbol{\omega}(T) \in \mathcal{O}] = 1. \tag{105}$$

On the other hand, (ii) and Lemma 24 imply that $L_{\boldsymbol{\mu}}(\boldsymbol{\omega}) \geq \Psi_{\boldsymbol{\mu}}^\star$ if $\boldsymbol{\omega} \neq \boldsymbol{\omega}^\star(\boldsymbol{\mu})$, hence

$$\varliminf_{T \to \infty} \frac{1}{T} \log \frac{1}{\mathbb{P}_{\boldsymbol{\mu}}[\boldsymbol{\omega}(T) \in \mathcal{O}]} \geq \Psi_{\boldsymbol{\mu}}^\star. \tag{106}$$

Now applying a change-of-measure argument (see Lemma 1 in Kaufmann et al. (2016) or equation (6) in Garivier et al. (2019)), one can derive

$$\sum_{k=1}^{K} \mathbb{E}_{\boldsymbol{\pi}}\left[\omega_k(T)\right] d(\pi_k, \mu_k) \geq \frac{1}{T} \mathrm{kl}(\mathbb{P}_{\boldsymbol{\pi}}[\boldsymbol{\omega}(T) \in \mathcal{O}], \mathbb{P}_{\boldsymbol{\mu}}[\boldsymbol{\omega}(T) \in \mathcal{O}]) \tag{107}$$

Using the assumption that $\max_{k \in [K]} d(\pi_k, \mu_k) < \Psi_{\boldsymbol{\mu}}^\star$, the left-hand side of (107) is strictly smaller $\Psi_{\boldsymbol{\mu}}^\star$. However, by letting $T \to \infty$ on the r.h.s. of (107), (105) and (106) implies the limitinf is larger than $\Psi_{\boldsymbol{\mu}}^\star$. This is a contradiction. $\qquad\square$

The consequence of Theorem 11 is that either our conjecture is true and in which case, for any algorithm there are two instances for which it cannot match the error lower bound (1) or the conjecture is not true (the bound provided in Theorem 1 is not tight). We finally note that recent results presented in Degenne (2023); Wang et al. (2023) suggest that indeed the lower bound (1) cannot be achieved.

## J Discussion on the conjectured lower bound (1)

In this section, we discuss two points: (i) (1) indeed corresponds to the conjectured lower bound proposed by Garivier and Kaufmann (2016), see their Section 7; (ii) however, as far as we know, there is no proof for (1), but one can derive a lower bound by inverting $\max$ and $\inf$ in (1).

(i) Without loss of generality, assume that $\mu$ is such that 1 is the best arm. We start from (1) and show that this is equivalent to Garivier-Kaufmann's formula. First, it can be easily checked that in (1), we can replace $\Sigma$ by $\Sigma_{>0} = \{\omega \in \Sigma : \omega_k > 0, \forall k \in [K]\}$. Then, for any $\omega \in \Sigma_{>0}$, we have

$$\inf_{\lambda \in \text{Alt}(\mu)} \sum_k \omega_k d(\lambda_k, \mu_k) = \min_{m \neq 1} \inf_{\mu_m < x < \mu_1} \omega_1 d(x, \mu_1) + \omega_m d(x, \mu_m).$$

Indeed, we can decompose $\text{Alt}(\mu)$ as $\cup_{m \neq 1} \{\lambda \in \Lambda : \lambda_m > \lambda_1\}$, and thus, we have:

$$\inf_{\lambda \in \text{Alt}(\mu)} \sum_k \omega_k d(\lambda_k, \mu_k) = \min_{m \neq 1} \inf_{\lambda_m > \lambda_1} \sum_k \omega_k d(\lambda_k, \mu_k).$$

We conclude by observing that

$$\inf_{\lambda_m > \lambda_1} \omega_1 d(\lambda_1, \mu_1) + \omega_m d(\lambda_m, \mu_m) = \inf_{\mu_m < x < \mu_1} \omega_1 d(x, \mu_1) + \omega_k d(x, \mu_m),$$

which holds for all families of distributions such that $x \mapsto d(x, y)$ is monotonic (decreasing before $y$ and increasing after $y$) – this holds for Bernoulli, Gaussian, etc.

(ii) Consider a consistent algorithm, and denote by $\omega_k(\lambda)$ the expected proportion of rounds where the algorithm selects arm $k$ under the probability $\mathbb{P}_\lambda$. Using the classical change-of-measure arguments, we get:

$$\limsup_{T \to \infty} \log \frac{1}{\mathbb{P}_\mu[\hat\imath \neq 1]} \leq T \sum_k \omega_k(\lambda) d(\lambda_k, \mu_k) \leq T \max_{\omega \in \Sigma} \sum_k \omega_k d(\lambda_k, \mu_k).$$

We can only deduce that:

$$\limsup_{T \to \infty} \frac{1}{T} \log \frac{1}{\mathbb{P}_\mu[\hat\imath \neq 1]} \leq \inf_{\lambda \in \text{Alt}(\mu)} \max_{\omega \in \Sigma} \sum_k \omega_k d(\lambda_k, \mu_k).$$

One cannot directly apply Sion's minimax theorem to derive (1) (as $\text{Alt}(\mu)$ is not a convex domain).

## K Numerical experiments

We consider various problem instances to numerically evaluate the performance of `CR`. In these instances, we vary the number of arms from 5 to 55; we use Bernoulli distributed rewards, and vary the shape of the arm-to-reward mapping. For each instance, we compare `CR-C` and `CR-A` to `SR` (Audibert et al., 2010), `SH` (Karnin et al., 2013), and `UGapE` (Gabillon et al., 2012). As discussed in Section 2, `UGapE` requires prior knowledge about a parameter depending on the underlying problem. We hence implement its heuristic version which estimates the parameter on the fly, such modification was suggested in previous works e.g. (Audibert et al., 2010; Karnin et al., 2013). We implement all algorithms in `Julia 1.7.3` and run all experiments on a machine with Apple M1 with 16 GB RAM.[3] The error probabilities averaged over $40,000$ independent runs. In all experiments, we set $\theta_0 = 10^{-5}$ for `CR`.

We vary the shape of the arm-to-reward function and consider one shape in each of the subsections below. We present the error probability of all algorithms in tables and figures. In the latter, the error probability is presented using the log scale, which sometimes makes the curves for some algorithms close to each other. In the tables, we present the error probability for a few budgets only, and there, we can see a clearer separation between the performance of the various algorithms.

Observe that our algorithms, `CR`, perform better than the other algorithms for most arm-to-reward function shapes, except for linear arm-to-reward functions. In this specific scenario, `SH` performs better but may perform very poorly in some other setups. In contrasts, `CR` is always among the two best algorithms in all scenarios.

### K.1 One group of suboptimal arms

This instance is considered by (Karnin et al., 2013): $\mu_1 = 0.5$ and $\mu_k = 0.45$ for all $k \geq 2$. We can see that the performances of `SR`, `CR-C`, and `CR-A` are significantly better than `UGapE` and `SH`.

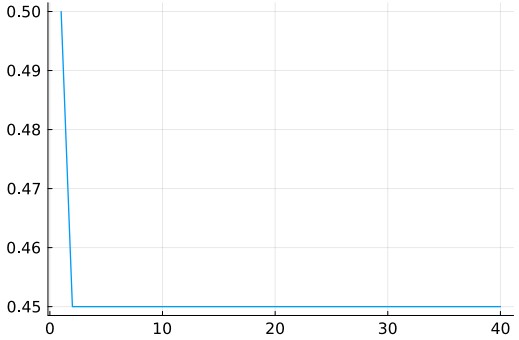

Figure 1: (One group of suboptimal arms) $\boldsymbol{\mu}$ with $K = 40$.

---

[3]Our Julia implementation can be found at https://github.com/rctzeng/NeurIPS2023-CR.

Table 2: (One group of suboptimal arms) error probabilities (in %).

| $K = 10$ | | | $T = 6,400$ | $T = 7,200$ | $T = 8,000$ |
|---|---|---|---|---|---|
| `UGapE` | (Gabillon et al., 2012) | | 34.66 | 33.22 | 32.15 |
| `SH` | Karnin et al. (2013) | | 16.09 | 13.83 | 11.72 |
| `SR` | Audibert et al. (2010) | | 7.86 | 5.86 | 4.29 |
| `CR-C` | (this paper) | | **7.29** | **5.47** | 4.17 |
| `CR-A` | (this paper) | | 7.37 | 5.52 | **4.07** |
| $K = 20$ | | | $T = 12,000$ | $T = 14,000$ | $T = 16,000$ |
| `UGapE` | (Gabillon et al., 2012) | | 42.77 | 39.93 | 38.62 |
| `SH` | Karnin et al. (2013) | | 24.48 | 20.30 | 17.02 |
| `SR` | Audibert et al. (2010) | | 9.43 | 6.59 | 4.35 |
| `CR-C` | (this paper) | | **8.39** | **5.92** | **4.08** |
| `CR-A` | (this paper) | | 8.80 | 6.16 | 4.42 |
| $K = 40$ | | | $T = 30,000$ | $T = 35,000$ | $T = 40,000$ |
| `UGapE` | (Gabillon et al., 2012) | | 42.84 | 40.46 | 38.11 |
| `SH` | Karnin et al. (2013) | | 23.24 | 19.20 | 15.93 |
| `SR` | Audibert et al. (2010) | | 6.51 | 4.48 | 3.14 |
| `CR-C` | (this paper) | | **5.96** | **3.99** | **2.89** |
| `CR-A` | (this paper) | | 6.56 | 4.27 | 3.06 |

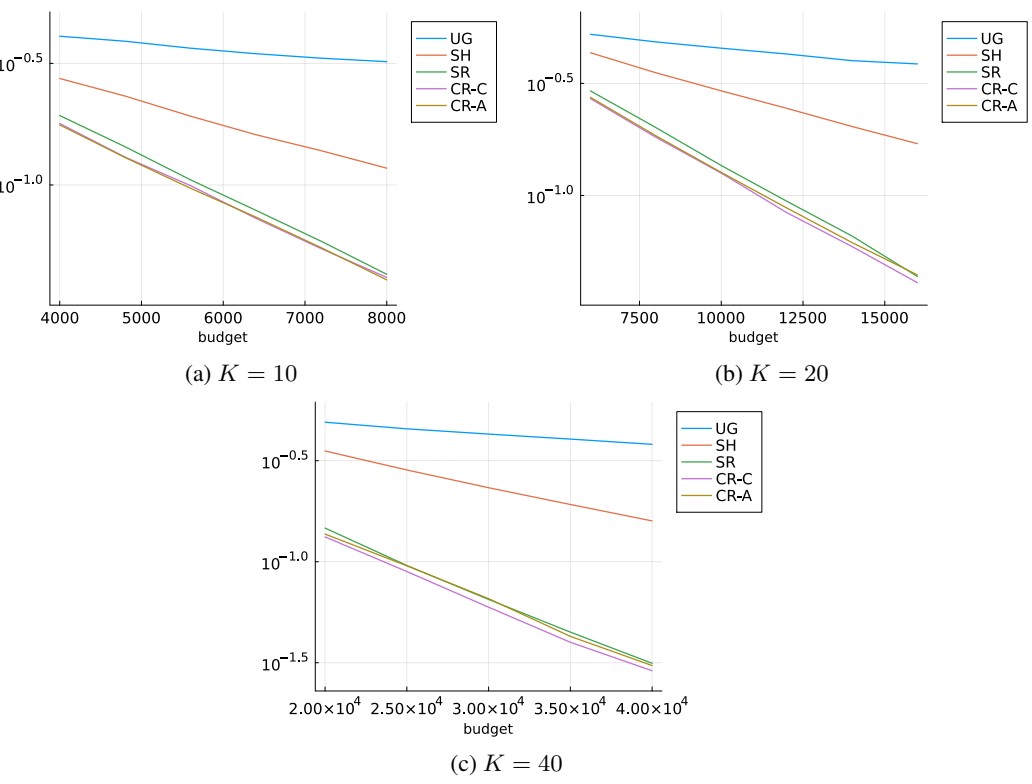

(a) $K = 10$

(b) $K = 20$

(c) $K = 40$

Figure 2: (One group of suboptimal arms) error probabilities averaged over $40,000$ independent runs.

## K.2 Two groups of suboptimal arms

In this instance, we set $\mu_1 = 0.5$, $\mu_k = 0.45$ for $k = 2, \cdots, \lfloor \frac{K-1}{2} \rfloor$, and $\mu_k = 0.4$ for $k = \lfloor \frac{K-1}{2} \rfloor + 1, \cdots, K$. Compared to K.1, where `CR-C` is always the best, `CR-A` becomes relatively better here.

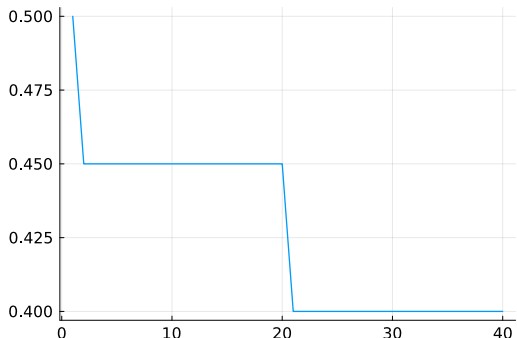

Figure 3: (Two groups of suboptimal arms) $\boldsymbol{\mu}$ with $K = 40$.

Table 3: (Two groups of suboptimal arms) error probabilities (in %).

| $K = 10$ | | | $T = 5,600$ | $T = 6,800$ | $T = 8,000$ |
|---|---|---|---|---|---|
| UGapE | (Gabillon et al., 2012) | | 25.50 | 23.02 | 20.97 |
| SH | Karnin et al. (2013) | | 7.41 | 5.05 | 3.33 |
| SR | Audibert et al. (2010) | | 4.05 | 2.30 | 1.20 |
| CR-C | (this paper) | | 3.68 | 2.05 | 1.13 |
| CR-A | (this paper) | | **3.44** | **1.88** | **0.99** |
| $K = 20$ | | | $T = 4,000$ | $T = 7,000$ | $T = 10,000$ |
| UGapE | (Gabillon et al., 2012) | | 48.49 | 40.68 | 36.12 |
| SH | Karnin et al. (2013) | | 38.64 | 23.13 | 14.64 |
| SR | Audibert et al. (2010) | | 28.26 | 12.03 | 5.03 |
| CR-C | (this paper) | | 26.54 | 10.62 | 4.52 |
| CR-A | (this paper) | | **26.00** | **10.61** | **4.27** |
| $K = 40$ | | | $T = 15,000$ | $T = 20,000$ | $T = 25,000$ |
| UGapE | (Gabillon et al., 2012) | | 46.04 | 41.80 | 38.30 |
| SH | Karnin et al. (2013) | | 27.29 | 19.69 | 14.24 |
| SR | Audibert et al. (2010) | | 9.97 | 5.03 | 2.43 |
| CR-C | (this paper) | | **9.12** | 4.49 | **2.24** |
| CR-A | (this paper) | | 9.26 | **4.78** | 2.38 |

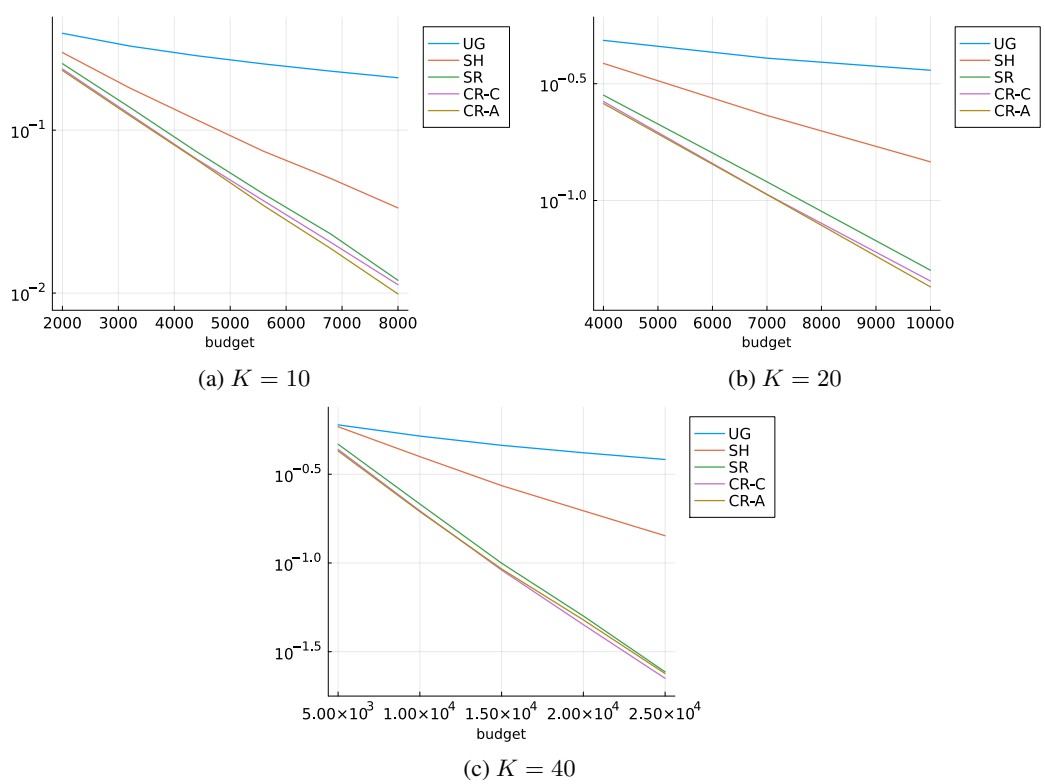

(a) $K = 10$

(b) $K = 20$

(c) $K = 40$

Figure 4: (Two groups of suboptimal arms) error probabilities averaged over $40,000$ independent runs.

## K.3 Linear arm-to-reward function

In this instance, we set $\mu_k = \frac{3}{4} - \frac{k-1}{2K}$ for $k = 1, \cdots, K$. As commented at the beginning of this section, SH is the best, but the performance of CR-A is rather close to it.

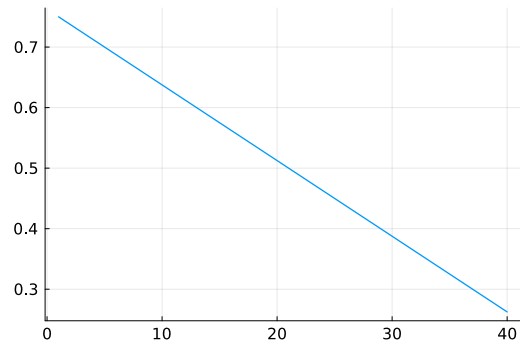

Figure 5: (Linear arm-to-reward function) $\mu$ with $K = 40$.

Table 4: (Linear arm-to-reward function) error probability (in %).

| $K = 10$ | | | $T = 3,200$ | $T = 3,600$ | $T = 4,000$ |
|---|---|---|---|---|---|
| UGapE | (Gabillon et al., 2012) | | 4.96 | 4.22 | 3.24 |
| SH | Karnin et al. (2013) | | **0.80** | **0.48** | **0.30** |
| SR | Audibert et al. (2010) | | 2.09 | 1.53 | 1.03 |
| CR-C | (this paper) | | 1.59 | 1.04 | 0.83 |
| CR-A | (this paper) | | 1.20 | 0.81 | 0.55 |
| $K = 20$ | | | $T = 6,000$ | $T = 8,000$ | $T = 10,000$ |
| UGapE | (Gabillon et al., 2012) | | 15.49 | 11.20 | 8.78 |
| SH | Karnin et al. (2013) | | **6.87** | **3.86** | **2.12** |
| SR | Audibert et al. (2010) | | 10.76 | 7.57 | 5.24 |
| CR-C | (this paper) | | 10.03 | 6.96 | 4.68 |
| CR-A | (this paper) | | 8.78 | 5.72 | 3.73 |
| $K = 40$ | | | $T = 15,000$ | $T = 20,000$ | $T = 25,000$ |
| UGapE | (Gabillon et al., 2012) | | 25.09 | 20.21 | 16.44 |
| SH | Karnin et al. (2013) | | **16.01** | **11.14** | **7.80** |
| SR | Audibert et al. (2010) | | 20.29 | 15.86 | 13.07 |
| CR-C | (this paper) | | 19.93 | 15.56 | 12.30 |
| CR-A | (this paper) | | 17.99 | 13.74 | 10.67 |

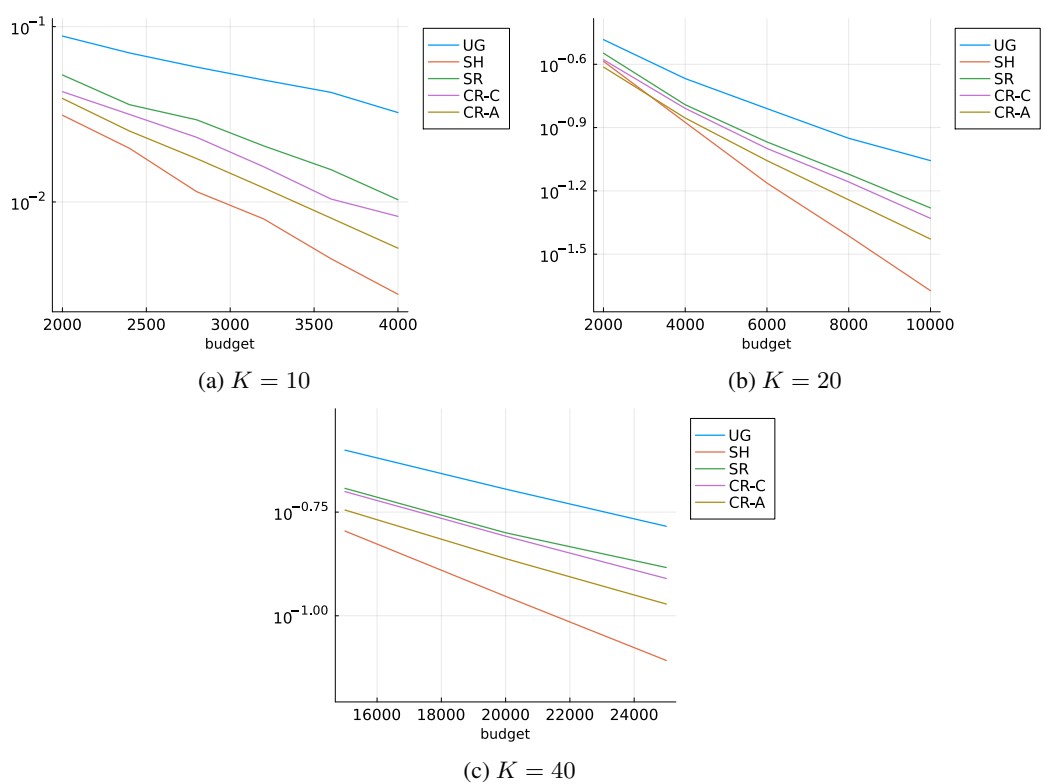

Figure 6: (Linear arm-to-reward function) error probabilities averaged over $40,000$ independent runs.

### K.4 Concave arm-to-reward function

In this instance, we set $\mu_1 = \sin(\frac{(K-1)\pi}{2K})$ and $\mu_k = \sin(\frac{9\pi(K-k+1)}{20K})$ for $k = 2, \cdots, K$. When $K$ is small, the arm-to-reward function is close to the linear shape, and hence SH does well. while $K$ becomes larger, CR-A becomes the best.

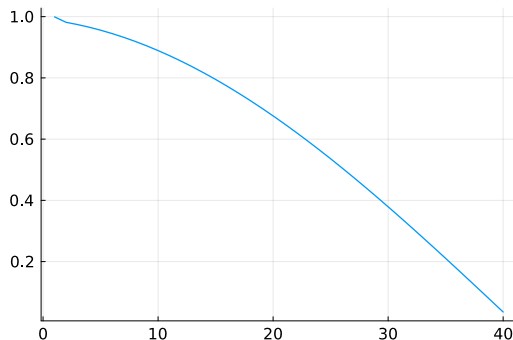

Figure 7: (Concave arm-to-reward function) $\boldsymbol{\mu}$ with $K = 40$.

Table 5: (Concave arm-to-reward function) error probability (in %).

| $K = 10$ | | | $T = 900$ | $T = 1,400$ | $T = 1,900$ |
|---|---|---|---|---|---|
| UGapE | (Gabillon et al., 2012) | | 2.23 | 1.36 | 0.94 |
| SH | Karnin et al. (2013) | | 1.10 | **0.17** | **0.04** |
| SR | Audibert et al. (2010) | | 1.85 | 0.54 | 0.20 |
| CR-C | (this paper) | | 1.24 | 0.35 | 0.10 |
| CR-A | (this paper) | | **0.94** | 0.21 | **0.04** |
| $K = 20$ | | | $T = 900$ | $T = 1,400$ | $T = 1,900$ |
| UGapE | (Gabillon et al., 2012) | | 2.44 | 1.85 | 1.59 |
| SH | Karnin et al. (2013) | | 2.66 | 0.51 | 0.13 |
| SR | Audibert et al. (2010) | | 2.81 | 0.86 | 0.31 |
| CR-C | (this paper) | | 1.87 | 0.47 | 0.14 |
| CR-A | (this paper) | | **1.36** | **0.36** | **0.09** |
| $K = 40$ | | | $T = 2,400$ | $T = 2,800$ | $T = 3,200$ |
| UGapE | (Gabillon et al., 2012) | | 1.03 | 0.98 | 0.94 |
| SH | Karnin et al. (2013) | | 0.75 | 0.34 | 0.15 |
| SR | Audibert et al. (2010) | | 0.23 | 0.10 | 0.02 |
| CR-C | (this paper) | | 0.18 | 0.08 | 0.04 |
| CR-A | (this paper) | | **0.08** | **0.03** | **0.02** |

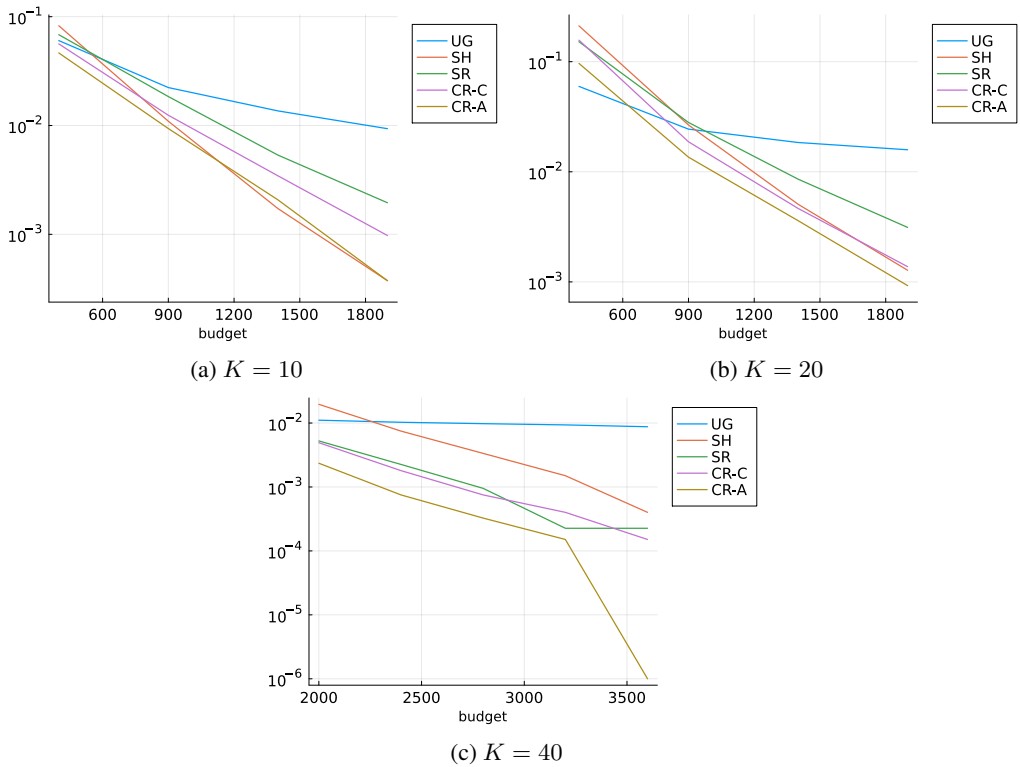

(a) $K = 10$

(b) $K = 20$

(c) $K = 40$

Figure 8: (Concave arm-to-reward function) error probabilities averaged over $40,000$ independent runs.

### K.5 Convex arm-to-reward function

In this instance, we set $\mu_k = \frac{3}{10(k+1)}$ for $k = 1, \cdots, K$. Although SR sometimes does better than CR-C, CR-C becomes better than SR when there is more budget given. This confirms our theoretical analysis for CR-C (see Theorem 7).

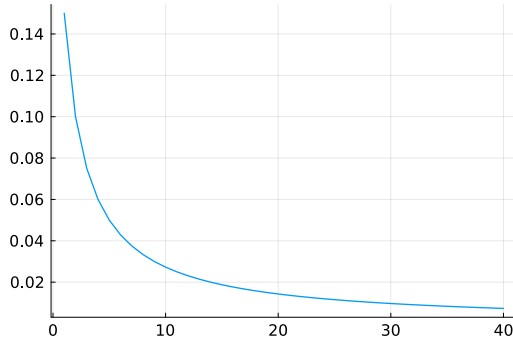

Figure 9: (Convex arm-to-reward function) $\boldsymbol{\mu}$ with $K = 40$.

Table 6: (Convex arm-to-reward function) error probability (in %).

| $K = 10$ | | | $T = 1,500$ | $T = 2,000$ | $T = 2,500$ |
|---|---|---|---|---|---|
| UGapE | (Gabillon et al., 2012) | | 13.48 | 10.08 | 7.77 |
| SH | Karnin et al. (2013) | | 4.93 | 2.16 | 0.93 |
| SR | Audibert et al. (2010) | | 3.15 | 1.45 | 0.83 |
| CR-C | (this paper) | | 3.12 | 1.47 | 0.65 |
| CR-A | (this paper) | | **2.99** | **1.27** | **0.62** |
| $K = 20$ | | | $T = 3,000$ | $T = 3,500$ | $T = 4,000$ |
| UGapE | (Gabillon et al., 2012) | | 10.67 | 8.95 | 7.76 |
| SH | Karnin et al. (2013) | | 2.91 | 1.66 | 0.99 |
| SR | Audibert et al. (2010) | | 0.96 | 0.55 | 0.29 |
| CR-C | (this paper) | | 0.93 | 0.56 | 0.28 |
| CR-A | (this paper) | | **0.79** | **0.39** | **0.21** |
| $K = 40$ | | | $T = 4,400$ | $T = 5,200$ | $T = 6,000$ |
| UGapE | (Gabillon et al., 2012) | | 12.16 | 9.95 | 8.28 |
| SH | Karnin et al. (2013) | | 3.68 | 2.11 | 1.44 |
| SR | Audibert et al. (2010) | | **1.21** | 0.58 | 0.30 |
| CR-C | (this paper) | | 1.28 | **0.49** | **0.25** |
| CR-A | (this paper) | | 1.34 | 0.60 | 0.29 |

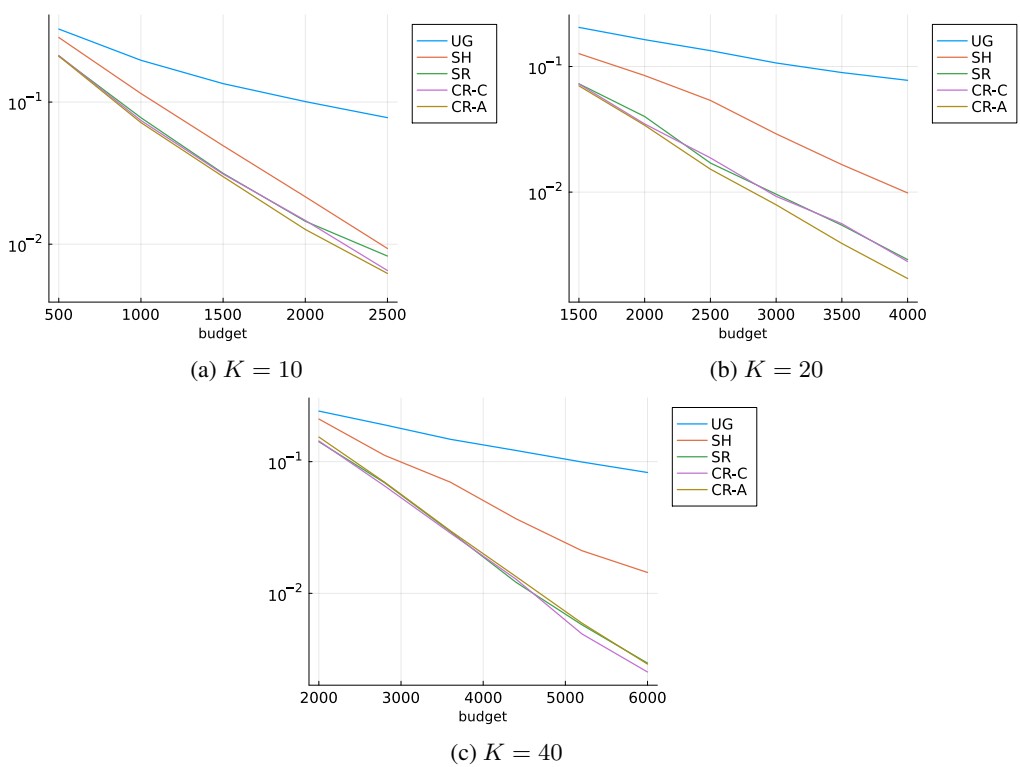

Figure 10: (Convex arm-to-reward function) error probabilities averaged over $40,000$ independent runs.

### K.6 Stair arm-to-reward function

In this instance, we consider $M \in \{5, 6, 10\}$ and a $M(M+1)/2$-dimensional vector $\boldsymbol{\mu}$. For each $M$, we define $\boldsymbol{\mu}$ as: for all positive integers $m$ smaller than $M$, there are $m$ arms on the same level with value, $\frac{3}{4} \cdot 3^{-\frac{m}{M}}$. For example, we plot the values for $M = 10$ (hence $K = 55$) in Figure 11. One can see in this instance, our algorithms are by far stronger than the others.

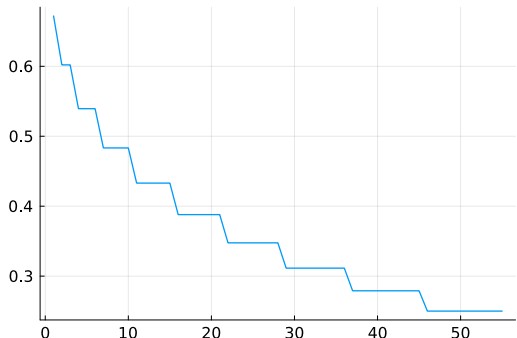

Figure 11: (Stair arm-to-reward function) $\boldsymbol{\mu}$ with $K = 55$.

Table 7: (Stair arm-to-reward function) error probability (in %).

| $K = 15$ | | | $T = 1,600$ | $T = 2,000$ | $T = 2,400$ |
|---|---|---|---|---|---|
| UGapE | (Gabillon et al., 2012) | | 12.00 | 9.89 | 8.47 |
| SH | Karnin et al. (2013) | | 1.47 | 0.72 | 0.38 |
| SR | Audibert et al. (2010) | | 0.43 | 0.19 | 0.06 |
| CR-C | (this paper) | | 0.36 | 0.10 | **0.02** |
| CR-A | (this paper) | | **0.20** | **0.07** | 0.04 |
| $K = 21$ | | | $T = 1,500$ | $T = 2,000$ | $T = 2,500$ |
| UGapE | (Gabillon et al., 2012) | | 17.44 | 13.97 | 12.08 |
| SH | Karnin et al. (2013) | | 4.31 | 1.77 | 0.83 |
| SR | Audibert et al. (2010) | | 2.27 | 0.92 | 0.32 |
| CR-C | (this paper) | | 1.68 | 0.55 | 0.24 |
| CR-A | (this paper) | | **1.14** | **0.35** | **0.09** |
| $K = 55$ | | | $T = 3,000$ | $T = 4,000$ | $T = 5,000$ |
| UGapE | (Gabillon et al., 2012) | | 24.74 | 21.29 | 18.91 |
| SH | Karnin et al. (2013) | | 10.23 | 5.87 | 3.23 |
| SR | Audibert et al. (2010) | | 5.55 | 2.80 | 1.26 |
| CR-C | (this paper) | | 7.10 | 2.58 | 1.05 |
| CR-A | (this paper) | | **4.70** | **1.62** | **0.57** |

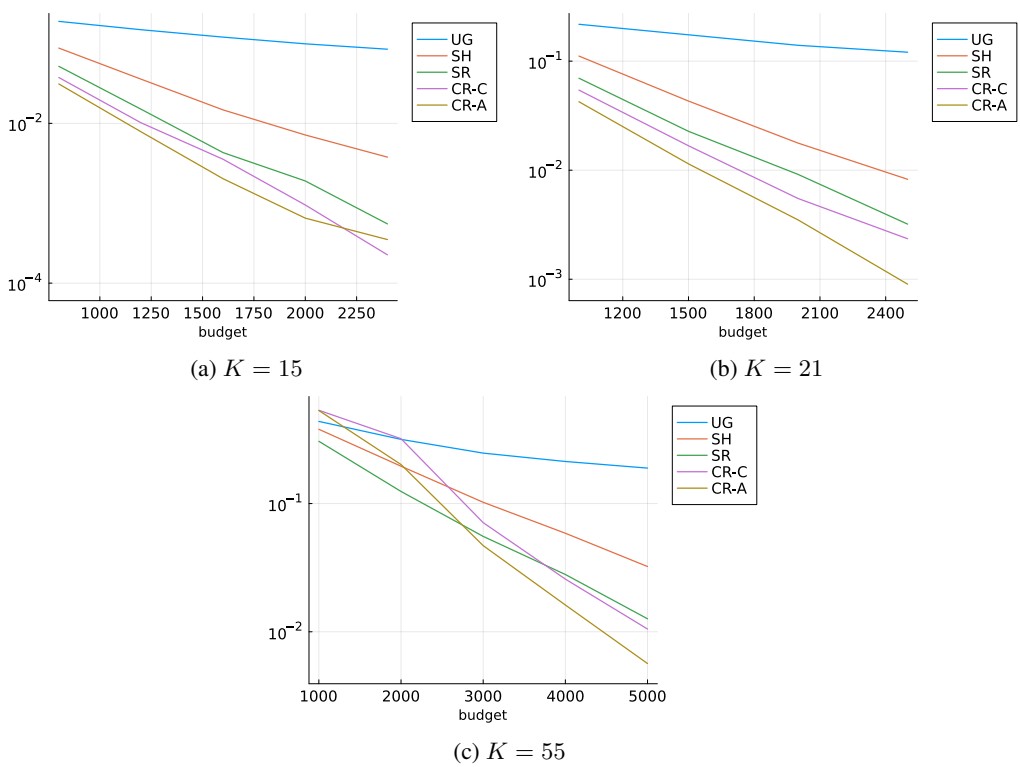

(a) $K = 15$        (b) $K = 21$

(c) $K = 55$

Figure 12: (Stair arm-to-reward function) error probabilities averaged over $40,000$ independent runs.