# OpenReview forum: "Best Arm Identification with Fixed Budget: A Large Deviation Perspective"
_NeurIPS.cc/2023/Conference — NeurIPS 2023 spotlight_

### Official Review · Reviewer_ooBf · 2023-06-23

**Soundness:** 4 excellent
**Presentation:** 4 excellent
**Contribution:** 4 excellent
**Rating:** 8
**Confidence:** 3

**Summary:**

The authors study the problem of best arm identification with fixed budget in the $K$-arm bandit problem. Using tools from the large deviation literature, they propose a new methodology to analyse algorithms for this problem. They apply this method to a state of the art algorithm and obtain sharper asymptotical bounds on the error probability of this algorithm. They propose two new algorithms and establish bounds on their error probability, showing that one of them is asymptotically better that existing algorithms. Finally, they illustrate the good performances of their algorithms by extensive numerical simulations.

**Strengths:**

Clarity : This article is very well written and enjoyable to read. The literature review is complete and well presented. Methods and results are clearly introduced and discussed, and the sketches of proofs are insightful. The level of detail in the evidence greatly helps the reader, and the authors' efforts to make this article self-contained are commendable.

Quality : This article introduces important new methodological tools and theoretical results, that are supported by rigorous proofs.

Originality : The authors study a central problem in the bandit literature from a new perspective, leveraging tools from large deviation literature to obtain sharper results and develop new algorithms.

Significance : The error bounds in this paper improve over existing results on a central question, and the methodology used to obtain them could be applied to study other related problems.

**Weaknesses:**

Why the authors discuss the motivations behind the two algorithms they introduce, they do not compare their theoretical guarantees.

**Questions:**

Could you please discuss more the theoretical guarantees of the two algorithms? Is there settings where one would perform better than the other, and vice-versa?

Could you perhaps add an introduction to your Appendix, to present its content and the different aspects of the problem that have been postponed there?

**Limitations:**

The authors adequately address the limitations of their work.

---

> ### Author Rebuttal · Authors · 2023-08-09
>
> Many thanks for your careful review and positive feedback.
> 1. About
> > Could you please discuss more the theoretical guarantees of the two algorithms? Is there settings where one would perform better than the other, and vice-versa?
>
> In general, we cannot say that one of our two algorithms, CR-A or CR-C, is better than the other. We illustrate this below for the instances presented in Appendix J.1 and J.4. For the instances in J.1, we can easily prove that the performance guarantees for CR-C is better than that those for CR-A. On the contrary, for the instances in J.4, CR-A has better performance guarantees than CR-C. We will add the above discussion to the paper. We introduced these two algorithms, CR-A and CR-C, because we wanted to design algorithms that discard arms with different levels of aggressivity.
>
> 2. About
> > Could you perhaps add an introduction to your Appendix, to present its content and the different aspects of the problem that have been postponed there?
>
>
> Thanks for this suggestion. We agree that it could be nice to have a summary of the appendices. We will add an introduction before the appendices to state the role and content of each of them.

---

> > ### Comment · Reviewer_ooBf · 2023-08-14
> >
> > Thank you for this reply. I have read the rebuttal and will keep the score as it is.

---

### Official Review · Reviewer_64fe · 2023-07-04

**Soundness:** 4 excellent
**Presentation:** 3 good
**Contribution:** 3 good
**Rating:** 6
**Confidence:** 3

**Summary:**

This paper studies the problem of best arm identification (BAI) with fixed budget (FB), with bandit feedback on K arms. The asymptotically optimal sample complexity for this problem has been of significant interest recently and has proved surprisingly difficult to understand compared to the fixed confidence problem.

This paper gives a new upper bound on the failure probability of a given algorithm, and uses it to study specific algorithms: successive rejects and continuous rejects.



**Strengths:**

The paper achieves new large deviation bounds for the failure probability of specific algorithms using a new LDP bound, and shows their superior empirical performance at minimizing failure probability. The analysis of the successive rejects algorithm is nice and improves over previous results. The required analyses for the continuous rejections (in the appendix) seem to be quite involved, but show even better performance guarantees.

The writing is pretty good too.

**Weaknesses:**

The result seem a little incremental as they most concern specific algorithms; the paper does not seem like it will help much toward an eventual solution to  the general FB-BAI problem. But getting better non-sharp bounds is still nice (and a general solution might be abstract enough not to subsume them).

**Questions:**

I didn't see the relevant Polish spaces specified anywhere.

I suggest using i_* instead of \hat i, since it is hard to see the latter.

---

> ### Author Rebuttal · Authors · 2023-08-09
>
> Many thanks for your careful review and positive feedback.
>
> 1. About
> > The result seems a little incremental as they most concern specific algorithms; the paper does not seem like it will help much toward an eventual solution to the general FB-BAI problem. But getting better non-sharp bounds is still nice (and a general solution might be abstract enough not to subsume them).
>
> We believe that our general LDP result (Theorem 1) is not incremental, for it allows us to analyze in a simple way various algorithms and to design new algorithms with improved performance. It allows us to analyze truly adaptive algorithms, and we plan to search for even more adaptive algorithms than CR. Now, solving the general FB-BAI problem (finding a problem-specific lower bound of the error rate and a matching algorithm) seems actually very hard if not impossible. This was recently discussed in Degenne [12] (reference is from the supplementary material).
>
> 2. About
> > I didn't see the relevant Polish spaces specified anywhere.
>
> We mentioned Polish spaces (separable and complete metric spaces) because this is the traditional (and the most general) framework to define and work with Large Deviation principles (see the book [30]). Here, you are right, we just work with Euclidian spaces; we will clarify this.

---

> > ### Comment · Reviewer_64fe · 2023-08-14
> >
> > Thanks for the reply. I don't have further questions and will keep my score as is for now.

---

### Official Review · Reviewer_zb7g · 2023-07-04

**Soundness:** 3 good
**Presentation:** 3 good
**Contribution:** 2 fair
**Rating:** 6
**Confidence:** 1

**Summary:**

This paper considers the problem of best arm identification with fixed budget, where the learner pulls an arm in each round for $T$ rounds, then outputs a candidate for the arm with the largest mean. The objective is to minimize the probability of mis-identification. Characterizing the instance specific complexity for this problem is an open question. The authors present a more refined analysis based on large deviation principles resulting on a refined upper bound for the successive rejects (SR) algorithm. Further, the authors present two algorithms based on the previous analysis where arms elimination is done following conditions rather than at pre-specified rounds as in SR.

**Strengths:**

The paper is well-organized, and the discussions are interesting. Using LDP for best arm identification appears to be novel.

**Weaknesses:**

The presented bounds are not easy to read (especially in Section 4). It would be nicer to provide a more detailed discussion about the comparisons with bounds in the literature.

Some theorem statements (Proposition and Theorem 1) don't clearly state the assumptions made.

**Questions:**

Minor comment/question: line 147, about returning the arm with the highest empirical reward: can't we have a scenario in SR algorithm where the at the end (when the budget is exhausted), the empirical mean of the wining arm is smaller than the empirical mean of the first eliminated arm (which was computed only during the first epoch) ?  If true, the latter statement is not valid.

**Limitations:**

---

> ### Author Rebuttal · Authors · 2023-08-09
>
> Many thanks for your careful review and positive feedback.
>
> 1.	About
> > The presented bounds are not easy to read (especially in Section 4). It would be nicer to provide a more detailed discussion about the comparisons with bounds in the literature.
>
> Thanks for this remark. We will add a more detailed discussion to compare our bounds to those of the literature (using the extra space allowed in the final version of the paper). From the results of Theorem 3, we can conclude that CR-C has better performance guarantees than SR, the state-of-the-art algorithm. Indeed, one can readily see that the bound derived in Theorem 3 for CR-C is higher than $2\xi_j/j\overline{\log}K$, which is the improved bound for SR presented in Section 3. We will make this clear in the paper.
>
> In general, we cannot say that one of our two algorithms, CR-A or CR-C, is better than the other. We illustrate this below for the instances presented in Appendix J.1 and J.4. For the instances in J.1, we can easily prove that the performance guarantees for CR-C is better than that those for CR-A. On the contrary, for the instances in J.4, CR-A has better performance guarantees than CR-C. We will add the above discussion to the paper.
>
>
>
> 2. About
> > Minor comment/question: line 147, about returning the arm with the highest empirical reward: can't we have a scenario in SR algorithm where the at the end (when the budget is exhausted), the empirical mean of the wining arm is smaller than the empirical mean of the first eliminated arm (which was computed only during the first epoch) ? If true, the latter statement is not valid.
>
> Yes, you are right. More precisely, in Line 147, we state that *if* the algorithm returns the arm with the highest empirical reward, then the error probability is $P_{\mu}[\hat{\imath}\neq 1(\mu)]=P_{\mu}[\hat{\mu}(T)\in Alt(\mu)]$. As you noticed, algorithms (SR, SH, or CR) may not always return the best empirical arm, but we accounted for this possibility in our analysis. To understand why, please refer to the proof of Theorem 2 for example: the analysis consists in upper bounding the probability of eliminating the best arm in each phase of the algorithm. This probability is first connected to an event related to the empirical rewards of the arms, and then the probability of this event is bounded using Theorem 1 (our Large Deviation result).

---

> > ### Comment · Reviewer_zb7g · 2023-08-14
> >
> > Thank you for the reply. I have read the rebuttal and I will not change the score for now.

---

### Official Review · Reviewer_vyXZ · 2023-07-05

**Soundness:** 4 excellent
**Presentation:** 4 excellent
**Contribution:** 4 excellent
**Rating:** 7
**Confidence:** 4

**Summary:**

This paper contributes an important tool geared towards closing the performance gap for best arm identification problems under the fixed budget setting, a well-known open problem in the area. In particular, the result is a large deviation bound on the sample means of the arm rewards as a function of any large deviation bound on the empirical means of the number of times each of the arms is pulled under a fixed policy (which could very well be adaptive). Since for popular policies such as successive rejects, the latter bound could be easier to specify due to their structure (batched pulls, etc.), the result can be used to derive tighter upper bounds on the probabilities of misidentification under such policies. The paper illustrates this power by deriving a tighter upper bound on the probability of misidentification for the successive rejects (SR) policy and derives a new policy called continuous rejects (CR) that achieves a smaller probability of error.

**Strengths:**

-  While it doesn't seem surprising that such a result would hold, I think this is a solid result that I anticipate being of importance not only for the best arm-identification problem but as a general tool in the bandit analysis toolkit.
- The paper is very well written.

**Weaknesses:**

- I don't think there are any major weaknesses except stating that the considered problem is the "last fundamental open problem in MABs" is quite presumptuous.

**Questions:**

Hasn't it been shown, as stated in the open problem paper by Chao Qin, that the conjecture in Equation (1) is false? Am I missing something?

**Limitations:**

The paper addresses a specific problem in bandits that does not seem to have any more nefarious use cases than typical optimization models.

---

> ### Author Rebuttal · Authors · 2023-08-09
>
> Many thanks for your careful review and positive feedback.
>
> 1. About
> > I don't think there are any major weaknesses except stating that the considered problem is the "last fundamental open problem in MABs" is quite presumptuous.
>
> We agree and we will rephrase.
>
> 2. About
> > Hasn't it been shown, as stated in the open problem paper by Chao Qin, that the conjecture in Equation (1) is false? Am I missing something?
>
> (1) is the lower bound conjectured by Kaufmann and Garivier in [17] (references are from the supplementary material). No one has proven yet whether (1) is a correct lower bound or not (as far as we are aware). Now the conjecture stated by Chao Qin in [28] (Conjecture 2 in [28]) is whether there exists a single algorithm with error rate matching the r.h.s. of (1) for *all* instances. This last conjecture does not hold as mentioned in [28] – this is a consequence of results from [1]. Recently, Degenne [12] (published in COLT 2023) also proved that the conjecture was false for 2-arm bandit problems with Bernoulli rewards.

---

> > ### Comment · Reviewer_vyXZ · 2023-08-17
> >
> > Thanks for the clarification. I am in support of the contribution and will maintain my score.

---

### Decision · Program_Chairs · 2023-09-21

**Decision:**

Accept (spotlight)

**Comment:**

The reviewers came to consensus that this paper makes a good progress on the problem of fixed-budget best arm identification. I agree with these opinions and please polish the manuscript addressing the minor concerns raised by the reviewers (mostly on presentation).